# A polymer tethering strategy to achieve high metal loading on catalysts for Fenton reactions

Lixin Wang[1,4], Longjun Rao[1,4], Maoxi Ran[2], Qikai Shentu[1], Zenglong Wu[1], Wenkai Song[1], Ziwei Zhang[1], Hao Li[1], Yuyuan Yao[1,3], Weiyang Lv [1,3,5] ✉ & Mingyang Xing [1,2,5] ✉

The development of heterogenous catalysts based on the synthesis of 2D carbon-supported metal nanocatalysts with high metal loading and dispersion is important. However, such practices remain challenging to develop. Here, we report a self-polymerization confinement strategy to fabricate a series of ultrafine metal embedded N-doped carbon nanosheets (M@N-C) with loadings of up to 30 wt%. Systematic investigation confirms that abundant catechol groups for anchoring metal ions and entangled polymer networks with the stable coordinate environment are essential for realizing high-loading M@N-C catalysts. As a demonstration, Fe@N-C exhibits the dual high-efficiency performance in Fenton reaction with both impressive catalytic activity (0.818 min$^{-1}$) and $H_2O_2$ utilization efficiency (84.1%) using sulfamethoxazole as the probe, which has not yet been achieved simultaneously. Theoretical calculations reveal that the abundant Fe nanocrystals increase the electron density of the N-doped carbon frameworks, thereby facilitating the continuous generation of long-lasting surface-bound ·OH through lowering the energy barrier for $H_2O_2$ activation. This facile and universal strategy paves the way for the fabrication of diverse high-loading heterogeneous catalysts for broad applications.

Hydroxyl radicals (·OH), which own supreme oxidation potential and non-selective reaction capability, have played a pivotal role in environmental remediation, organic synthesis and biomedical fields in the past century[1–3]. As the benchmark process to produce ·OH, Fenton and Fenton-like reactions based on transition metal activators have been intensively explored due to their rapid conversion rate and moderate operating conditions[4]. In particular, transition metal nanoparticles (NPs) with high surface-to-volume ratios and unique electronic structures can appreciably augment active sites and regulate intrinsic

activity[5,6], thereby boosting the cleavage of $H_2O_2$. Nevertheless, the potential application of transition metal NPs remains challenging in view of easy conglomeration and uncontrollable passivation during either the synthesis procedure or the subsequent reaction process[7,8], thus deteriorating the Fenton reaction. Besides the catalytic activity, it is often overlooked that only 30% of $H_2O_2$ is utilized in the conventional Fenton process[9], mainly resulting from the self-quenching of short-lived ·OH in solution as well as the undesired reaction between ·OH and metal ions[10,11], which further hinder the technology from large-

[1]National Engineering Lab of Textile Fiber Materials & Processing Technology (Zhejiang), Zhejiang Sci-Tech University, Hangzhou 310018, China. [2]National Engineering Research Center of Industrial Wastewater Detoxication and Resource Recovery, School of Resources and Environmental Engineering, East China University of Science and Technology, 130 Meilong Road, Shanghai 200237, China. [3]Zhejiang Provincial Innovation Center of Advanced Textile Technology, Shaoxing 312000, China. [4]These authors contributed equally: Lixin Wang, Longjun Rao. [5]These authors jointly supervised this work: Weiyang Lv, Mingyang Xing. ✉e-mail: wylv@zstu.edu.cn; mingyangxing@ecust.edu.cn

scale implementation. To revive the century-old reaction for addressing emerging environmental concerns and other chemical domains, breakthroughs in constructing novel Fenton catalysts capable of continuous generation and efficient usage of ·OH are urgently demanded.

Recent advancements in nanotechnology and nanomaterials have offered great opportunities for the rational design of high-performance Fenton catalysts. Among them, encapsulating metal NPs into the nitrogen-doped carbon matrix (denoted as M@N-C) has emerged as the most promising strategy since it can not only elevate the metal stability against passivation and leaching, but also avoid the agglomeration of neighboring metal NPs[12,13], which enables the rapid and long-lasting activation of $H_2O_2$. A key concept of constructing M@N-C hybrids with optimal catalytic activity is to take advantage of the size effect and metal loading on the support. Thus far, a variety of approaches have been developed for the fabrication of M@N-C catalysts and can be mainly classified into top-down and bottom-up strategies. Generally, for the top-down route, metal-containing molecules are spatially confined into the structural or electronic defects of existing carbon materials such as graphene sheets, carbon nanotubes and biochar[14–16], followed by the chemical reduction process to obtain the target catalysts (Supplementary Fig. 1a). However, due to the limited vacant sites on the conventional carbon matrix, the metal loading should be strictly controlled to circumvent undesired metal aggregation. For instance, Hai et al. recently demonstrated that simply annealing the carbon nitride and Ni precursor led to the formation of metal NPs with sizes ranging from the sub-nanometer to 100 nm when the metal loading increased from 1 to 10 wt%[17]. On the other hand, the bottom-up approaches generally involve a high-temperature pyrolysis process comprising metal, nitrogen and carbon sources (including pyrrole, urea and melamine)[18–20]. Although the small organic molecules are able to supply abundant anchoring sites for high loadings of metal ions, the drastic structural evolution of these complexes, which lack a sufficient confinement effect, inevitably leads to metal aggregation and formation of bulk composites during the annealing procedure (Supplementary Fig. 1b)[21]. A prominent study by Chen et al. showed that the direct thermolysis of metal-organic framework precursors yielded the Co NPs over 30 nm in size, although the metal content reached 20 wt%[22].

Despite the tremendous efforts made in material design, concurrently achieving the goals of both high dispersion and high metal loading (>20 wt%, similar to commercial Pt/C catalysts) on M@N-C composites remains a significant challenge. The major obstacle is the lack of sufficient confinement in the routine precursors, which fail to anchor large amounts of metal ions and prevent their migration at high temperatures. This dilemma restricts the further promotion of the overall catalytic activity[23]. Another scarcely addressed puzzle is the mass production of such materials, which is essential for practical applications[24,25]. Most strategies for the synthesis of well-dispersed M@N-C rely heavily on the delicate control of support defects and synthetic procedures[26,27]. These generally require substantial cost and time-consuming treatment, hampering expansion to scale up for industrialization. Under these circumstances, it is highly desirable to establish a swift, straightforward and general protocol for the large-scale synthesis of M@N-C catalysts with high metal loading.

Here, a self-polymerization confinement strategy is proposed for the construction of ultrafine metal NPs encased in N-doped two-dimensional (2D) carbon nanosheets with high metal content via a grind-assisted pyrolysis method. Specifically, dopamine (DA), a mussel-inspired material featuring abundant chelating groups, self-polymerization capability, and tunable assembled structure[28,29], is intentionally selected as the carbon and nitrogen precursor for the subsequent thermolysis treatment. The key point of this approach lies in that DA with catechol groups can anchor large numbers of metal ions and then undergo a fast self-polymerization process to provide a stable coordinate environment, thus suppressing metal aggregation and guaranteeing the in situ formation of well-dispersed metal NPs under carbothermal reduction. This method can be easily scaled up and enables the synthesis of various M@N-C composites (M = Fe, Co, Cu, Ni and Ag) with a high metal loading of up to 30 wt%. As a demonstration, Fe@N-C exhibits superior catalytic performance with the $H_2O_2$ utilization efficiency as high as 84.1% in treating refractory organic pollutants, far outperforming the previously reported heterogeneous catalysts. Detailed experimental results and density functional theory (DFT) analysis revealed that the high activity of Fe@N-C stems from the Fe NPs boosted pyridinic-N sites, which can induce electron transfer from catalyst to $H_2O_2$ for the continuous generation of surface-bound ·OH. This study opens new avenues for efficiently designing high-performance catalysts for the Fenton reaction and other related chemical processes.

## Results
### Synthesis of Fe@N-C with high metal loading
To demonstrate the self-polymerization confinement strategy, Fe@N-C was first synthesized as a proof-of-concept sample via a grind-assisted pyrolysis method. As shown in Fig. 1a, iron (III) chloride hexahydrate ($FeCl_3 \cdot 6H_2O$) was mixed with DA in a mortar and vigorously ground. Over time, the visible color and phase change appeared (Supplementary Fig. 2). Only after 5 min, the mixture transformed into a viscous black paste due to the polymerization of dopamine, which was supported by the gel permeation chromatography (GPC) test. The result showed that the weight-average molecular weight ($M_W$) of polydopamine (PDA) was $1.48 \times 10^4$ (Supplementary Fig. 3), consistent with the previous report[30]. In this process, ferric ions could be coordinated with the abundant catechol groups in PDA (Supplementary Fig. 4), facilitating the in situ generation of well-dispersed Fe NPs in the subsequent heat treatment. The essence of our methodology lays in the adoption of DA incorporated with $FeCl_3 \cdot 6H_2O$, in which the $FeCl_3 \cdot 6H_2O$ with intrinsic moisture adsorption (Supplementary Fig. 5) and oxidizing capability (Supplementary Fig. 6) could induce the rapid polymerization of DA without additional solvents. Subsequently, the self-assembled layered structure of PDA provided a stable coordinate environment for large loadings of isolated metal atoms (Supplementary Fig. 7), thereby averting the significant structural evolution of metal-organic precursors and guaranteeing the successful preparation of ultrafine Fe NPs with a high content on the carbon support.

Visualization of scanning electronic microscopy (SEM) and atomic force microscopy (AFM) revealed the 2D sheet-like morphology of Fe@N-C with a thickness as thin as 20 nm (Fig. 1b, c), which was inherited from the layered structure of PDA[31,32]. Transmission electronic microscopy (TEM) images of Fe@N-C showed the homogenous dispersion of Fe NPs with an average size of 4 nm on the nanosheet (Fig. 1d, e). Moreover, a further high-resolution TEM (HRTEM) image demonstrated the well-crystallized nanoparticle with lattice fringes of 0.206 nm corresponding to the (110) plane of metallic iron (Fig. 1f). The overall iron loading was determined to be 30.5 wt% by inductively coupled plasma optical emission spectrometry (ICP-OES), representing several-fold improvements over previous well-dispersed M@N-C materials[13,16]. In addition, the XPS depth profiles in Supplementary Fig. 8 showed that the Fe signals gradually increased with prolonged etching times, indicating the existence of protective carbon layers coated on the Fe NPs[33]. This structure could also be confirmed by HRTEM images. As shown in Supplementary Fig. 9a, the dispersed Fe NPs were covered by brighter carbon layers, which could be evidenced by the lattice fringes of 0.335 nm corresponding to the (002) crystal plane of graphitic carbon. Moreover, the well-protected Fe NPs could be clearly observed at the edge of the carbon nanosheet (Supplementary Fig. 9b), confirming that the Fe NPs were coated by thin carbon layers. Such a 2D sheet-like hybrid structure not only effectively protects Fe NPs from surface passivation, but also provides abundant accessible active sites to minimize the mass/charge transfer

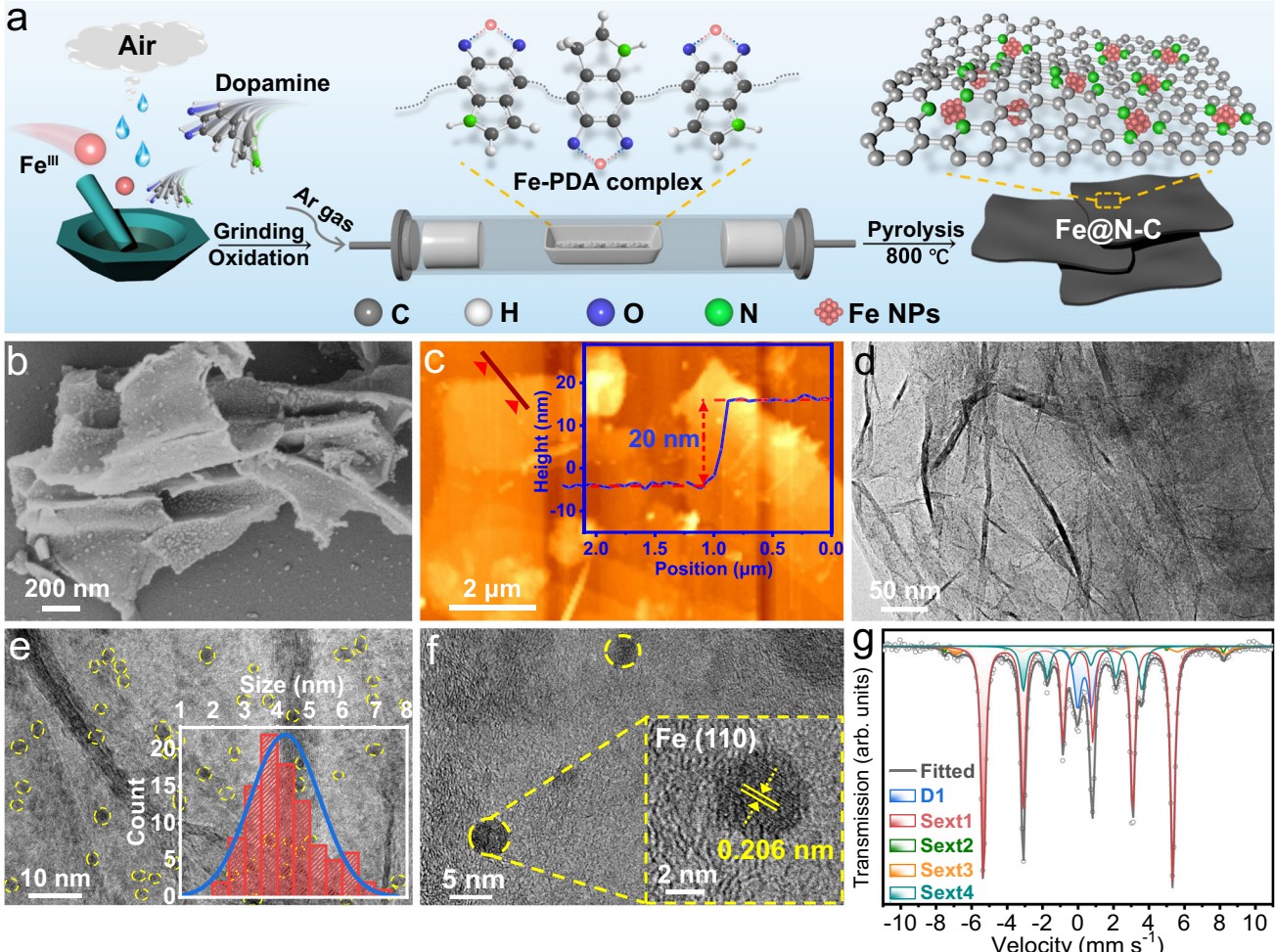

**Fig. 1 | Synthesis and structural characterization of Fe@N-C. a** Schematic illustration of the Fe@N-C synthesis procedure. **b** SEM image of Fe@N-C-800. **c** AFM image of Fe@N-C-800 nanosheet, and the inset shows the thickness of the selected position. **d, e** TEM images of Fe@N-C-800 nanosheet and the inset in (**e**) is the size distribution of nanoparticles. **f** HRTEM image of Fe@N-C-800. **g** $^{57}$Fe Mössbauer spectrum of pristine Fe@N-C-800 measured at room temperature. Panel (**a**) was generated by the Cinema 4D software.

distance[34,35], which is essential for achieving efficient and sustainable catalytic performance in $H_2O_2$ activation.

To further analyze the Fe species in Fe@N-C, $^{57}$Fe Mössbauer spectroscopy measurement was conducted. As shown in Fig. 1g, the Mössbauer spectrum of pristine Fe@N-C could be deconvoluted into four sextets and one doublet, which corresponded to four spectral components (Supplementary Table 1). The major sextet (Sext1), with a high hyperfine magnetic field (330.65 KOe) and a zero-isomer shift, should be assigned to α-Fe with a relative content of 64.0%[36], while the other minor sextets were close to $Fe_3O_4$ and iron carbide with a total relative content of 23.7%[37,38]. In addition, a ferric high-spin doublet component (D1) with a relatively high quadrupole splitting value was detected in the spectrum with a relative content of 12.3%[39]. Accordingly, we concluded that α-Fe was the dominant component in Fe@N-C.

The successful preparation of carbon-supported ultrafine Fe NPs with high metal content mainly benefited from the following desirable characteristics of DA. The first is the chelating effect. The strong chelation between the catechol groups of DA and metal ions provided the first line of protection, enabling the stabilization of isolated metal atoms with large loadings. In the control experiment, iron (III) acetylacetonate [Fe(acac)$_3$] was chosen as a substitute for FeCl$_3$·6H$_2$O since the strong interaction between iron (III) and acetylacetone would replace the chelation of catechol group and iron (III). As expected, the

severe aggregation of NPs with an average size of over 20 nm was clearly observed after weakening the chelation effect (Supplementary Fig. 10a). In addition, the self-polymerization capability of DA was the secondary protection, providing a stable coordinate environment to avoid metal aggregation under annealing. For comparison, the contact between DA and oxidant was cut to prevent the polymerization process (Supplementary Fig. 11, details in Supplementary Information). Without the protection of the polymer support, the measured diameter of Fe NPs reached an astonishing value of 130 nm (Supplementary Fig. 10b). These observations revealed that the abundant catechol groups for anchoring metal ions and the entangled polymer networks with a stable coordinate environment played two indispensable roles in achieving high-loading Fe@N-C catalysts with well-distributed metal NPs.

To monitor the synthesis transition of the Fe-PDA complex into the Fe@N-C catalyst, characterizations including X-ray diffraction (XRD), TEM, HRTEM, thermogravimetry analysis, Raman spectroscopy and $N_2$ adsorption-desorption measurements were performed for determining the structural evolution under different pyrolysis temperatures. It was found that with an increase in pyrolysis temperature, iron (III) chloride hexahydrate was first reduced to ferrous chloride and its hydrate, which subsequently evolved into Fe NPs at 800 °C (Supplementary Figs. 12–14). When the temperature was further elevated, the composition of the materials remained constant but the NPs

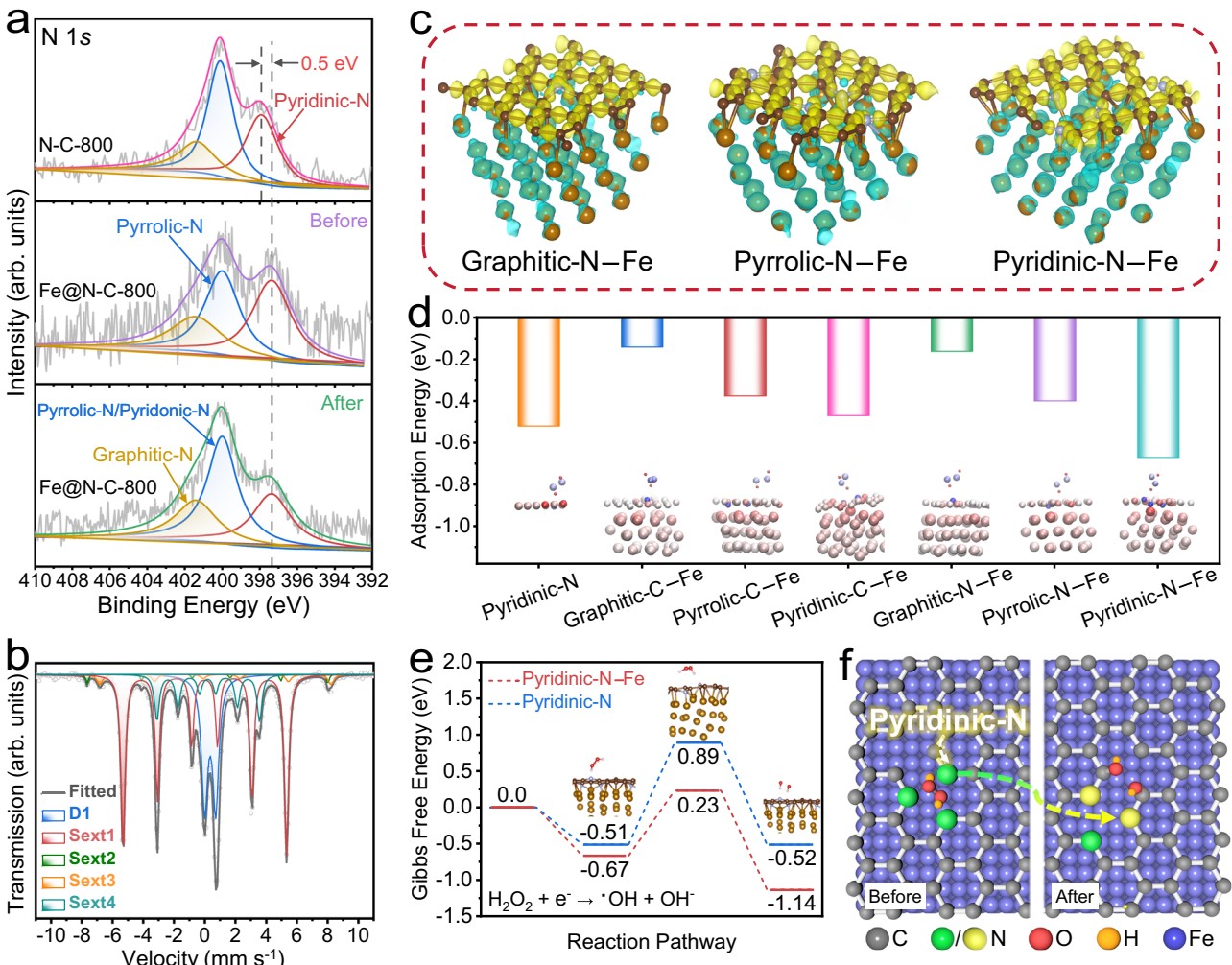

**Fig. 2 | Active center identification through XPS, Mössbauer and DFT calculations. a** High resolution of N 1*s* XPS spectra of N-C-800, Fe@N-C-800 before and after reaction. **b** $^{57}$Fe Mössbauer spectrum of used Fe@N-C-800 measured at room temperature. **c** The charge density differences in graphitic-N–Fe, pyrrolic-N–Fe and pyridinic-N–Fe interfaces, yellow and cyan regions represent electron accumulation and depletion, respectively. **d** The adsorption energy of $H_2O_2$ at different sites on Fe@N-C. Insets are electron transfer diagram of Bader charge analysis and the blue and red colors represent electron gain and loss. **e** Calculated Gibbs free energy for the $H_2O_2$ to ·OH on pyridinic-N-Fe and pyridinic-N sites. **f** The schematic illustration of $H_2O_2$ activation at pyridinic-N-Fe site on Fe@N-C.

aggregated excessively owing to the inevitable metal migration under high temperature (Supplementary Fig. 13b–i). In Raman spectroscopy, the relative intensity ratio of the D and G bands ($I_D/I_G$) of the carbon matrix was also evaluated, where the $I_D/I_G$ first increased as the temperature increased from 600 to 800 °C and subsequently decreased at pyrolysis temperatures above 900 °C (Supplementary Fig. 15). A similar trend was also found for the specific surface area of the derived Fe@N-C-X (X represents the pyrolysis temperature), where the highest BET surface area (194.42 m² g⁻¹) was found at 800 °C (Supplementary Fig. 16 and Supplementary Table 2). In short, these results demonstrated that the samples calcined under 800 °C exhibited the optimal morphology of NPs as well as a relatively high specific surface area, which were highly desirable for $H_2O_2$ activation.

### Active center identification over Fe@N-C catalyst

To better reveal the active sites for the Fenton reaction in Fe@N-C-800, XPS was performed to investigate the chemical compositions of the catalyst and its variations after the reaction. As shown in Fig. 2a, the N 1*s* signal for Fe@N-C-800 could be deconvoluted into three peaks at 397.4, 400.1 and 401.5 eV, which were assigned to pyridinic-N, pyrrolic-N, and graphitic-N, respectively[40]. Compared to the N-C-800, a negative shift in the binding energy for pyridinic-N was observed,

suggesting that the introduced Fe NPs could donate electrons to the N-doped carbonaceous matrix and increase the charge density on pyridinic-N moieties[41,42]. Moreover, both N-C-800 and Fe@N-C-800 catalysts had similar N contents around 3.0 at%, while the presence of Fe NPs could significantly regulate the N configuration, where the relative content of pyridinic-N was increased from 34.1 to 40.4%, indicating that Fe NPs had a stabilization effect on pyridinic-N moieties during high-temperature calcination[23]. After the reaction, the pyridinic-N content showed a marked decrease (40.4–29.7%), while the pyrrolic-N content increased significantly (37.1–46.0%). The notable compositional variation of the pyridinic-N species that was caused by the Fenton reaction might be ascribed to the fact that the carbon atoms next to pyridinic-N could react with ·OH species, inducing the transformation of pyridinic-N to pyridonic-N (400.1 eV)[43,44]. To further verify whether the pyridinic-N sites participated in the Fenton reaction, $H_3PO_4$ was used as the poisoning agent since it could protonate pyridinic-N to form pyridinic-N-H[45]. As expected, a significant inhibition effect was observed in the Fe@N-C-800/$H_2O_2$ system after the addition of $H_3PO_4$ (Supplementary Fig. 17), indicating the crucial role of pyridinic-N site in the Fenton reaction. Furthermore, we compared the $^{57}$Fe Mössbauer spectra of Fe@N-C-800 acquired before and after the Fenton reaction to investigate the transition of Fe NPs (Fig. 2b).

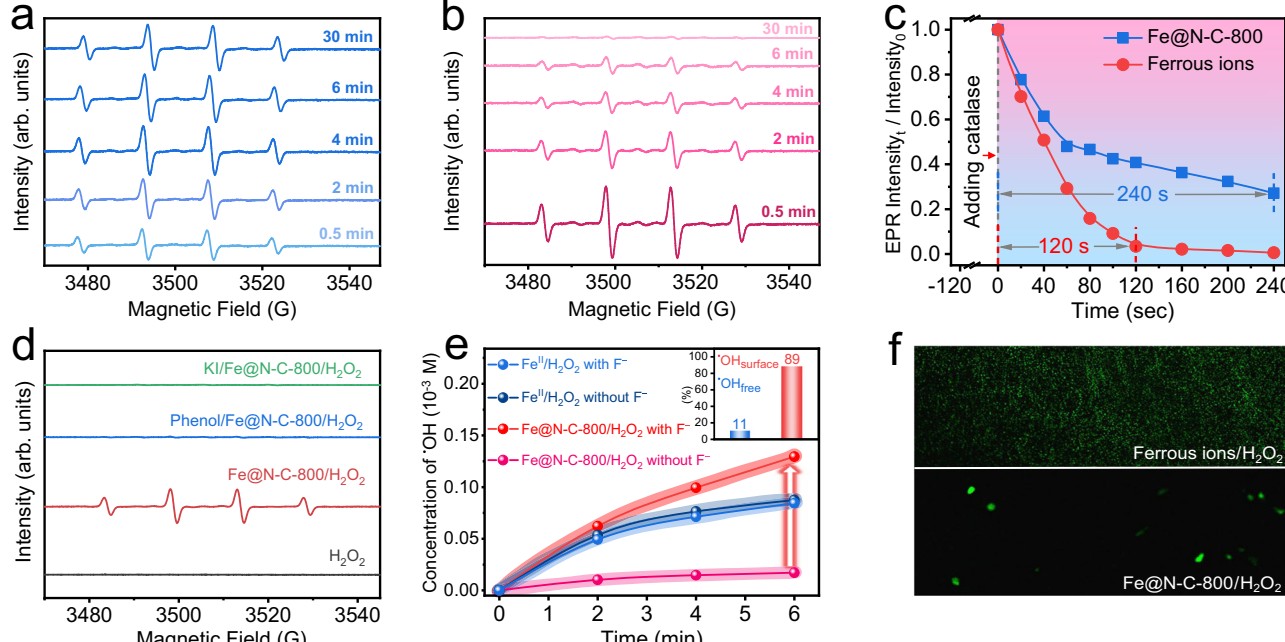

**Fig. 3 | Detection and quantitation of surface-bound ˙OH.** EPR spectra of DMPO-˙OH adduct signal in the two systems: **a** Fe@N-C-800/$H_2O_2$, **b** conventional Fenton. **c** Normalized decaying plot of the DMPO-˙OH adduct EPR signal after the addition of catalase. **d** EPR spectra of DMPO-˙OH adduct signal in the different systems. **e** Quantitative determination of free ˙OH radicals generated in different systems. **f** Fluorescence images of different systems with phthalhydrazide as the chemiluminescence probe.

Notably, the relative proportion of $Fe^{III}$ increased tremendously (from 12.3 to 27.1%) while that of α-Fe markedly decreased (from 64.0 to 51.3%). This phenomenon might be that the encapsulated Fe NPs underwent the in situ oxidation process after donating electrons to the carbon layers. To maintain the electroneutrality, the oxidized Fe ions in the lattice would migrate outward to the surface and react with the permeated $H_2O$ and dissolved $O_2$ to form the iron (oxyhydr) oxide[46]. Consequently, we could surmise that the pyridinic-N cooperating with the Fe site on Fe@N-C-800 was essential for $H_2O_2$ activation.

To further confirm the roles of pyridinic-N and Fe sites, density functional theory (DFT) calculations were conducted to gain insight into $H_2O_2$ activation over Fe@N-C catalyst. Considering that it was challenging to monitor the catalyst structure and construct the corresponding models at every moment during the reaction, we thus constructed the simulation model based on the pristine structure of the catalyst to explore the active sites. To simulate our samples, the model heterointerface was built by exposing the (110) facet of Fe to single-layered graphene because it was one of the most stable facets, as determined by its Wulff structure and clearly observed in the HRTEM image[47,48]. Compared with the carbon structure without Fe NPs (Supplementary Fig. 18), the charge density difference plots displayed in Fig. 2c showed that the embedded Fe NPs could donate electrons to the N-doped carbon layer and significantly modulate the electronic structure on the carbon surface, resulting in an increased charge density for enhanced catalysis. To further explore the favorable active sites within Fe@N-C for $H_2O_2$ activation, the adsorption behaviors of $H_2O_2$ on the different N configurations and their adjacent C atoms were performed, and the optimized configurations were depicted in Supplementary Fig. 19. Notably, the N atoms showed lower adsorption energy than the C atoms irrespective of the N-doping configurations, suggesting that the $H_2O_2$ molecule was more likely to be adsorbed on N atoms than on C atoms (Fig. 2d and Supplementary Table 3). Moreover, the lowest adsorption energy of −0.67 eV was observed for pyridinic-N⁻Fe, indicating preferable $H_2O_2$ accessibility at this site. In addition, electron transfer between Fe@N-C and $H_2O_2$ was observed

by charge density analysis (Supplementary Fig. 20), which indicated that significant electron accumulation was concentrated on the $H_2O_2$ molecule (especially on the O-O bond area) as well as the nitrogen species. Similarly, among all the adsorption models, pyridinic-N⁻Fe exhibited the highest charge density around the $H_2O_2$ molecule, which could also be visualized through Bader charge analysis (inset in Fig. 2d). Combined with the XPS analysis results, we could conclude that the pyridinic-N⁻Fe position, which had the highest affinity to $H_2O_2$, was more conducive to rapid electron transfer between Fe@N-C and $H_2O_2$, thus contributing to the efficient activation of $H_2O_2$.

For a better understanding of the synergistic effect between pyridinic-N and Fe NPs, the reaction pathway for ˙OH generation was further explored. Figure 2e shows the three-step reactions for $H_2O_2$ activation on the pyridinic-N⁻Fe site including $H_2O_2$ adsorption, electron transfer and peroxide bond dissociation. As could be seen, pristine pyridinic-N needed to overcome the activation energy barrier of 1.40 eV for ˙OH formation, while that of pyridinic-N⁻Fe was much lower (0.90 eV), indicating that the introduction of Fe NPs significantly reduced the energy barrier of $H_2O_2$ dissociation and facilitated ˙OH production. Moreover, compared with pristine pyridinic-N, pyridinic-N⁻Fe also showed a higher affinity to both $H_2O_2$ and ˙OH (Supplementary Fig. 21), which was conducive to the efficient activation of $H_2O_2$ and the confinement of ˙OH on the catalyst surface (Fig. 2f). This interaction mode might give rise to the unexpected performance in the following catalytic reaction.

## Detection of surface-bound ˙OH

$H_2O_2$ decomposition on the Fe@N-C-800 catalyst was further investigated through electron paramagnetic resonance (EPR) by detecting the spin-reactive ˙OH using 5,5-dimethyl-1-pyrroline N-oxide (DMPO) as a spin-trapping agent. As shown in Fig. 3a, strong four-line EPR spectra with relative intensity of 1:2:2:1 corresponding to the DMPO-˙OH adduct were observed, indicating the formation of ˙OH by activated $H_2O_2$[49]. Moreover, the intensity of EPR signals in the Fe@N-C-800/$H_2O_2$ system gradually increased at first and remained strong

even when the reaction time lengthened to 30 min. In contrast to the conventional Fenton system, in which the intensity of the same signals fiercely generated at the beginning but immediately went down (Fig. 3b), the long-lasting $\cdot$OH signals in the Fe@N-C-800/$H_2O_2$ system should be ascribed to the successive electron donation by the Fe center to carbon layer for the continuous activation of $H_2O_2$, as confirmed by the Mössbauer spectra and DFT analysis. Besides, the adsorption of $\cdot$OH on the Fe@N-C surface was thermodynamically favorable as predicted by theoretical calculation (with an adsorption energy of −2.47 eV), which might exhibit chemical characteristics different from those of free $\cdot$OH in solution[50]. To confirm this hypothesis, we intentionally adopted the catalase to quench $H_2O_2$ (via $2H_2O_2 \rightarrow 2H_2O + O_2$) and tracked the survival time of $\cdot$OH in different reaction systems (Fig. 3c). Intriguingly, the $\cdot$OH generated by Fe@N-C-800 exhibited a slower decay rate, and its lifetime was two times longer than that of the conventional Fenton system after normalizing the incident EPR signal intensities by the ones obtained after reaction for 2 min. The increased lifetime of $\cdot$OH should increase the reaction probabilities toward substances, leading to high $H_2O_2$ utilization in the practical application.

To clarify the mechanism of these desirable characteristics, the type of $\cdot$OH generated from the Fe@N-C-800/$H_2O_2$ system was further explored. For ease of understanding, we denoted the $\cdot$OH adsorbed on the catalyst as surface-bound $\cdot$OH ($\cdot$OH$_{surface}$), and the $\cdot$OH released into the solution as free $\cdot$OH ($\cdot$OH$_{free}$). As shown in Fig. 3d, the intensity of EPR signals greatly weakened with the addition of KI or phenol (typical surface-bound $\cdot$OH scavengers), which demonstrated that most of the generated ROS might be $\cdot$OH$_{surface}$. To further quantify the ratios of different types of $\cdot$OH in the Fe@N-C-800/$H_2O_2$ system, we added $F^-$ in the reaction system to desorb $\cdot$OH$_{surface}$ by forming a strong $\cdot$OH$_{surface}\cdots F^-$ hydrogen bond (see the Supplementary Information for the detailed procedure). As illuminated in Fig. 3e, the concentration of $\cdot$OH$_{free}$ rapidly increased over Fe@N-C-800 after the addition of $F^-$, whereas that of the conventional Fenton system barely changed. Assuming that all $\cdot$OH$_{surface}$ could be desorbed from Fe@N-C-800 with the introduction of $F^-$, the relative ratio of $\cdot$OH$_{surface}$ to overall $\cdot$OH was calculated to be 89% (inset of Fig. 3e), further confirming the dominant proportion of $\cdot$OH$_{surface}$ in the Fe@N-C-800/$H_2O_2$ system. This conclusion was also supported by the results of fluorescence microscopy (Fig. 3f), in which the fluorescence signals of 5-hydroxy-2,3-dihydro-1,4-phthalazinedione (the product obtained via the reaction between $\cdot$OH and phthalhydrazide) could be viewed in the whole solution of the conventional Fenton system, while only sparsely accumulated signals were observed on the surface of dispersed catalysts in the Fe@N-C-800/$H_2O_2$ reaction system. The continuous generation of $\cdot$OH$_{surface}$ enabled us to explore their versatile applications in extensive domains.

## Applications in wastewater purification

As one of the most important applications of the Fenton reaction, wastewater purification was chosen to evaluate the catalytic performance of the Fe@N-C-800 catalyst. Sulfamethoxazole (SMX), a pharmaceutical residuum commonly detected in urban wastewater, was selected as the model organic pollutant[51]. As predicted in the above characterization, the sample calcined at 800 °C exhibited superior heterogeneous catalytic performance with a low activation energy (Supplementary Figs. 22–24), achieving the SMX removal of nearly 100% within only 6 min. Notably, without the addition of $H_2O_2$, only 11.2% of SMX was removed in 6 min, suggesting the negligible contribution of catalyst adsorption for SMX removal (Fig. 4a). Meanwhile, compared with other $H_2O_2$-based catalytic systems, Fe@N-C-800/$H_2O_2$ showed the overwhelming catalytic activity, the reaction rate ($K_{obs}$) of which was 46.1 times that of the commercial ZVI induced $H_2O_2$ catalytic system (Supplementary Fig. 25). Benefiting from the well-protected carbon layer, the concentration of leached ions for the

Fe@N-C-800 catalyst was only 0.076 mg L$^{-1}$ by ICP-OES analysis, which was far below the EU standards of 2.0 mg L$^{-1}$ and met the basic emission standard for wastewater treatment. It should be noted that the addition of SCN$^-$ ions could hardly affect the removal rate of SMX (Supplementary Fig. 26), which confirmed that the pyridinic-N sites played a major role in $H_2O_2$ activation rather than the exposed Fe species. As for the conventional Fenton system, even when the concentration of Fe$^{II}$ was increased to the upper limit of 2.0 mg L$^{-1}$, an SMX removal of only 48.5% was achieved under the same conditions, indicating the crucial role of $\cdot$OH$_{surface}$ in efficient SMX removal.

In addition, a total organic carbon (TOC) removal test was performed to compare the catalytic activities of the different systems. The Fe@N-C-800/$H_2O_2$ achieved a favorable TOC removal of over 75% (Supplementary Fig. 27a), which was 3.5 times that of the homogeneous Fenton system and 35.9 times that of the commercial ZVI/$H_2O_2$ system. The degradation pathway of SMX was also illustrated in detail to account for the excellent TOC removal in the Fe@N-C-800/$H_2O_2$ system (Supplementary Fig. 28 and Supplementary Table 4). Based on the results of TOC removal and $H_2O_2$ consumption efficiency of various systems (Supplementary Fig. 27b), we applied the theoretical stoichiometry of oxidant consumption to calculate the real $H_2O_2$ utilization efficiency ($C_{10}H_{11}N_3O_3S + 33\ H_2O_2 \rightarrow 10\ CO_2 + 36\ H_2O + 3\ HNO_3 + H_2SO_4$, the calculation details were provided in Supplementary Information), and the results were presented in Fig. 4b. Encouragingly, the $H_2O_2$ utilization efficiency of Fe@N-C-800/$H_2O_2$ reached 84.1%, which was far higher than that of the control groups.

To gain a more intuitive understanding of the catalytic performance in this work, we compared the catalytic properties of Fe@N-C-800 with those of previously reported heterogeneous catalysts toward Fenton reaction, including metal oxides, bimetallic hybrids and N-doped carbon-supported metal composites (Supplementary Tables 5–7). Notably, the Fe@N-C-800 simultaneously exhibited remarkable TOC removal and the highest $K_{obs}$ with an impressive $H_2O_2$ utilization efficiency (Fig. 4c), which has seldom been achieved in previous work. Moreover, as shown in Fig. 4d, we found that this catalyst exhibited excellent generality and could be extended to degrade other types of organic pollutants, including pharmaceuticals, phenolic compounds and organic dyes. According to the results obtained by experimental and DFT analysis, the exceptional performance of Fe@N-C-800 in terms of the high utilization of $H_2O_2$ and efficient elimination of organic pollutants could be attributed to the following critical factors: (1) the nanometer thickness of the 2D porous carbon framework enabled the sufficient mass transfer for both $H_2O_2$ and target contaminants, (2) the uniform distribution of ultrafine Fe nanoparticles with high loading could effectively boost the pyridinic-N sites on carbon surface and offer the abundant active sites for the adsorption and activation of $H_2O_2$, and (3) the continuous generation but not fierce production of $\cdot$OH$_{surface}$ with an increased lifetime could significantly improve the reaction probabilities toward contaminants and decrease the risk of self-quenching side reactions that usually occurred in homogeneous Fenton system (Supplementary Fig. 29 and Supplementary Text 1).

In addition to the catalytic performance, the environmental adaptability of the catalyst is also crucial for environmental remediation[52,53]. As shown in Supplementary Fig. 30, neither common anions (Cl$^-$, SO$_4^{2-}$, NO$_3^-$ and H$_2$PO$_4^-$) nor natural organic matter (humic acid: HA) presented in the wastewater inhibited the degradation of organic pollutants. Meanwhile, the removal rates of contaminants could still reach 98.2% and 95.6% within 6 min in tap water and lake water (Supplementary Fig. 31), respectively, indicating the strong anti-interference ability of the Fe@N-C-800/$H_2O_2$ system in different aquatic environments. Moreover, unlike the previously reported heterogeneous catalysts, Fe@N-C-800 exhibited broad pH adaptability (2–9), especially a neutral or alkalescent condition was favorable for its application (Supplementary Fig. 32). This should be attributed to the

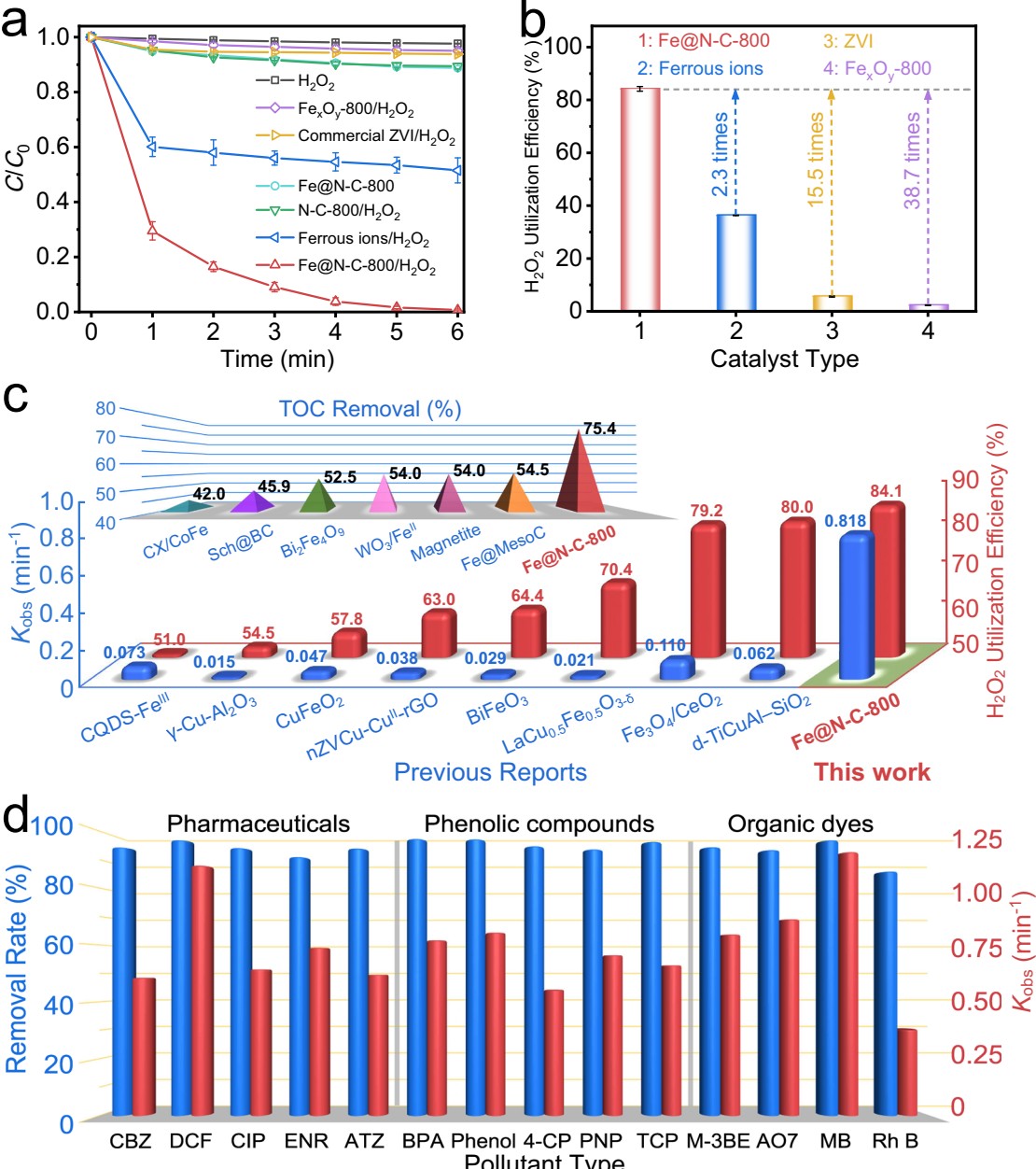

**Fig. 4 | Fenton reaction performance of Fe@N-C-800. a** Time profiles of SMX removal in different systems. **b** $H_2O_2$ utilization efficiency of different catalysts. **c** Comparison of the catalytic performance of Fe@N-C-800 with other recently reported heterogeneous catalysts in terms of $K_{obs}$, $H_2O_2$ utilization efficiency and TOC removal. **d** Elimination efficiency and corresponding $K_{obs}$ for different types of pollutants in the Fe@N-C-800/$H_2O_2$ system in 6 min (condition: [Pharmaceuticals]$_0$ = 10 × 10$^{-6}$ M, [Phenolic compounds]$_0$ = [Organic dyes]$_0$ = 50 × 10$^{-6}$ M, [$H_2O_2$]$_0$ = 1 × 10$^{-3}$ M, [Fe@N-C-800] = 50 mg L$^{-1}$, initial pH = 6.2, $T$ = 25 °C). The error bars in the figures represent the standard deviations.

protonation of pyridinic-N sites in an acidic environment[54], resulting in decreased active sites and weakened activity. In addition, recyclability and stability are essential properties for the heterogeneous catalysts in practical application. As shown in Supplementary Fig. 33, a saturated magnetization of 20 emu g$^{-1}$ was calculated for the catalyst, suggesting that Fe@N-C-800 inherited the magnetism from the metallic Fe phase, which was beneficial for the effective separation of catalysts. Besides, no significant variation was detected in the XRD pattern of Fe@N-C-800 after storage for 2 months (Supplementary Fig. 34a), and the catalyst after storage could still activate $H_2O_2$ to eliminate SMX in 6 min (Supplementary Fig. 34b). Moreover, over 95% SMX removal rate could be still achieved by the used Fe@N-C-800 after six successive degradation tests (Supplementary Fig. 35), suggesting the excellent

long-term activity of Fe@N-C-800. After the reaction, the whole structure and morphology of Fe@N-C-800 were well maintained through the XRD and TEM characterizations (Supplementary Fig. 36), further verifying that the encapsulation strategy could effectively protect the Fe center and improve the stability of the catalyst.

Furthermore, we collected the industrial wastewater from Zhejiang Longsheng Group Co., Ltd. to investigate the potential practical application of the obtained catalyst. It should be pointed out that the specific components in industrial wastewater were complicated, and the initial chemical oxygen demand (COD) value was still as high as 968 mg L$^{-1}$ after biological treatment and could not be discharged directly (Fig. 5a). After treatment for just 0.5 h via the Fe@N-C-800/$H_2O_2$ system, the COD value of wastewater

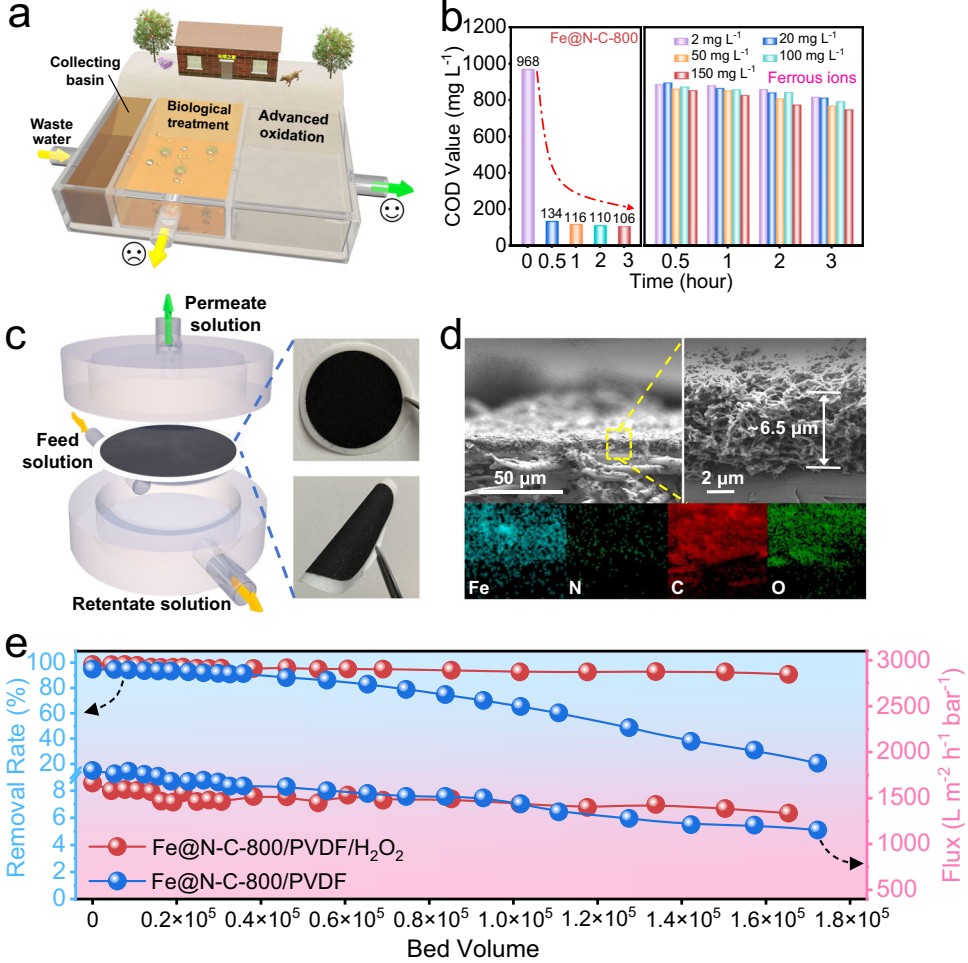

**Fig. 5 | Evaluation of the practical application of Fe@N-C-800 in the Fenton reaction. a** Schematic illustration of treatment technologies for wastewater. **b** COD removal of industrial sewage in different systems (condition: [Fe@N-C-800] = 0.5 g L$^{-1}$, [H$_2$O$_2$]$_0$ = 5 × 10$^{-3}$ M, $T$ = 25 °C). **c** Schematic illustration of the cross-flow filtration system and images of Fe@N-C-800/PVDF membrane. **d** SEM images of the cross-sectional view of Fe@N-C-800/PVDF membrane and its elemental mapping images. **e** SMX removal performance of Fe@N-C-800/PVDF/H$_2$O$_2$ system and Fe@N-C-800/PVDF system as a function of bed volume at a pressure of 0.5 bar (condition: [H$_2$O$_2$]$_0$ = 2 × 10$^{-3}$ M, [SMX]$_0$ = 5 × 10$^{-6}$ M, $T$ = 25 °C, time on stream = 2 h). Panels (**a**) and (**c**) were generated by the Cinema 4D software.

significantly decreased to 134 mg L$^{-1}$ and then maintained at approximately 110 mg L$^{-1}$ in the following reaction (Fig. 5b). In contrast, the conventional Fenton system failed to deal with such intricate sewage even when the concentration of Fe$^{II}$ reached 150 mg L$^{-1}$, reflecting the superb practicability of Fe@N-C-800/H$_2$O$_2$ in treating realistic wastewater. Meanwhile, the Fe@N-C-800/H$_2$O$_2$ system also exhibited the feasibility of treating a large amount of wastewater in the pilot-scale experiment (Supplementary Fig. 37). Besides, we could also deposit Fe@N-C-800 catalysts on the commercially available PVDF membrane and evaluate their continuous pollutant removal performance using a flow-through setup (Fig. 5c and Supplementary Fig. 38). The SEM images in Fig. 5d and Supplementary Fig. 39 showed the porous surface and interconnected face-to-edge stacking of layers in the catalytic membrane with a thickness of 6.5 μm. Benefiting from the abundant percolating channels and equitably distributed Fe elements within the filtration membrane, the flow-through catalytic system maintained the SMX removal of over 90% and the high water flux (-1300 L m$^{-2}$ h$^{-1}$ bar$^{-1}$) even after handling an influent equivalent to 1.65 × 10$^5$ bed volume (Fig. 5e), which was far beyond the performance of previously developed catalytic membranes[55–59], whereas the filtration system without the H$_2$O$_2$ showed a significant decline in SMX removal, mainly resulting from the saturated adsorption of Fe@N-C-800/PVDF membrane. In short, the excellent adaptability, durability as

well as oxidizing capacity exhibited in complex wastewater environments consolidated the application potential of the catalyst synthesized in this work.

### General synthesis of other high-loading M@N-C composites

Inspired by the native catechol groups in DA for strong metal ion anchoring, we further explored the generality of our self-polymerization confinement strategy to synthesize a series of M@N-C composites with different metal precursors, including cobalt (Co), copper (Cu), nickel (Ni) and silver (Ag), using similar procedures with optimal polymerization conditions. As shown in the XRD patterns (Fig. 6a), the characteristic diffraction peaks of different metal crystals could be clearly observed, and the distinguishable lattice fringes displayed in the HRTEM images matched well with the corresponding crystal plane values (insets of Fig. 6b–e), which suggested that the target metal NPs were successfully obtained via our strategy. Meanwhile, the TEM images verified the highly dispersed metallic components in the carbon layers with uniform particle sizes centered at 3–6 nm (Supplementary Fig. 40). ICP-OES analysis confirmed the relatively high metal loadings of Co, Cu, Ni and Ag with values of up to 28.6, 41.0, 39.7 and 20.3 wt%, respectively (Fig. 6f), which were much higher than those of previously reported M@N-C composites[60–63]. Furthermore, this strategy could be readily scaled up by simply replacing manual agitation with a blender or by increasing the volume

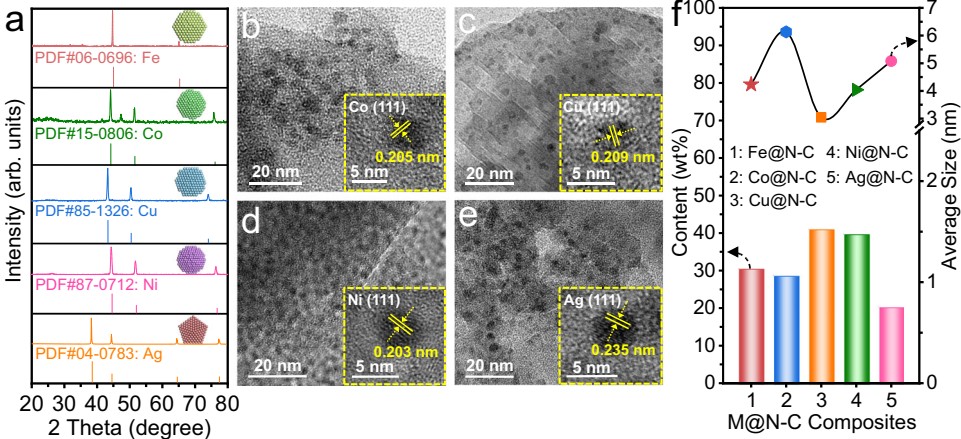

**Fig. 6 | Generality of synthesizing other M@N-C composites. a** XRD patterns of different M@N-C composites. TEM images of **b** Co@N-C, **c** Cu@N-C, **d** Ni@N-C, **e** Ag@N-C, insets are the corresponding HRTEM images. **f** Metal loading and particle size in different M@N-C composites.

of the vessel, thus achieving the gram or even kilogram-scale synthesis of M@N-C composites without any by-products (Supplementary Fig. 41). Therefore, we anticipate that our strategy will serve as a versatile platform to explore the unparalleled properties of high-metal-loading M@N-C catalysts and their widespread applications in sustainable chemical synthesis, energy conversion and environmental remediation.

## Discussion

In summary, a self-polymerization confinement strategy was developed to synthesize a series of ultrafine metal NPs embedded N-doped 2D carbon nanosheets with metal loadings of up to 30 wt%. The entire process was facile, solvent-free, easy to scale up, and suitable for mass production and industrialization. The Fe@N-C-800 catalyst prepared with this strategy demonstrated high catalytic activity ($0.818\ min^{-1}$) and $H_2O_2$ utilization efficiency (84.1%) in the treatment of refractory organic pollutants, far outperforming previously reported heterogeneous catalysts. Systematic investigations on the mechanism revealed that the plentiful Fe NPs boosted pyridinic-N sites on the carbon layer were essential to the exceptional catalytic performance, which resulted in electron transfer from the catalyst to $H_2O_2$ for the continuous generation of $^{\cdot}OH_{surface}$. More profoundly, with their combined flexible 2D structure, remarkable catalytic capability, and strong environmental tolerance, the obtained catalysts could be readily assembled into the porous filtration membrane for the long-term treatment of practical wastewater. These results suggest that our strategy may open up fascinating avenues for the general and efficient production of high-performance M@N-C catalysts for diverse applications.

## Methods

### General synthetic strategy for M@N-C

Firstly, the Fe@N-C was synthesized by a simple grind-assisted pyrolysis method and the specific fabrication procedure was as follows: dopamine hydrochloride (0.47 g) and $FeCl_3 \cdot 6H_2O$ (1.35 g) were mixed with mole ratio of 1:2 by grinding under air ambiance condition with about 50% humidity, and the grinding process continued for about 5 min until the solid powder converted into viscous liquid. Then, the resulting mixture was calcined in a tube furnace for 120 min with a heating rate of $5\ ^{\circ}C\ min^{-1}$ under the argon atmosphere to obtain the Fe@N-C-X catalyst, where X referred to the calcination temperature. For large-scale preparation, the dosages of dopamine hydrochloride (13.28 g) and $FeCl_3 \cdot 6H_2O$ (37.84 g) were accordingly increased. Besides, the synthetic strategy for Fe@N-C-800 with other ratios of dopamine hydrochloride and $FeCl_3 \cdot 6H_2O$ was conducted by the same

procedures except adjusting the ratio of 1:2 into 3:1, 1:1 or 1:3. The other M@N-C samples (including copper, cobalt, nickel and silver) were prepared by using the similar procedures except for different metal chloride salts. Considering the weak moisture adsorption and oxidizing capability of these metal ions compared with that of $FeCl_3 \cdot 6H_2O$, we would intentionally add a small amount of $H_2O_2$ solution (1 mL, 30 wt%) during the grinding process to promote dopamine polymerization. In addition, the $Fe_xO_y$-800 was fabricated by the same procedure without the addition of dopamine hydrochloride, and the N-C-800 was prepared through the calcination of dopamine after polymerization except the addition of $FeCl_3 \cdot 6H_2O$.

### Contaminant degradation

The catalytic experiments were carried out in a 100 mL conical flask containing 20 mL organic pollutant aqueous solution. Unless otherwise specified, the initial concentration of the pharmaceuticals (SMX, CBZ, ATZ, DCF, CIP and ENR) was $10 \times 10^{-6}$ M, while the concentrations of phenolic contaminants (BPA, phenol, 4-CP, TCP, PNP) and organic dyes (AO7, M-3BE, Rh B, MB) were $50 \times 10^{-6}$ M. The initial pH of the typical degradation experiments was uniformly set at a pH = 6.2 to simulate the realistic wastewater, other initial pH was adjusted on request with a diluted aqueous solution of $H_2SO_4$ or NaOH before the addition of catalysts and oxidant. The reaction temperature was maintained at 25 °C or other temperatures on request by a constant temperature shaker water bath. Typically, the catalytic reaction was initiated by adding a certain dosage of catalyst sample and $H_2O_2$ into a 100 mL glass flask containing 20 mL contaminant aqueous solution. At specific time intervals (1 min), 0.5 mL reaction solution was withdrawn and filtered with a 0.45 μm PTFE membrane, followed by high-performance liquid chromatography (HPLC) test or UV-Vis spectrometer test. NaCl, $Na_2SO_4$, $NaNO_3$, $NaH_2PO_4$ and HA were selected to measure the effects of anions and natural organic matter on SMX removal.

## Data availability

Source data underlying Figs. 2d, 3c, e, 4a, b, d, 5b, e and 6f and Supplementary Figs. 17a, b, 22a, b, 23a, b, 25a, 26, 29, 31, 32, 34b and 35 are provided as a Source Data file with this paper. All other relevant data are available from the corresponding author upon reasonable request.

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

## Acknowledgements

We acknowledge the financial support of the National Natural Science Foundation of China (No. 52003240, 22325602, 22176060), Zhejiang Provincial Natural Science Foundation of China (No. LQ21B070007), and the Program of Shanghai Academic/Technology Research Leader (23XD1421000).

## Author contributions

L.X.W. and L.J.R. conceived the idea and designed the experiments. L.X.W., L.J.R., M.X.R., Q.K.S., Z.L.W., W.K.S., Z.W.Z. and H.L. carried out the experiments, and analyzed the experimental data. L.X.W. and L.J.R. performed SEM, TEM, XPS, XRD, etc. H.L. conducted EPR characterizations. L.X.W. and Q.K.S. analyzed theoretical calculations data. L.X.W., L.J.R., M.X.R. and Q.K.S. wrote the paper. All the authors discussed the results and gave rational suggestions.

## Competing interests

The authors declare no competing interests.
