## [Peer Review File · Nature Communications]

A polymer tethering strategy to achieve high metal loading on catalysts for Fenton reactionsREVIEWER COMMENTS

Reviewer #1 (Remarks to the Author):

The topic of study is relevant and the number of experiments performed is impressive. In special, the synthesis procedure and the surface-structure properties of the new material are of interest. However, the large number of figures and characterizations presented is detrimental to giving an in-depth discussion and analysis of the results presented. Moreover, the experimental methodologies used are not sufficiently well described. Considering the literature published in Nature Communications, the manuscript do not meet the standard of the journal and I cannot recommend its publication.

I suggest the following changes before being suitable for publication:

* Overall, the English must be improved;

* The authors are using a molar ratio of $H_2O_2/SMX = 1000$ which is far from the stoichiometric ratio of $H_2O_2/SMX = 33$. The oxidant dosage is a critical aspect of Fenton systems and this condition must be properly justified.

*Homogeneous Fenton experiments are being performed at circumneutral pH which produces precipitation of Fe species. Classic Fenton experiments should be performed at acidic pH. The possibility of operate at circumneutral pH is an advantage of heterogeneous Fenton-like systems, although is it surprisingly outstanding to reach such a high catalytic activity at circumneutral pH and room temperature with Fe species.

*Adsorption experiments (SMX and reaction intermediates, in absence of H_2O_2) must be evaluated and discussed in the body of the manuscript.

*The long term activity of the catalyst must be assessed by performing successive tests with the used catalyst sample.

*Final TOC level is being measured at 6 min of reaction time? This is not clear throughout the work.

*The initial concentration of SMX is very low (10^{-6} mol/L) which is representative of a real pharmaceutical wastewater at WWTP. However, the analytics of this operating condition is challenging. In this sense, the quality of the water matrix used and details of the HPLC detector employed must be specified in the experimental section. In addition, the detection limit of the analytical techniques employed must be indicated.

*From Fig. 5, time on stream for the continuous oxidation experiment is not clear.

Reviewer #2 (Remarks to the Author):

The authors present a facile and general method to prepare metal loaded N-doped carbon materials with high metal loading and good dispersion. Importantly, the Fe loaded material shows excellent Fenton catalytic activity for the removal of various organic pollutants, exhibiting great potential in practical applications. I recommend the manuscript for publication after solving the following issues.

1. The material was synthesized under air ambience condition with about 50% humidity. It seems that humid condition is necessary and the metal precursor should be easy to adsorb moisture. If the metal precursor does not adsorb moisture, should some solvents like water be added?
2. The authors show that $[Fe(acac)_3]$ is not a good Fe precursor to prepare Fe@NC using this method, how about $Fe(NO_3)_3$? The meal precursors for the preparation of other M@NC materials (Co, Cu, Ni

and Ag) should be indicated.

3. The authors mentioned that Fe are coordinated to pyridine N in the final catalyst, why not the carbon precursor with rich pyridine N is directly selected as precursor instead of DA? Why should Fe be first coordinated by O in catechol groups and how they transfer to Fe-N coordination?

4. Considering the iron loading is 30.5 wt% while the Fe content on catalyst surface is as low as 2.3 wt%, the authors conclude that the Fe NPs are covered by carbon layers. Does the carbon layer affect the ability of Fe to activate H₂O₂?

5. "the relative intensity ratio of the D and G bands (ID/IG) of the carbon matrix first increased as the temperature increased from 600 to 800 °C and subsequently decreased at pyrolysis temperatures above 900 °C". The G band presents the degree of graphitization and the D band represents the defect, higher ID/IG indicates lower degree of the graphitization.

6. "The pyridinic-N cooperating with the Fe site on Fe@N-C was essential for H₂O₂ activation." α -Fe markedly decreased (from 64% to 51.3%), and pyridinic-N content also showed a marked decrease after reaction (from 40.3% to 34.9%). How about the cycling performance of the catalyst? The cycling tests of several consecutive runs should be performed.

7. For the comparison of hydroxyl radicals produced by Fe@NC and conventional Fenton system (Fig. 3a,b), is the content of Fe same or not? Why the Fe center in Fe@NC can continuously activate H₂O₂?

8. For the degradation of different kinds of organic pollutants, why the concentrations of pharmaceuticals and that of phenolic compounds and dyes are different?

9. The symbol "MW" and Supplementary Figure 2 should be explained.

10. How about the adsorption of organic pollutants on the catalyst considering its large surface area? All removal curves start from C/C₀=1, is the removal the contribution of both adsorption and degradation?

Reviewer #3 (Remarks to the Author):

In this manuscript, Wang et al. reported a polymerization confinement strategy for synthesizing a carbon nanosheet-supported Fe nanoparticle catalyst (Fe@N-C) with impressive catalytic performance toward the Fenton reaction. The subject—designing high-performance heterogeneous catalysts for Fenton reaction and understanding the activity origin is of high interest and importance, and has still been widely investigated due to its application prospect in various fields. Despite the synthesis is relatively facile and holds the potential for large-scale production of Fe@N-C and the catalyst showed the good catalytic activity and H₂O₂ utilization efficiency, the reviewer cannot see sufficient new insights or breakthroughs revealed by the manuscript to advance the field.

The authors presented a long discussion about the high metal loading and ultrafine size of the Fe NPs to highlight the advantage of the self-polymerization confinement strategy. However, similar M–N–C catalysts synthesized through the grinding-polymerization-coordination-pyrolysis process have already been reported considerably and high-density cluster or single-atom catalysts can also be obtained through this method. The actual activity centers are the pyridinic-N moieties and the Fe NPs function as the electron donors, so the emphasis on the high-loading and small-size characteristics and the structural evolution process of Fe@N-C somewhat sidestep the discussion about what really counts, which has been elaborated clearly in a considerable amount of work. In addition, the interaction between Fe NPs and adjacent N moieties has not been fully elaborated: in addition to electron transfer, can the Fe NPs improve the number and stability of active N sites? The synergistic interaction between Fe NPs and adjacent N sites should be investigated and interpreted more.

Also, several previous works have reported that N-coordinated metal catalysts exhibit high catalytic reactivity toward Fenton or Fenton-like reactions (Fe: ACS Nano 2018, 12, 9441; Adv. Mater. 2022, 34, 2110653; Co: J. Am. Chem. Soc. 2018, 140, 12469; Cu: Angew. Chem. Int. Ed. 2022, 61, e202207268; Mn: J. Am. Chem. Soc. 2019, 141, 12005). Are there any advantages to utilizing Fe NPs as the regulator of pyridinic-N? Do other M@N-C (M = Co, Cu, Ni, Ag) analogues synthesized through the same method have similar Fenton activity? I understand this might be a lot of additional work, but could you be more specific about the role Fe NPs play?

Based on these facts and given the high standard for the publication in Nat. Commun., the reviewer

does not believe the manuscript contains enough innovative, significant content to be considered for publication in this journal.

The following comments are provided for improving the quality of the manuscript.

1. The Fe content detected by XPS is 2.3 wt%, much lower than that of the ICP-OES results. It seems that the Fe NPs are encapsulated in thick carbon layers. The carbon layer structure should be characterized in detail.
2. The stability of the catalyst is of great importance. Therefore, the structure, surface composition, and morphology of the catalyst after the reaction should be provided.
3. The removal performance of contaminants of Fe@N-C-600 and Fe@N-C-700 is very close to that of Fe@N-C-800 despite their high iron leaching. Can the catalysts maintain high removal performance after the leaching of water-soluble iron species?
4. State-of-the-art M–N–C catalysts for Fenton reactions should be added in Supplementary Table 5 and 6 for comparison.
5. The authors should correct some mistakes in the manuscript, such as the ordinate of Supplementary Figure 7a.

Reviewer #4 (Remarks to the Author):

In this paper, the authors reported a facile and efficient strategy to synthesise a series of ultrafine metal NPs embedded N-doped 2D carbon nanosheets with high metal loadings. The Fe@N-C-800 nanocatalyst synthesised via this strategy showed outstanding performance for H₂O₂ activation. The authors gave a solid proof to demonstrate the mechanism for the exceptional catalytic performance. After carefully scrutiny of this work, the reviewer thinks this work demonstrates sufficient novelty and high quality in the interpretation of the outcomes. Thus, it is recommended the publication of this work after minor revision.

1. The authors claimed that the essence of self-polymerisation confinement strategy were chelation and polymerization of dopamine. Catalysts with the size of metal particles much larger than Fe@N-C-800 are obtained by blocking these two effects, respectively. Please compare their catalytic performance.
2. Considering the important role of PDA in synthesising the high-performance catalyst, can other catecholic derivatives to be used as precursors to synthesise the efficient catalyst?
3. The authors compared the performance of different catalysts in Figure 4a. In the figure, N-C-800 is the product of dopamine calcined at 800 °C, which the reviewer thinks is inaccurate. It should be more reasonable to use polydopamine as the precursor to synthesise N-C-800.
4. The effect of water matrix should be included in practical application. At least tap water and surface water should be employed for that purpose.
5. It is exciting that the catalyst reported in this work could achieve the both high catalytic activity and H₂O₂ utilisation efficiency, the reasons should be explained.
6. The authors claimed the continuous generation of •OH in the Fe@N-C-800/H₂O₂ system, so more evidence should be supplemented to confirm the durability of the synthesised catalyst.

Reviewer #5 (Remarks to the Author):

1. How to prove that the Fe NPs are encapsulated in carbon layers not supported on the surface? In some previous studies, they think Fe NPs are the active sites, while the electrons will be provided by carbon and transferred via the N atoms. Can you exclude this possibility?
2. Moreover, the author claimed that Fe@N-C have porous structure (L355), the characterization for the porous structure should be performed. This porous structure should expose some Fe sites, which will also contact with H₂O₂ and act as active sites.
3. The author demonstrated that the high activity of Fe@N-C stem from pyridinic-N coordinated Fe

sites, which can induce electron transfer from catalyst to H₂O₂. This coordinated structure is usually in the atomic catalyst, where N and Fe has a special ratio. While in this manuscript, the Fe is in the form of nanoparticles, the ratio of Fe is much larger than N, how did the pyridinic-N coordinate with the Fe atoms? I believe that most of the Fe atoms are not coordinated with N. How about the percentage of N? Are there enough pyridinic-N to coordinated with Fe? How about the evolution of these N species after Fe leaching?

4. What's the exact active sites for this Fe@N-C material? In fact, the activation of H₂O₂ should be an electron consuming process. So what are these electrons come from? Since lots of electrons are needed, the catalysts after the long-time testing should be totally different compared with the pristine one. The materials after reaction should be carefully analyzed, including the Fe, the N and carbon species.

5. Can the relationship between leached Fe and consumed H₂O₂ be analyzed? This will be helpful for the analysis of the active sites.

Respective answer to the detailed comments:

Reviewer: 1

The topic of study is relevant and the number of experiments performed is impressive. In special, the synthesis procedure and the surface-structure properties of the new material are of interest. However, the large number of figures and characterizations presented is detrimental to giving an in-depth discussion and analysis of the results presented. Moreover, the experimental methodologies used are not sufficiently well described. Considering the literature published in Nature Communications, the manuscript does not meet the standard of the journal and I cannot recommend its publication.

I suggest the following changes before being suitable for publication:

Comment 1: Overall, the English must be improved.

Answer: Thank you very much for this helpful comment, and we have asked a professional language polishing agency to help us correct our English writing (English Editing Certificate as shown in the following Figure).

We have carefully checked the whole manuscript and the language has been improved in the revised manuscript. Some examples are as follows:

Page 1: We have revised “such methods remain substantially challenging to develop” to “such practices remain substantially challenging to develop”.

Page 2: We have revised “Hydroxyl radicals ($\cdot\text{OH}$), which has supreme oxidation potential and non-selective reaction capability” to “Hydroxyl radicals ($\cdot\text{OH}$), which own supreme oxidation potential and non-selective reaction capability”.

Page 3: We have revised “The typical top-down route is as follows: metal-containing molecules are spatially confining into the structural or electronic defects of existing carbon materials (including graphene sheets, carbon nanotubes and biochar), following which they are subjected to chemical reduction to obtain the target catalysts” to “Generally, for top-down route, metal-containing molecules are spatially confined into the structural or electronic defects of existing carbon materials such as graphene sheets, carbon nanotubes and biochar, followed by the chemical reduction process to obtain the target catalysts”.

Page 5: We have revised “This study opens new avenues for the efficient design of high-performance catalysts for Fenton reaction and other related chemical processes” to “This study opens new avenues for efficiently designing high-performance catalysts for Fenton reaction and other related chemical processes”.

Page 6: We have revised “Subsequently, the formed PDA assembled into layered bulk structures to provide a stable coordinate environment for large loadings of isolated metal atoms” to “Subsequently, the self-assembled layered structure of PDA provided a stable coordinate environment for large loadings of isolated metal atoms”.

Page 8: We have revised “but also provides abundant accessible active sites while minimizing the mass/charge transfer distance” to “but also provides abundant accessible active sites to minimize the mass/charge transfer distance”.

Page 10: We have revised “In Raman spectroscopy, the relative intensity ratio of the D and G bands (I_D/I_G) of the carbon matrix first increased as the temperature

increased from 600 to 800 °C and subsequently decreased at pyrolysis temperatures above 900 °C” to “In Raman spectroscopy, the relative intensity ratio of the D and G bands (I_D/I_G) of the carbon matrix was also evaluated, where the I_D/I_G first increased as the temperature increased from 600 to 800 °C and subsequently decreased at pyrolysis temperatures above 900 °C”.

Page 20: We have revised “The reason for this phenomenon was that the protonation of pyridinic-N sites on the carbon layers in the presence of superfluous H^+ accelerated the transformation of pyridinic-N to pyridinic-N-H (pyridinium) in an acidic environment, resulting in decreased active sites and weakened activity” to “This should be attributed to the protonation of pyridinic-N sites on the carbon layers in an acidic environment, resulting in decreased active sites and weakened activity”.

Page 21: We have revised “the COD value of wastewater significantly decreased to 134 mg L^{-1} and was subsequently maintained at approximately 110 mg L^{-1} ” to “the COD value of wastewater significantly decreased to 134 mg L^{-1} and then maintained at approximately 110 mg L^{-1} in the following reaction”.

Other revised sentences have been also marked in red in the manuscript. Thanks again!

Comment 2: The authors are using a molar ratio of $H_2O_2/SMX = 1000$ which is far from the stoichiometric ratio of $H_2O_2/SMX = 33$. The oxidant dosage is a critical aspect of Fenton systems and this condition must be properly justified.

Answer: Thanks a lot for reviewer’s valuable comments. In our work, the added concentration of H_2O_2 and SMX were $1 \times 10^{-3} \text{ M}$ and $10 \times 10^{-6} \text{ M}$, respectively, so the molar ratio of $H_2O_2/SMX = 100$ located on the same magnitude order of the stoichiometric ratio ($H_2O_2/SMX = 33$). Generally, less than 30% of H_2O_2 was effectively utilized in the conventional Fenton system, thus the practical added oxidant dosage was higher than that of the stoichiometric amount¹. To achieve complete removal of contaminant and have an intuitionistic comparison with the traditional Fenton system, a reasonable molar ratio of $H_2O_2/SMX = 100$ was selected. Thanks again!

References

1. Xing, M. et al. Metal sulfides as excellent co-catalysts for H₂O₂ decomposition in advanced oxidation processes. *Chem* **4**, 1359–1372 (2018).

Comment 3 Homogeneous Fenton experiments are being performed at circumneutral pH which produces precipitation of Fe species. Classic Fenton experiments should be performed at acidic pH. The possibility of operate at circumneutral pH is an advantage of heterogeneous Fenton-like systems, although is it surprisingly outstanding to reach such a high catalytic activity at circumneutral pH and room temperature with Fe species.

Answer: Thanks for the reasonable comment. According to your suggestion, the SMX degradation performance of classic homogenous Fenton system (Ferrous ions/H₂O₂) in the acidic condition was evaluated. As shown in **Fig. R1**, improved SMX removal rate (75.4%) was achieved by the Ferrous ions/H₂O₂ system at an initial pH = 3.0, which

was superior than that of the same reaction tested at a circumneutral pH = 6.2 (48.5%). Generally, the pH of the real wastewater containing pharmaceuticals such as antibiotics ranges at a circumneutral level¹. In this case, the acidification pretreatment of the real wastewater is an essential process before employing classic Fenton system for practical application, which increases the operation costs and the risk of equipment corrosion². To meet the needs of practical application, the initial pH of the typical degradation experiments was uniformly set at a circumneutral pH = 6.2 in this manuscript. According to your valuable suggestion, the reason for initial pH selection has been also supplemented in the revised manuscript. Thanks again!

Fig. R1 Time profiles of SMX removal in Fe@N-C-800/H₂O₂ (initial pH = 6.2) and

ferrous ions/H₂O₂ (initial pH = 3.0 and initial pH = 6.2) systems. Condition: [SMX]₀ = 10 × 10⁻⁶ M, [H₂O₂]₀ = 1 × 10⁻³ M, [Fe@N-C-800] = 50 mg L⁻¹, [Ferrous ions] = 2 mg L⁻¹, T = 25 °C.

References

1. Zainab, SM., Junaid, M., Xu, N., Malik, RN. Antibiotics and antibiotic resistant genes (ARGs) in groundwater: A global review on dissemination, sources, interactions, environmental and human health risks. *Water Res.* **187**, 116455 (2020).
2. Yan, Q. et al. Constructing an acidic microenvironment by MoS₂ in heterogeneous Fenton reaction for pollutant control. *Angew. Chem. Int. Ed.* **60**, 17155–17163 (2021).

Comment 4 Adsorption experiments (SMX and reaction intermediates, in absence of H₂O₂) must be evaluated and discussed in the body of the manuscript.

Answer: Thanks for your valuable comment. According to your suggestion, the adsorption experiment in the absence of H₂O₂ have been carried out. SMX and its two representative degradation intermediates (sulfanilic acid (CAS: 121-57-3) and 3-amino-5-methylisoxazole (CAS: 1072-67-9)) detected in the LC-MS test, were selected as the target pollutants to evaluate the adsorption capability of Fe@N-C-800. As shown in **Fig. R2**, the Fe@N-C-800 exhibited negligible adsorption performance, only 11.2%, 11.0% and 11.6% adsorption rate for SMX, sulfanilic acid and 3-amino-5-methylisoxazole could be reached within 6 min, respectively. These results suggested that the removal of SMX and its reaction intermediates in Fe@N-C-800/ H₂O₂ system was mainly caused by •OH-induced degradation rather than adsorption. According to your suggestion, the results of adsorption experiments have been also discussed in the body of the revised manuscript. Thanks again!

Fig. R2 The adsorption performance of Fe@N-C-800 for SMX and its intermediate product. Condition: $[\text{Fe@N-C-800}] = 50 \text{ mg L}^{-1}$, $[\text{Contaminant}]_0 = 10 \times 10^{-6} \text{ M}$, $T = 25 \text{ }^\circ\text{C}$.

Comment 5 The long term activity of the catalyst must be assessed by performing successive tests with the used catalyst sample.

Answer: According to your helpful suggestions, the successive tests of the used Fe@N-C-800 samples were carried out to assess the long-term activity of Fe@N-C-800. As shown in **Fig. R3**, over 95% SMX removal rate could be still achieved by the used Fe@N-C-800 after six successive degradation tests, suggesting the excellent long-term activity of Fe@N-C-800 in this work. The relevant discussion has been also supplemented in the revised manuscript. Thanks again!

Fig. R3 Cyclic degradation of SMX by the recycled Fe@N-C-800. Condition: $[\text{SMX}]_0 = 10 \times 10^{-6} \text{ M}$, $[\text{H}_2\text{O}_2]_0 = 1 \times 10^{-3} \text{ M}$, $T = 25 \text{ }^\circ\text{C}$.

Comment 6 Final TOC level is being measured at 6 min of reaction time? This is not clear throughout the work.

Answer: We sincerely appreciate reviewer's reasonable comments. To have a more accurate and intuitionistic understanding on the TOC removal performance of the Fe-NC-800/H₂O₂ system, the dosages of target pollutant and H₂O₂ in the routine degradation evaluation experiment were improved to five-fold (50×10^{-6} M) and two-fold (2×10^{-3} M) during the TOC test, respectively. Therefore, the final TOC removal test was prolonged to 30 min to achieve the complete mineralization of SMX¹. Based on the TOC result, the obtained real H₂O₂ utilization efficiency (84.1%) in the Fe@N-C-800/H₂O₂ system was calculated by the following Equation R1²:

$$\eta = [\Delta\text{H}_2\text{O}_2]_s / [\Delta\text{H}_2\text{O}_2]_c \quad (\text{R1})$$

where $[\Delta\text{H}_2\text{O}_2]_s$ referred to the stoichiometric amount of H₂O₂ for SMX complete mineralization (1.184×10^{-3} M), $[\Delta\text{H}_2\text{O}_2]_c$ was the total real consumed amount of H₂O₂ during the reaction (1.408×10^{-3} M). The details about TOC experiment have been also supplemented in the Supporting Information. Thanks again!

References

1. Zhan, S., Zhang, H., Mi, X., Zhao, Y., Hu, C., Lyu, L. Efficient Fenton-like process for pollutant removal in electron-rich/poor reaction sites induced by surface oxygen vacancy over cobalt–zinc oxides. *Environ. Sci. Technol.* **54**, 8333–8343 (2020).
2. Luo, W., Zhu, L., Wang, N., Tang, H., Cao, M., She, Y. Efficient removal of organic pollutants with magnetic nanoscaled BiFeO₃ as a reusable heterogeneous Fenton-like catalyst. *Environ. Sci. Technol.* **44**, 1786–1791 (2010).

Comment 7 The initial concentration of SMX is very low (10^{-6} mol/L) which is representative of a real pharmaceutical wastewater at WWTP. However, the analytics of this operating condition is challenging. In this sense, the quality of the water matrix used and details of the HPLC detector employed must be specified in the experimental

section. In addition, the detection limit of the analytical techniques employed must be indicated.

Answer: Thank you very much for the comment. The water matrix used in this study was the ultrapure water purchased from *Wahaha Group* of China, whose all indicators met the standard of analytical experimental water¹, thus the fine quality of the employed water matrix could be guaranteed. The characteristics of the water matrix was listed in **Table R1**. In order to make the experimental data accurate and reliable, the employed HPLC detector in this study was Waters 2998 Photodiode Array, which offered advanced optical detection and trace quantitation of sample in conjunction with spectral analysis capability². Detailed operation details were listed in the **Table R2**, which could guarantee the detection limit of SMX as low as 1×10^{-7} M. According to your suggestion, the corresponding information have been also supplemented in the Supporting Information in detail. Thanks again!

Table R1 The characteristics of ultrapure water used in the experiments.

Item	Standard requirement
Chromaticity / (°)	<5
Turbidity / (NTU)	<1
Smell and taste	no
pH	6.0~7.0
Conductivity / ($\mu\text{S}/\text{cm}$)	<5

Table R2 Operating details of HPLC

Item	Parameter
Flow rate / (mL min^{-1})	2
Mobile phase	Water : Acetonitrile = 70 : 30
Wavelength / (nm)	190 ~ 800
Chromatographic column	HPLC ® BEH C18 1.7 μm , 2.1 \times 100 mm Column
Temperature / (°C)	25
Sample volume / (μL)	20
Pressure / psi	~ 2200

References

1. Liu, J-M., Chen, J-T., Yan, X-P. Near infrared fluorescent trypsin stabilized gold nanoclusters as surface plasmon enhanced energy transfer biosensor and in vivo cancer imaging bioprobe. *Anal. Chem.* **85**, 3238–3245 (2013).
2. Shao, B., Dong, H., Sun, B., Guan, X. Role of ferrate (IV) and ferrate (V) in activating ferrate (VI) by calcium sulfite for enhanced oxidation of organic contaminants. *Environ. Sci. Technol.* **53**, 894–902 (2018).

Comment 8 From Fig. 5, time on stream for the continuous oxidation experiment is not clear.

Answer: Thanks for your valuable suggestion. In the process of the continuous oxidation experiment, the concept of “bed volume” was introduced to reflect the processing capacity of the catalytic membrane in treating wastewater¹, which represented the ratio of total volume of the solution flowing through the Fe@N-C-800/PVDF membrane to the volume of the catalytic membrane itself at every fixed time. In this work, the flow-through catalytic system maintained the effective SMX removal even after handling an influent equivalent to 1.65×10^5 bed volume, which was far beyond the performance of previously developed catalytic membranes. According to your advice, we converted the “bed volume” into “time” in **Fig. R4** for better understanding. As could be seen, the removal rate of SMX in the Fe-NC-800/PVDF/H₂O₂ system still maintained a high level (over 90 %) for 2 hours at a relatively high flux up to $1300 \text{ L m}^{-2} \text{ h}^{-1} \text{ bar}^{-1}$. According to your suggestion, the operation time in the continuous oxidation experiment has been added in the revised manuscript. Thank you again!

Figure R4 SMX removal performance using Fe@N-C-800/PVDF/H₂O₂ system and Fe@N-C-800/PVDF system as a function of time at a pressure of 0.5 bar. Condition: [H₂O₂]₀ = 2 × 10⁻³ M, [SMX]₀ = 5 × 10⁻⁶ M, T = 25 °C.

References

- Chen, Y., Zhang, G., Liu, H., Qu, J. Confining free radicals in close vicinity to contaminants enables ultrafast fenton-like processes in the interspacing of MoS₂ membranes. *Angew. Chem. Int. Ed.* **58**, 8134–8138 (2019).

Reviewer: 2

The authors present a facile and general method to prepare metal loaded N-doped carbon materials with high metal loading and good dispersion. Importantly, the Fe loaded material shows excellent Fenton catalytic activity for the removal of various organic pollutants, exhibiting great potential in practical applications. I recommend the manuscript for publication after solving the following issues.

Comment 1 The material was synthesized under air ambience condition with about 50% humidity. It seems that humid condition is necessary and the metal precursor should be easy to adsorb moisture. If the metal precursor does not adsorb moisture, should some solvents like water be added?

Answer: Thanks for your valuable suggestion. The humid condition is indeed crucial for the synthesis of Fe@N-C-800 catalyst, which not only provides the moist condition for obtaining polydopamine (PDA), but also guarantees the sufficient tethering effect

of metal ions by PDA. For other metal precursors with poor or no moisture adsorption capability such as iron (III) nitrate nonahydrate and iron (III) acetylacetonate, a little amount of water should be added to construct a necessary humid environment. In addition, the related synthesis details have been also supplemented in the revised manuscript. Thanks again!

Comment 2 The authors show that $[\text{Fe}(\text{acac})_3]$ is not a good Fe precursor to prepare Fe@NC using this method, how about $\text{Fe}(\text{NO}_3)_3$? The metal precursors for the preparation of other M@NC materials (Co, Cu, Ni and Ag) should be indicated.

Answer: Thanks for your valuable suggestion. As we mentioned in the results and discussion, one of the key points of the self-polymerization confinement strategy was the chelating effect between metal precursor and DA. Since the strong interaction between iron (III) and acetylacetonate excluded the chelation of DA and iron (III), $[\text{Fe}(\text{acac})_3]$ was not a suitable Fe precursor to prepare Fe@N-C using this method. However, for $\text{Fe}(\text{NO}_3)_3$, the interaction between iron (III) and nitrate was moderate, which would not affect the process of the chelating reaction. To identify the universality of self-polymerization confinement strategy, $\text{Fe}(\text{NO}_3)_3 \cdot 9\text{H}_2\text{O}$ was chosen as the metal precursors for Fe@N-C synthesis. In view of the poor moisture adsorption of $\text{Fe}(\text{NO}_3)_3 \cdot 9\text{H}_2\text{O}$, an appropriate amount of H_2O was added during grinding process to guarantee the impregnation of iron into the DA and induce the rapid polymerization process. As shown in **Fig. R5**, the synthesized catalyst displayed similar XRD pattern compared with that of FeCl_3 -prepared samples. Moreover, $\text{Fe}(\text{NO}_3)_3$ -prepared samples could also effectively activate H_2O_2 to degrade SMX, where nearly 97 % of SMX removal was achieved within 6 min. Considering the relatively simple operating procedure and superior catalytic performance, the $\text{FeCl}_3 \cdot 6\text{H}_2\text{O}$ was ingeniously selected in this work to prepare the efficient Fenton catalyst.

In addition, chlorine salts were used as the metal precursor for the synthesis of other M@NC materials (Co, Cu, Ni and Ag) in this work. According to your suggestion, the relevant information has been added in the revised manuscript. Thanks again!

Fig. R5 **a** XRD patterns of the Fe@N-C catalysts prepared by $\text{FeCl}_3 \cdot 6\text{H}_2\text{O}$ and $\text{Fe}(\text{NO}_3)_3 \cdot 9\text{H}_2\text{O}$. **b** Plots of SMX concentration versus time of the Fe@N-C-800 catalysts prepared by $\text{FeCl}_3 \cdot 6\text{H}_2\text{O}$ and $\text{Fe}(\text{NO}_3)_3 \cdot 9\text{H}_2\text{O}$. Condition: [Catalyst] = 50 mg L^{-1} , [SMX]₀ = 10×10^{-6} M, [H_2O_2]₀ = 1×10^{-3} M, T = 25 °C.

Comment 3 The authors mentioned that Fe are coordinated to pyridine N in the final catalyst, why not the carbon precursor with rich pyridine N is directly selected as precursor instead of DA? Why should Fe be first coordinated by O in catechol groups and how they transfer to Fe-N coordination?

Answer: Thanks for your valuable suggestion. The essence of polymer tethering strategy lies in that the carbon precursor can anchor large numbers of metal ions and then undergo a polymerization process to provide a stable coordinate environment, thus suppressing metal aggregation and guaranteeing the in-situ formation of well-dispersed metal NPs under carbothermal reduction. To this end, dopamine (DA), a well-known mussel-inspired material featuring abundant chelating groups, self-polymerization capability, and tunable assembled structure¹, was intentionally selected as the carbon and nitrogen precursor for the subsequent thermolysis treatment. Through systematic experiments and density functional theory (DFT) analyses, the authors finally revealed that pyridinic-N coordinated Fe was the most optimal site, which had the highest affinity to H_2O_2 for the generation of $\cdot\text{OH}$. The above results provided a positive inspiration and a solid foundation for our subsequent research. Just as the reviewer speculated, the precursor containing higher content of pyridinic-N should have the great

potential for the preparation of efficient Fenton catalyst, and the related work was in orderly progress. From the data obtained so far, it was found that the Fe@N-C catalyst prepared using the precursor containing higher content of pyridinic-N exhibited more superior performance in degrading SMX. Hence, the ingenious selection of precursor in the following work was based on the comprehensive experiment in this manuscript.

Besides, the existence of abundant functional groups (such as catechol groups, amine groups) in DA is expected to offer a large number of active sites for binding metal ions². Thereinto, the coordination bond between catechol and Fe^{III} is one of the highest stability constants of metal-ligand complexes ($\log\beta_3 \approx 40$, where β_3 is the cumulative stability constant of tri-catechol-Fe^{III})³. Therefore, the Fe^{III} would first coordinate with the catechol groups in DA. During the pyrolysis process, the PDA-Fe^{III} complexes began to decompose and released metal ions at about 500 °C. When the temperature was further elevated, the active CN_x species would instantly capture the Fe ions by forming Fe-N coordination since it was an energy-favorable process. These processes were similar to the recently reported single-atom catalyst, where the metal ions were first chelated by glucose and then formed the Fe-N bonds at the calcination temperature of 800 °C⁴. Thanks for this constructive comment !

References

1. Harrington, MJ., Masic, A., Holten-Andersen, N., Waite. JH., Fratzl, P. Iron-clad fibers: a metal-based biological strategy for hard flexible coatings. *Science* **328**, 216–220 (2010).
2. Liu, Y., Ai, K., Lu, L. Polydopamine and its derivative materials: synthesis and promising applications in energy, environmental, and biomedical fields. *Chem. Rev.* **114**, 5057-5115 (2014).
3. Harris, WR., Weitzl, FL., Raymond, KN. Synthesis and evaluation of an enterobactin model compound. 1, 3, 5-Tris-(2, 3-dihydroxybenzoylaminomethyl) benzene and its iron (III) complex. *J. Chem. Soc., Chem. Commun.* **4**, 177–178 (1979).

4. Zhao, L. et al. Cascade anchoring strategy for general mass production of high-loading single-atomic metal-nitrogen catalysts. *Nat. Commun.* **10**, 1278 (2019).

Comment 4 Considering the iron loading is 30.5 wt% while the Fe content on catalyst surface is as low as 2.3 wt%, the authors conclude that the Fe NPs are covered by carbon layers. Does the carbon layer affect the ability of Fe to activate H₂O₂?

Answer: Thanks for the comment. The encapsulation of Fe NPs by the carbon layer was confirmed by the TEM images as well as the remarkable difference of Fe content detected by ICP-OES and XPS results. In this case, the embedded Fe NPs could donate electrons to the N-doped carbon layer and significantly modulate the electronic structure on the carbon surface, favoring the adsorption of H₂O₂ onto the pyridinic-N site and lowering the energy barrier for H₂O₂ activation. These results were confirmed by the density functional theory and experimental analyses, which suggested that the reaction was occurred on the N-doped carbon layer while the Fe NPs could significantly promote this process through electron transfer. Thanks again!

Comment 5: “the relative intensity ratio of the D and G bands (I_D/I_G) of the carbon matrix first increased as the temperature increased from 600 to 800 °C and subsequently decreased at pyrolysis temperatures above 900 °C”. The G band presents the degree of graphitization and the D band represents the defect, higher I_D/I_G indicates lower degree of the graphitization.

Answer: We sincerely appreciate reviewer’s comments. Generally, the D and G band represents the defects or disorder arrangement of carbon atoms and the E_{2g} mode vibration of sp²-hybridized carbon, respectively¹. The intensity ratio of D and G band (I_D/I_G) is an important criterion for characterizing the atomic ordered degree of carbonaceous materials. According to the three-stages model proposed by Ferrari and Robertson², the I_D/I_G values would increase when tiny nanocrystalline graphite presented in amorphous carbon materials. However, a reverse trend could be achieved as the nanocrystalline graphite further turned into perfect graphite, resulting in the contraction of carbon framework by transferring the cluster of sp² phase into ordered

rings. In our case, the carbon matrixes were mainly amorphous before the pyrolysis temperature reached 800 °C, thus the increased I_D/I_G values as the temperature elevated from 600 to 800 °C suggested the improvement of relative graphitization degree. As the temperature surpassed 900 °C, most nanocrystalline graphite transferred into perfect graphite, the decreased I_D/I_G value indeed indicated the improved graphitization degree. However, the decreased I_D/I_G value also suggested the consolidation of sp^2 bond in the graphite, resulting in the reduced surface area of Fe@N-C-X, which was confirmed by the N_2 adsorption-desorption results in this work. Thanks again!

References

1. Ferrari, AC. et al. Raman spectrum of graphene and graphene layers. *Phys. Rev. Lett.* **97**, 187401 (2006).
2. Ferrari, AC., Robertson, J. Interpretation of Raman spectra of disordered and amorphous carbon. *Phys. Rev. B* **61**, 14095 (2000).

Comment 6 “The pyridinic-N cooperating with the Fe site on Fe@N-C was essential for H_2O_2 activation.” α -Fe markedly decreased (from 64% to 51.3%), and pyridinic-N content also showed a marked decrease after reaction (from 40.3% to 34.9%). How about the cycling performance of the catalyst? The cycling tests of several consecutive runs should be performed.

Answer: According to your helpful suggestions, the successive tests were carried out to evaluate the cycling performance of Fe@N-C-800. As shown in **Fig. R6**, over 95% SMX removal rate could be still achieved by the used Fe@N-C-800 after six consecutive degradation tests, suggesting the excellent cycling ability of Fe@N-C-800. The relevant discussion has been also supplemented in the revised manuscript. Thanks again!

Fig. R6 Cyclic degradation of SMX by the recycled Fe@N-C-800. Condition: $[\text{SMX}]_0 = 10 \times 10^{-6} \text{ M}$, $[\text{H}_2\text{O}_2]_0 = 1 \times 10^{-3} \text{ M}$, $T = 25 \text{ }^\circ\text{C}$.

Comment 7 For the comparison of hydroxyl radicals produced by Fe@NC and conventional Fenton system (Fig. 3a,b), is the content of Fe same or not? Why the Fe center in Fe@NC can continuously activate H_2O_2 ?

Answer: Thanks for the comment. The concentration of ferrous ions in the EPR spectra was set to be 2 mg L^{-1} , which was comparable with the Fe content on the catalyst surface though the XPS test. It should be noted that even the Fe concentration was improved to 15 mg L^{-1} in conventional Fenton system (similar to the total Fe content in Fe@N-C-800 through the ICP-OES result), the variation trend of EPR signals still had a stark contrast to those in Fe@N-C-800 system, in which the intensity of the DMPO- HO^\bullet signals fiercely generated at the beginning but immediately went down (**Fig. R7**), further confirming that the Fe@N-C-800 catalyst could continuously activate H_2O_2 for the generation of HO^\bullet . Since the Fe center had a lower work function than that of the carbon layer¹, the electrons in Fe NPs could successively transfer to the carbon surface, thus favoring the adsorption of H_2O_2 onto the pyridinic-N site and lower the energy barrier for H_2O_2 activation, which was supported by the XPS and density functional theory analyses. Thanks again!

Fig. R7 EPR spectra of DMPO-•OH adduct signal in the system of conventional Fenton. Condition: [Ferrous ions]₀ = 15 mg L⁻¹, [H₂O₂]₀ = 1 × 10⁻³ M, T = 25 °C.

References

1. Deng, J., Deng, D., Bao, X. Robust catalysis on 2D materials encapsulating metals: concept, application, and perspective. *Adv. Mater.* **29**, 1606967 (2017).

Comment 8 For the degradation of different kinds of organic pollutants, why the concentrations of pharmaceuticals and that of phenolic compounds and dyes are different.

Answer: Thanks for your valuable comment. According to the “Integrated Wastewater Discharge Standard” of China (GB 8798—1996), the aromatic contaminants discharged into wastewater treatment plant should meet Class 3 standards, where the concentrations range from 1 ~ 5 mg L⁻¹, hence the concentration of phenolic compounds and dyes in this work was set to be 50 × 10⁻⁶ M. Besides, as one of the emerging contaminants, pharmaceuticals have recently been recognized as significant water pollutants that may cause undesirable effects on humans and the ecosystem. These pharmaceuticals originate from diverse sources and no standard has been established till now. Generally, the emerging contaminants often exist in the concentrations that range from ng L⁻¹ to µg L⁻¹ (similar to 10⁻⁹ ~ 10⁻⁶ M)¹. Therefore, the concentration of pharmaceuticals in this work was set to be 10 × 10⁻⁶ M. Thanks again!

References

1. Pal, A., He, Y., Jekel, M., Reinhard, M., Gin, KY-H. Emerging contaminants of public health significance as water quality indicator compounds in the urban water cycle. *Environ. Int.* **71**, 46–62 (2014).

Comment 9 The symbol “MW” and Supplementary Figure 2 should be explained.

Answer: Thanks for your valuable comment. M_w represents the weight-average molecular weight, a statistical data that is widely used to characterize the molecular weight of polymer materials. In this work, M_w was employed as an index to verify the occurrence of dopamine polymerization, which had been reported over 10000 in the previous research¹. Similarly, M_p , M_n , M_z , M_{z+1} and M_v in the table under Supplementary **Fig. 2** are also the common index for molecular weight evaluation, which represents peak molecular weight, number-average molecular weight, z-average molecular weight, z+1 average molecular weight and viscosity average molecular weight, respectively. In addition, the explanation for Supplementary **Fig. 2** is as follow: the abscissa represents the logarithm of molecular weight (MW), which enlarges along the abscissa to the right. The left ordinate in Supplementary **Fig. 2** represents the signal intensity of each MW, which corresponds to the blue curve. Generally, the higher of the signal intensity refers to the more molecules in this MW. Therefore, the blue curve can also be considered as the MW distribution curve of the polymer. The right ordinate in Supplementary **Fig. 2** corresponds to the result of the cumulative integral, that is, the red curve in the figure slowly rises from 0% to 100%. Any point on the curve represents the proportion of the molecules less than or equal to the corresponding MW. With the help of cumulative integral curve, the proportion of molecules in any MW range could be directly estimated. For example, it could be directly found from Supplementary **Fig. 2** that molecules with the MW between 1000 and 10,000 account for about 70% of the total. According to your comment, the relevant annotation has been added in the Supplementary Information. Thanks again!

References

1. Delparastan, P., Malollari, KG., Lee, H., Messersmith, PB. Direct evidence for the polymeric nature of polydopamine. *Angew. Chem.* **131**, 1089–1094 (2019).

Comment 10 How about the adsorption of organic pollutants on the catalyst considering its large surface area? All removal curves start from $C/C_0=1$, is the removal the contribution of both adsorption and degradation?

Answer: Thanks for your kind comment. The adsorption effect of Fe@N-C-800 catalyst to SMX has been given in **Fig. R8**. As can be seen, without adding H_2O_2 to the system, the catalyst could only achieve 11.2% of SMX removal in 6 min, suggesting the negligible contribution of catalyst adsorption for SMX removal. Moreover, in the process of realistic wastewater treatment, the adsorption and degradation of pollutants were generally carried out simultaneously. For better reflecting the value of Fe@N-C-800 in practical application, the weak adsorption effect was not considered separately in the experiment. Thank you again!

Fig. R8 The SMX removal performance of the Fe@N-C-800 with or without the addition of H_2O_2 . Condition: $[Fe@N-C-800] = 50 \text{ mg L}^{-1}$, $[SMX]_0 = 10 \times 10^{-6} \text{ M}$, $[H_2O_2]_0 = 1 \times 10^{-3} \text{ M}$, $T = 25 \text{ }^\circ\text{C}$.

Reviewer: 3

In this manuscript, Wang et al. reported a polymerization confinement strategy for synthesizing a carbon nanosheet-supported Fe nanoparticle catalyst (Fe@N-C) with impressive catalytic performance toward the Fenton reaction. The subject—designing high-performance heterogeneous catalysts for Fenton reaction and understanding the activity origin is of high interest and importance, and has still been widely investigated due to its application prospect in various fields. Despite the synthesis is relatively facile and holds the potential for large-scale production of Fe@N-C and the catalyst showed the good catalytic activity and H₂O₂ utilization efficiency, the reviewer cannot see sufficient new insights or breakthroughs revealed by the manuscript to advance the field.

The authors presented a long discussion about the high metal loading and ultrafine size of the Fe NPs to highlight the advantage of the self-polymerization confinement strategy. However, similar M–N–C catalysts synthesized through the grinding-polymerization-coordination-pyrolysis process have already been reported considerably and high-density cluster or single-atom catalysts can also be obtained through this method. The actual activity centers are the pyridinic-N moieties and the Fe NPs function as the electron donors, so the emphasis on the high-loading and small-size characteristics and the structural evolution process of Fe@N-C somewhat sidestep the discussion about what really counts, which has been elaborated clearly in a considerable amount of work. In addition, the interaction between Fe NPs and adjacent N moieties has not been fully elaborated: in addition to electron transfer, can the Fe NPs improve the number and stability of active N sites? The synergistic interaction between Fe NPs and adjacent N sites should be investigated and interpreted more.

Also, several previous works have reported that N-coordinated metal catalysts exhibit high catalytic reactivity toward Fenton or Fenton-like reactions (Fe: ACS Nano 2018, 12, 9441; Adv. Mater. 2022, 34, 2110653; Co: J. Am. Chem. Soc. 2018, 140, 12469; Cu: Angew. Chem. Int. Ed. 2022, 61, e202207268; Mn: J. Am. Chem. Soc. 2019, 141, 12005). Are there any advantages to utilizing Fe NPs as the regulator of pyridinic-N? Do other M@N-C (M = Co, Cu, Ni, Ag) analogues synthesized through the same

method have similar Fenton activity? I understand this might be a lot of additional work, but could you be more specific about the role Fe NPs play? Based on these facts and given the high standard for the publication in Nat. Commun., the reviewer does not believe the manuscript contains enough innovative, significant content to be considered for publication in this journal.

Answer: The authors appreciate the valuable comments from Reviewer 3. We really understand the concerns of Reviewer 3 about the sufficient novelty required for publishing the manuscript in such a high impact journal and we want to further elaborate the breakthroughs presented in this work: **1) a universal and efficient strategy to synthesize a series of M-N-C materials with both high dispersion and metal loadings.** It is widely acknowledged that the overall catalytic activity of a catalyst closely depends on the number of catalytic sites, besides its intrinsic activity¹. Therefore, it is of great importance to improve the metal loadings without aggregation for the synthesis of high-performance M-N-C catalysts². Recently, tremendous efforts have been paid to synthesize such materials and we have listed the synthetic details of latest researches that can highlight the superiority of our material and synthetic method (**Table R3**). As can be seen, our strategy owns the significant advance in synthesizing the M-N-C catalysts in terms of production efficiency and metal loading. Actually, the synthesis of M-N-C catalysts usually adopt the small organic molecules as complexing agents, which will undergo the thermal polymerization and carbonization process during calcination. In this case, the drastic structural evolution of these complexes under high temperature will lead to the severe metal aggregation³. Moreover, compared with the van der Waals forces and coordination bonds in the recently reported carbon-metal precursors^{4,5}, the formation of strong covalent bonds during polymerization can provide sufficient confinement to prevent the aggregation of metal species under high temperature⁶. Till now, only a very few reports have performed the polymerization process prior to calcination for the synthesis of M-N-C catalysts, while these polymerization reactions generally require substantial cost and time-consuming treatment^{7,8}. As far as we know, it is the first time that such a swift grind-assisted polymerization process is applied to synthesize the M-N-C catalysts with high metal

loading, which may open new avenues for the efficient production of high-performance M@N-C catalysts for diverse applications. **2) Simultaneously achieve the high catalytic activity and H₂O₂ utilization efficiency in Fenton reaction.** To clearly reveal the remarkable performance of the catalyst obtained in this work, we have researched the latest state-of-the-art M-N-C catalysts for comparison (**Table R4**). Despite the varied reaction parameters set for catalytic performance evolution, all the catalysts can efficiently activate the oxidant to degrade pollutants within several minutes. However, none of these researches have paid attention to the oxidant utilization efficiency during the reaction. In fact, it is very difficult to improve the oxidant utilization efficiency at a relatively high reaction rate because the fiercely produced high concentration of radicals will be consumed by self-quenching in solution. Some researches even have to slow down the H₂O₂ decomposition rate or successively add the oxidant into the reaction solution to improve the oxidant utilization efficiency^{9,10}. Considering the high cost percentage of H₂O₂ in Fenton reaction (over 50%), it is of great significance to simultaneously achieve the high catalytic activity and H₂O₂ utilization efficiency. In this work, the obtained catalyst exhibits the dual high-efficiency performance in Fenton reaction through the continuous generation of surface-bound [•]OH confined on the catalyst, thus overcoming the trade-off between reaction rate and H₂O₂ utilization efficiency, which has not yet been achieved before. Based on the above analyses, we believe that the findings in this work owns sufficient novelty and can take a step toward the desired Fenton-like catalyst for realistic applications.

To better reveal the interaction between Fe NPs and adjacent N sites, we first conducted the XPS test and found that compared to the N-doped carbon, a negative shift in the binding energy for pyridinic-N was observed in Fe@N-C-800 (**Fig. R9a**), suggesting that the introduced Fe NPs could donate electrons to the N-doped carbonaceous matrix and increase the charge density on pyridinic-N moieties. Moreover, both N-C-800 and Fe@N-C-800 catalysts had the similar N contents around 3.0 at%, while the presence of Fe NPs could significantly regulate the N configuration, where the relative content of pyridinic-N showed a marked increase from 34.5% to 40.3%

(**Fig. R9b**). Since the pyridinic-N with lone-pair electrons preferred to coordinate with metal NPs, the above results suggested that Fe NPs had a stabilization effect on pyridinic-N moieties during high temperature calcination¹¹. In addition, density functional theory (DFT) calculations also confirmed that the embedded Fe NPs could donate electrons to the N-doped carbon layer and significantly modulate the electronic structure on the carbon surface, which could favor the adsorption of H₂O₂ onto the pyridinic-N site and lower the energy barrier for H₂O₂ activation. Although a certain amount of work has been done to elaborate the synergistic interaction between metal and N sites to improve the catalytic performance, most of them pointed out that the reaction mainly occurred on the metal sites. While in this work, both experimental results and DFT analyses revealed that the reaction started on the pyridinic-N moieties, and these sites could further confine the $\cdot\text{OH}$ on the catalyst surface, giving rise to unprecedented performance in the Fenton reaction.

To further verify the role of Fe NPs, we compared the catalytic performance of N-C-800 and Fe@N-C-800, and the catalyst without Fe NPs displayed a drastically decreased activity (**Fig. R10**), implying that the underlying Fe NPs could significantly boost the catalytic process. Besides, we also evaluated the catalytic performance of other M@N-C (M = Co, Cu, Ni, Ag) catalysts and the results were shown in **Fig. R11**. Obviously, all the catalysts could activate H₂O₂, while their catalytic performance were inferior than that of Fe@N-C-800. These results should be attributed to the lower work function of Fe NPs than that of Co, Cu and Ni¹², which could transfer more electrons to the carbon surface for the activation of H₂O₂. As for Ag, although it had relatively small work function values, the weak chemical interaction between Ag NPs and carbon layer suppressed the electron transfer due to the minimized orbital overlaps¹³, resulting in the unsatisfactory catalytic performance in SMX removal. Besides the Fenton reaction, the authors believe that these M@N-C with high metal loadings can find their application in oxygen reduction reaction¹⁴, hydrogen production¹⁵, carbon dioxide reduction¹⁶ and other fields. Thanks again for your valuable comment!

Table R3 The synthetic method comparison of recently reported M-N-C catalysts.

Number	Reagent and method	Time required to fabricate the precursor before calcination	Metal Loading
1	CNT + NMP + pyrrole + (NH ₄) ₂ S ₂ O ₈ + cobalt porphyrin + SOCl ₂ + DMF, etc. Clicking confinement method	Over 70 h	Co (1.98 wt%) ⁶
2	Lysozyme + Zn(NO ₃) ₂ ·6H ₂ O + Co(NO ₃) ₂ ·6H ₂ O + 2-methylimidazole Stirring and freeze drying	Over 12 h	Co (2.14 wt%) ¹⁴
3	NaOH + phthalic acid + CuSO ₄ ·5H ₂ O + KCl-KBr salt Molten salt assisted pyrolysis	Over 29 h	Cu (11.2 wt%) ²
4	Potassium citrate + H ₂ SO ₄ + Fe(NO ₃) ₃ ·9H ₂ O + α-D-glucose + melamine Cascade anchoring strategy	Over 10 h	Fe (12.1 wt%) ¹
5	Ferrocene + thiourea + PFTA + pyrrole + ethanol + APS Multilayer stabilization method	Over 24 h	Fe (15.3 wt%) ¹⁷
6	FeSO ₄ ·7H ₂ O + 2-methylimidazole + PVP + methanol + melamine Ball milling method	Over 14 h	Fe (18.2 wt%) ⁵
7	Zn(NO ₃) ₂ ·6H ₂ O + 2-methylimidazole + KCl+ NiCl ₂ ·6H ₂ O Two-step annealing method	Over 55 h	Ni (23 wt%) ¹⁸
8	Formaldehyde + dicyandiamide + Fe(NO ₃) ₃ ·9H ₂ O Coordinated polymer pyrolysis	Over 12 h	Fe (30.00 wt%) ⁸
9	Pyrene + HNO ₃ + ammonia + IrCl ₃ ·xH ₂ O + urea Graphene quantum dot confinement	Over 80 h	Ir (41.6 wt% or 3.84 at%) ³
This work	Dopamine + FeCl ₃ ·6H ₂ O Grind-assisted polymerization	Only 5 min	Fe (30.5 wt%)

Table R4 The catalytic performance comparison of recently reported M-N-C catalysts.

Sample	Contaminant	Oxidant	Catalyst (mg L ⁻¹)	K _{obs} (min ⁻¹)	Oxidant utilization efficiency (%)
FeN _x /g-C ₃ N ₄ ⁶	MB	H ₂ O ₂	500	/	/
FeCo-NC ¹⁹	BPA	PMS	100	1.252	/
SA-Fe/CN ²⁰	BPA	PMS	20	/	/
Cu ₁ /NG ²¹	BPA	PDS	100	1.428	/
Mn-CN ²²	Oxalic acid	H ₂ O ₂ /O ₃	100	/	/
Fe-SAC ²³	BPA	PMS	200	0.104	/
Co-N ₂ ²⁴	BPA	PMS	200	0.695	/
SA-Cr/PN-g-C ₃ N ₄ ²⁵	BPA	H ₂ O ₂	200	0.096	/
Fe ₁ /CN ²⁶	4-CP	PMS	500	0.53	/
Co-SA ²⁷	CIP	PMS	200	0.105	/
Fe@N-C-800 (this work)	SMX	H ₂ O ₂	50	0.818	84.1

**Fig. R9** **a** High resolution of N 1s XPS spectra of N-C-800 and Fe@N-C-800. **b** The content percentages of different N in N-C-800 and Fe@N-C-800.

Fig. R10 The catalytic performance of N-C-800/H₂O₂ and Fe@N-C-800/H₂O₂ systems in SMX degradation. Condition: [Catalyst] = 50 mg L⁻¹, [SMX]₀ = 10 × 10⁻⁶ M, [H₂O₂]₀ = 1 × 10⁻³ M, T = 25 °C.

Fig. R11 Time profiles of SMX removal in different catalysts activated H₂O₂ systems. Condition: [Catalyst] = 50 mg L⁻¹, [SMX]₀ = 10 × 10⁻⁶ M, [H₂O₂]₀ = 1 × 10⁻³ M, T = 25 °C.

References

1. Zhao, L. et al. Cascade anchoring strategy for general mass production of high-loading single-atomic metal-nitrogen catalysts. *Nat. Commun.* **10**, 1278 (2019).
2. Zhao, X., Kong, X., Wang, F., Fang, R., Li, Y. Metal Sub-nanoclusters Confined within Hierarchical Porous Carbons with High Oxidation Activity. *Angew. Chem. Int. Ed.* **60**, 10842–10849 (2021).

3. Xia, C. et al. General synthesis of single-atom catalysts with high metal loading using graphene quantum dots. *Nat. Chem.* **13**, 887–894 (2021).
4. Qin, WK. et al. Reticular Synthesis of Hydrogen-Bonded Organic Frameworks and Their Derivatives via Mechanochemistry. *Angew. Chem.* **27**, e202202089 (2022).
5. An, S. et al. High-density ultra-small clusters and single-atom Fe sites embedded in graphitic carbon nitride (g-C₃N₄) for highly efficient catalytic advanced oxidation processes. *ACS Nano* **12**, 9441–9450 (2018).
6. Zhao, C-X. et al. A clicking confinement strategy to fabricate transition metal single-atom sites for bifunctional oxygen electrocatalysis. *Sci. Adv.* **8**, eabn5091 (2022).
7. He, X. et al. A versatile route to fabricate single atom catalysts with high chemoselectivity and regioselectivity in hydrogenation. *Nat. Commun.* **10**, 3663 (2019).
8. Xiong, Y. et al. Gram-scale synthesis of high-loading single-atomic-site Fe catalysts for effective epoxidation of styrene. *Adv. Mater.* **32**, 2000896 (2020).
9. Niu, H., He, D., Yang, Y., Lv, H., Cai, Y., Liang, Y. Long-lasting activity of Fe⁰-C internal microelectrolysis-Fenton system assisted by Fe@C-montmorillonites nanocomposites. *Appl. Catal. B Environ.* **256**, 117820 (2019).
10. Chen, F. et al. Efficient decontamination of organic pollutants under high salinity conditions by a nonradical peroxymonosulfate activation system. *Water Res.* **191**, 116799 (2021).
11. Ha, Y. et al. Atomically dispersed Co-pyridinic N-C for superior oxygen reduction reaction. *Adv. Energy Mater.* **10**, 2002592 (2020).
12. Wang, J., Wang, S-Q. Surface energy and work function of fcc and bcc crystals: Density functional study. *Surf. Sci.* **630**, 216–224 (2014).
13. Ha, Y., Kang, S., Ham, K., Lee, J., Kim, H. Experimental and density functional theory corroborated optimization of durable metal embedded carbon nanofiber for oxygen electrocatalysis. *J. Phys. Chem. Lett.* **10**, 3109–3114 (2019).

14. Tang, C. et al. Coordination tunes selectivity: two-electron oxygen reduction on high-loading molybdenum single-atom catalysts. *Angew. Chem.* **132**, 9256–9261 (2020).
15. Zhang, Z. et al. Enhanced and stabilized hydrogen production from methanol by ultrasmall Ni nanoclusters immobilized on defect-rich h-BN nanosheets. *Proc. Natl. Acad. Sci.* **117**, 29442–29452 (2020).
16. Zhang, T., Bui, J.C., Li, Z., Bell, A.T., Weber, A.Z., Wu, J. Highly selective and productive reduction of carbon dioxide to multicarbon products via in situ CO management using segmented tandem electrodes. *Nat. Catal.* **5**, 202–211 (2022).
17. Zhou, Y. et al. Multilayer stabilization for fabricating high-loading single-atom catalysts. *Nat. Commun.* **11**, 5892 (2020).
18. Hai, X. et al. Scalable two-step annealing method for preparing ultra-high-density single-atom catalyst libraries. *Nat. Nanotechnol.* **17**, 174–181 (2022).
19. Li, X. et al. Single cobalt atoms anchored on porous N-doped graphene with dual reaction sites for efficient Fenton-like catalysis. *J. Am. Chem. Soc.* **140**, 12469–12475 (2018).
20. Xiong, Y. et al. Single-Atom Fe Catalysts for Fenton-Like Reactions: Roles of Different N Species. *Adv. Mater.* **34**, 2110653 (2022).
21. Wang, B. et al. A Site Distance Effect Induced by Reactant Molecule Matchup in Single-Atom Catalysts for Fenton-Like Reactions. *Angew. Chem.* **134**, e202207268 (2022).
22. Guo, Z. et al. Single-atom Mn–N₄ site-catalyzed peroxone reaction for the efficient production of hydroxyl radicals in an acidic solution. *J. Am. Chem. Soc.* **141**, 12005–12010 (2019).
23. Liang, X., Wang, D., Zhao, Z., Li, T., Gao, Y., Hu, C. Coordination Number Dependent Catalytic Activity of Single-Atom Cobalt Catalysts for Fenton-Like Reaction. *Adv. Funct. Mater.* **38**, 2203001 (2022).
24. Chen, F. et al. Molecular engineering toward pyrrolic N-rich M–N₄ (M= Cr, Mn, Fe, Co, Cu) single-atom sites for enhanced heterogeneous Fenton-like reaction. *Adv. Funct. Mater.* **31**, 2007877 (2021).

25. Zhang, L.S. et al. Carbon Nitride Supported High-Loading Fe Single-Atom Catalyst for Activation of Peroxymonosulfate to Generate $^1\text{O}_2$ with 100% Selectivity. *Angew. Chem. Int. Ed.* **60**, 21751–21755 (2021).
26. Miao, J. et al. Spin-state-dependent peroxymonosulfate activation of single-atom M–N moieties via a radical-free pathway. *ACS Catal.* **11**, 9569–9577 (2021).
27. Mi, X. et al. Almost 100% peroxymonosulfate conversion to singlet oxygen on single-atom CoN_{2+2} sites. *Angew. Chem.* **133**, 4638–4643 (2021).

Comment 1 The Fe content detected by XPS is 2.3 wt%, much lower than that of the ICP-OES results. It seems that the Fe NPs are encapsulated in thick carbon layers. The carbon layer structure should be characterized in detail.

Answer: Thank you for the valuable comment. The carbon layer around the Fe NPs was characterized by the high-resolution transmission electron microscopy (HRTEM). As shown in **Fig. R12**, the Fe NPs covered by several carbon layers was clearly observed. Moreover, as the pyrolysis temperature increased to 1000 °C, we could find that the thickness of the carbon layers gradually increased from 2 nm to 10 nm, which further illustrated that the metal nanoparticles were encapsulated inside the carbon layers. According to your suggestion, the relevant discussion has been supplemented in the revised manuscript. Thanks again!

Fig. R12 HRTEM images of the Fe@N-C-800 and Fe@N-C-1000 materials.

Comment 2 The stability of the catalyst is of great importance. Therefore, the structure, surface composition, and morphology of the catalyst after the reaction should be provided.

Answer: Thank you for this helpful comment. To reveal the stability of the Fe@N-C-800, characterizations including XRD, TEM, XPS and Mössbauer spectrum were performed. As shown in **Fig. R13 a and b**, there were no obvious changes in the patterns of XRD and TEM image after the reaction, suggesting that the whole structure and morphology of Fe@N-C-800 were well maintained during the Fenton reaction. In addition, no significant variation was detected in the XRD pattern of Fe@N-C-800 after storage for two months (**Fig. R13c**), indicating that the encapsulation strategy could effectively protect the Fe center and improve the stability of the catalyst. To better reveal the active sites for Fenton reaction in Fe@N-C-800, XPS was performed to investigate the composition variation after reaction. As shown in **Fig. R13d**, the pyridinic-N content decreased (40.3% to 34.9%) while the pyrrolic-N content increased significantly (37.1% to 42.4%) after reaction, which could be attributed to the transformation of pyridinic-N to pyridonic-N (400.1 eV). Moreover, we further explored the composition variation of Fe species by ^{57}Fe Mössbauer spectroscopy measurements. As shown in **Fig. R13 e and f**, the relative proportion of Fe^{III} increased (from 12.3% to 27.1%) while that of $\alpha\text{-Fe}$ decreased (from 64% to 51.3%), suggesting that the encapsulated Fe NPs played an important role in the Fenton reaction by donating electrons. Combined the above results with H_3PO_4 poisoning experiment and density functional theory calculations in this work, the authors confirmed that the high activity of Fe@N-C-800 catalyst stemmed from pyridinic-N coordinated Fe sites, which exhibited high-efficiency performance and stable properties in Fenton reaction. The relevant discussion has been supplemented in the revised manuscript. Thanks again!

Fig. R13 **a** XRD patterns of Fe@N-C-800 before and after reaction. **b** TEM images of Fe@N-C-800 after reaction. **c** XRD pattern of Fe@N-C-800 after storage for two months. **d** High resolution of N 1s XPS spectra of Fe@N-C-800 before and after reaction. ^{57}Fe Mössbauer spectrum of Fe@N-C-800 **e** before and **f** after reaction.

Comment 3 The removal performance of contaminants of Fe@N-C-600 and Fe@N-C-700 is very close to that of Fe@N-C-800 despite their high iron leaching. Can the catalysts maintain high removal performance after the leaching of water-soluble iron species?

Answer: Thanks for the valuable comment. According to your suggestion, we tested the reusability of Fe@N-C-600/700 catalysts in order to explore the effect on catalyst performance after ion leaching. As shown in **Fig. R14**, the Fe@N-C-600/700 catalysts showed excellent performance in the first cycle due to the high iron ion leaching, achieving over 95 % of SMX removal within 6 min. However, these catalysts were powerless in the subsequent two cycles compared with the Fe@N-C-800 catalyst, further confirming that the Fe NPs encapsulated in carbon layers could significantly enhance the stability and reusability. The relevant discussion has been supplemented in the revised manuscript. Thanks again!

Fig. R14 **a** Cyclic degradation of SMX by the recycled Fe@N-C-600/700. **b** Leached iron ions of Fe@N-C-X synthesized under different pyrolysis temperatures. **c** Cyclic degradation of SMX by the recycled Fe@N-C-800. Condition: [SMX]₀ = 10 × 10⁻⁶ M, [H₂O₂]₀ = 1 × 10⁻³ M, T = 25 °C.

Comment 4 *State-of-the-art M-N-C catalysts for Fenton reactions should be added in Supplementary Table 5 and 6 for comparison.*

Answer: Thanks for your valuable comment. To clearly reveal the remarkable performance of the catalyst obtained in this work, we have researched the latest state-of-the-art M-N-C catalysts for comparison. From **Table R5** we can find that the catalysts obtained in this work not only exhibit the high catalytic activity and TOC removal in pollutant degradation, but also achieve the impressive oxidant utilization efficiency that is often overlooked in the previous researches. Therefore, the authors believe that the synthesized catalysts with outstanding comprehensive performance hold the potential to advance the Fenton reaction for practical application. According to your suggestion, the comparison result has been supplemented in the Supporting

Information. Thanks again!

Table R5 The catalytic performance comparison of recently reported M-N-C catalysts.

Sample	Contaminant	Oxidant	Catalyst (mg L ⁻¹)	K_{obs} (min ⁻¹)	TOC removal (%)	Oxidant utilization efficiency (%)
FeN _x /g-C ₃ N ₄ ¹	MB	H ₂ O ₂	500	/	/	/
FeCo-NC ²	BPA	PMS	100	1.252	/	/
SA-Fe/CN ³	BPA	PMS	20	/	/	/
Cu ₁ /NG ⁴	BPA	PDS	100	1.428	/	/
Mn-CN ⁵	Oxalic acid	H ₂ O ₂ /O ₃	100	/	/	/
Co-N ₂ ⁶	BPA	PMS	200	0.695	72.9	/
SA-Cr/PN-g-C ₃ N ₄ ⁷	BPA	H ₂ O ₂	200	0.096	/	/
Fe ₁ /CN ⁸	4-CP	PMS	500	0.53	59.2	/
Co-N-CNTs ⁹	SMX	PMS	100	0.157	/	/
Co-SA ¹⁰	CIP	PMS	200	0.105	60.0	/
FeCN ¹¹	BPA	PAA	150	0.75	58.3	/
Fe-N ₄ -PC ¹²	SMX	PMS	30	0.21	72.8	/
Fe@N-C-800 (this work)	SMX	H ₂ O ₂	50	0.818	75.4	84.1

References

1. Zhao, C-X. et al. A clicking confinement strategy to fabricate transition metal single-atom sites for bifunctional oxygen electrocatalysis. *Sci. Adv.* **8**, eabn5091 (2022).
2. Li, X. et al. Single cobalt atoms anchored on porous N-doped graphene with dual reaction sites for efficient Fenton-like catalysis. *J. Am. Chem. Soc.* **140**, 12469–12475 (2018).
3. Xiong, Y. et al. Single-Atom Fe Catalysts for Fenton-Like Reactions: Roles of Different N Species. *Adv. Mater.* **34**, 2110653 (2022).
4. Wang, B. et al. A Site Distance Effect Induced by Reactant Molecule Matchup in Single-Atom Catalysts for Fenton-Like Reactions. *Angew. Chem.* **134**, e202207268 (2022).

5. Guo, Z. et al. Single-atom Mn–N₄ site-catalyzed peroxone reaction for the efficient production of hydroxyl radicals in an acidic solution. *J. Am. Chem. Soc.* **141**, 12005–12010 (2019).
6. Liang, X., Wang, D., Zhao, Z., Li, T., Gao, Y., Hu, C. Coordination Number Dependent Catalytic Activity of Single-Atom Cobalt Catalysts for Fenton-Like Reaction. *Adv. Funct. Mater.* **38**, 2203001 (2022).
7. Chen, F. et al. Molecular engineering toward pyrrolic N-rich M–N₄ (M= Cr, Mn, Fe, Co, Cu) single-atom sites for enhanced heterogeneous Fenton-like reaction. *Adv. Funct. Mater.* **31**, 2007877 (2021).
8. Zhang, L.S. et al. Carbon Nitride Supported High-Loading Fe Single-Atom Catalyst for Activation of Peroxymonosulfate to Generate ¹O₂ with 100% Selectivity. *Angew. Chem. Int. Ed.* **60**, 21751–21755 (2021).
9. Miao, J. et al. Spin-state-dependent peroxymonosulfate activation of single-atom M–N moieties via a radical-free pathway. *ACS Catal.* **11**, 9569–9577 (2021).
10. Mi, X. et al. Almost 100% peroxymonosulfate conversion to singlet oxygen on single-atom CoN₂₊₂ sites. *Angew. Chem.* **133**, 4638–4643 (2021).
11. Chen, F., Liu, L.L., Wu, J.H., Rui, X.H., Chen, J.J., Yu, Y. Single-Atom Iron Anchored Tubular g-C₃N₄ Catalysts for Ultrafast Fenton-Like Reaction: Roles of High-Valency Iron-Oxo Species and Organic Radicals. *Adv. Mater.* **34**, 2202891 (2022).
12. Wang, J. et al. Facile Synthesis of Atomic Fe–N–C Materials and Dual Roles Investigation of Fe–N₄ Sites in Fenton-Like Reactions. *Adv. Sci.* **8**, 2101824 (2021).

Comment 5 *The authors should correct some mistakes in the manuscript, such as the ordinate of Supplementary Figure 7a.*

Answer: Thank you for pointing out this mistake. According to your suggestion, the mistake in the Supplementary Figure 7a has been corrected. In addition, the manuscript has been examined in detail. Thank you again for your kind suggestion.

Reviewer: 4

In this paper, the authors reported a facile and efficient strategy to synthesize a series of ultrafine metal NPs embedded N-doped 2D carbon nanosheets with high metal loadings. The Fe@N-C-800 nanocatalyst synthesized via this strategy showed outstanding performance for H₂O₂ activation. The authors gave a solid proof to demonstrate the mechanism for the exceptional catalytic performance. After carefully scrutiny of this work, the reviewer thinks this work demonstrates sufficient novelty and high quality in the interpretation of the outcomes. Thus, it is recommended the publication of this work after minor revision.

Comment 1 The authors claimed that the essence of self-polymerization confinement strategy were chelation and polymerization of dopamine. Catalysts with the size of metal particles much larger than Fe@N-C-800 are obtained by blocking these two effects, respectively. Please compare their catalytic performance.

Answer: Thanks for your kind comment. According to your suggestion, we evaluated the catalytic performance of Fe(acac)₃@N-C and Fe@N-C (without oxidization) in Fenton reaction for the degradation of SMX. As expected, the Fe(acac)₃@N-C and Fe@N-C (without oxidization) showed poor catalytic performance compared with that of Fe@N-C-800. The reasons for these results could be attributed to the severe agglomeration of metal NPs on the catalyst by blocking the polymerization and chelation effect, which greatly reduced the utilization of metal NPs.

Fig. R15 The performance of different catalysts for degradation of SMX. Condition: [Catalyst] = 50 mg L⁻¹, [SMX]₀ = 10 × 10⁻⁶ M, [H₂O₂]₀ = 1 × 10⁻³ M, T = 25 °C.

Comment 2 Considering the important role of PDA in synthesizing the high-performance catalyst, can other catecholic derivatives to be used as precursors to synthesize the efficient catalyst?

Answer: Thanks for the valuable suggestion. We have selected the several representative catecholic derivatives such as gallic acid, tannic acid and catechol to prepare the Fe embedded carbonaceous catalysts. As shown in **Fig. R16**, all the materials displayed the characteristic diffraction peaks indexed to (110) and (200) phase for α -Fe. Besides, these catalysts could also be used to activate H_2O_2 for the efficient removal of SMX. Among them, the catalyst prepared by dopamine exhibited the superior catalytic performance due to the N doping on the carbon layer, which was consistent with the density functional theory analyses. This comment provides an important guideline to our following work for the rational design of carbon precursors.

Thanks again!

Fig. R16 a XRD patterns of Fe embedded carbonaceous materials prepared by different catecholic derivatives and **b** their catalytic performance toward SMX. Condition: $[\text{SMX}]_0 = 10 \times 10^{-6} \text{ M}$, $[\text{H}_2\text{O}_2]_0 = 1 \times 10^{-3} \text{ M}$, $[\text{Catalyst}] = 50 \text{ mg L}^{-1}$, $T = 25 \text{ }^\circ\text{C}$.

Comment 3 The authors compared the performance of different catalysts in Figure 4a. In the figure, N-C-800 is the product of dopamine calcined at 800 °C, which the reviewer thinks is inaccurate. It should be more reasonable to use polydopamine as the precursor to synthesize N-C-800.

Answer: Thanks for the kind suggestion, we are sorry for not clearly describing the

synthesis condition of N-C-800. In this work, the N-C-800 was exactly prepared through the calcination of dopamine after polymerization. The synthesis procedure of N-C-800 has been supplemented in the revised manuscript. Thanks again!

Comment 4 The effect of water matrix should be included in practical application. At least tap water and surface water should be employed for that purpose.

Answer: Thank you for your constructive comment. As pointed out by the reviewer, the performance of catalyst in different water matrix plays a pivotal role in evaluating its practical application potential. Here, we selected two types of water including tap water and lake water. As shown in **Fig. R17**, the removal rates of SMX could still reach to 98.2% and 95.6% within 6 min in tap water and lake water, respectively, suggesting the strong flexibility of the Fe@N-C-800/H₂O₂ system to the actual water. The relevant discussion has been supplemented in the Supporting Information. Thanks again!

Fig. R17 Effect of water matrix on the removal of SMX in the Fe@N-C-800/H₂O₂ system. Condition: [Catalyst] = 50 mg L⁻¹, [SMX]₀ = 10 × 10⁻⁶ M, [H₂O₂]₀ = 1 × 10⁻³ M, T = 25 °C.

Comment 5 It is exciting that the catalyst reported in this work could achieve the both high catalytic activity and H₂O₂ utilization efficiency, the reasons should be explained.

Answer: Thanks for the kind suggestion. In this work, the exceptional performance of Fe@N-C-800 in terms of the high utilization of H₂O₂ and efficient elimination of organic pollutants could be attributed to the following critical factors: 1) the nanometer thickness of the porous carbon 2D framework enabled the sufficient mass transfer for

both H₂O₂ and target contaminants, 2) the uniform distribution of ultrafine Fe nanoparticles with high loading effectively coordinated with the pyridinic-N and offered the abundant active sites for the adsorption and activation of H₂O₂, 3) the continuous generation but not fierce production of •OH_{surface} with an increased lifetime could significantly improve the reaction probabilities towards contaminants and decrease the risk of self-quenching side reactions that usually occurred in homogeneous Fenton system, such as the unfavorable reactions of •OH between Fe^{II}, H₂O₂ and itself. All the above factors collectively render the obtained Fe@N-C-800 catalyst highly efficient in Fenton reaction.

Comment 6 The authors claimed the continuous generation of •OH in the Fe@N-C-800/H₂O₂ system, so more evidence should be supplemented to confirm the durability of the synthesized catalyst.

Answer: Thanks for the valuable suggestion. To further confirm the durability of the synthesized catalyst and its feasibility in treating large amount of wastewater, we conducted a pilot-scale setup equipped with a transparent acrylic plexiglass tank, peristaltic pump, electric motor stirrer and miniaturized Teflon filter holder for the continuous degradation of aromatic organics (**Fig. R18a**), where a certain concentration of H₂O₂ and aromatic organics were injected at a given time interval of 1 hour. Specifically, the aromatic wastewater containing SMX, 4-CP and MB was prepared with 10 L tap water, and the initial concentration of SMX, 4-CP and MB were 10 × 10⁻⁶ M, 50 × 10⁻⁶ M and 50 × 10⁻⁶ M, respectively, at time = 0 hour. For treating the aromatic wastewater, 1 g Fe@N-C-800 was added into the wastewater at time = 0 hour, 2 mL H₂O₂ (30%) was added every hours, additional SMX, 4-CP and MB with the same content as the initial values were added at every hours from time = 1 to 11 hour. To our delight, over 97% removal efficiency of three aromatic organics (SMX, 4-CP and MB) could be achieved after 13 hours continuous reaction (**Fig. R18b**), demonstrating the excellent durability of Fe@N-C-800. According to your suggestion, the relevant result has been added in the Supporting Information. Thank you again!

Fig. R18 **a** Photograph of the experimental device. **b** 13 hours continuous degradation of 10 L aromatic organics (total added [SMX] = 30.36 mg L⁻¹, [4-CP] = 77.16 mg L⁻¹ and [MB] = 192 mg L⁻¹). Condition: [H₂O₂]₀ = 1 × 10⁻³ M, [Fe@N-C-800] = 50 mg L⁻¹, T = 25 °C.

Reviewer: 5

Comment 1 How to prove that the Fe NPs are encapsulated in carbon layers not supported on the surface? In some previous studies, they think Fe NPs are the active sites, while the electrons will be provided by carbon and transferred via the N atoms. Can you exclude this possibility?

Answer: Thanks for the valuable comment. In this work, the encapsulation of Fe NPs by the carbon layer was confirmed by the HRTEM images as well as the remarkable difference of Fe content detected by ICP-OES and XPS results. As shown in **Fig. R19**, a well-defined core-shell structure with carbon shells and a metal core was obviously detected. In addition, the overall iron loading was determined to be 30.5 wt% by ICP-OES, which was much higher than the value detected by the XPS test (2.3 wt%), further indicating the existence of protective carbon layers coated on the Fe NPs¹.

As for the active site on the catalyst, we first used phosphate acid (H₃PO₄) as the poisoning agent to block the pyridinic-N site since it could protonate pyridinic-N to form pyridinic-N-H². As shown in **Fig. R20**, a significant inhibition effect was observed in the Fe@N-C-800/H₂O₂ system, while little effect about the SMX removal was found for the homogenous Fenton reaction, indicating the crucial role of pyridinic-N site in Fe@N-C-800/H₂O₂ system. Moreover, unlike the previously reported Fenton catalysts,

Fe@N-C-800 exhibited the superior catalytic ability at a circumneutral pH (**Fig. R21**). This result should be attributed to the protonation of pyridinic-N sites on the carbon layers in an acidic environment³, resulting in decreased active sites and weakened activity. While in the homogenous Fenton system, the acid environment could remarkably improve the reaction efficiency, which was totally different from the behavior in Fe@N-C-800/H₂O₂ system. Besides, we also synthesized the pristine Fe NPs through the liquid-phase reduction method by using NaBH₄ to further evaluate the catalytic performance of Fe NPs in Fenton reaction⁴. As shown in **Fig. R22**, only 11.2% of SMX was removed in the Fe NPs/H₂O₂ system, indicating that the Fe NPs had inferior catalytic ability due to the rapid passivation effect in water. The above results collectively confirmed that the reaction was occurred on the N-doped carbon layer while the embedded Fe NPs could significantly promote this process through electron transfer. Thanks again!

Fig. R19 HRTEM image of the Fe@N-C-800

Fig. R20 Effects of H_3PO_4 on catalytic performance of the different systems for SMX removal: **a** Fe@N-C-800/ H_2O_2 system, **b** the conventional Ferrous/ H_2O_2 system. Condition: $[Fe@N-C-800] = 0.05 \text{ g L}^{-1}$, $[Ferrous \text{ ions}] = 2 \text{ mg L}^{-1}$, $[H_2O_2]_0 = 1 \times 10^{-3} \text{ M}$, $[SMX]_0 = 10 \times 10^{-6} \text{ M}$, $[H_3PO_4] = 0.1 \text{ M}$, $T = 25 \text{ }^\circ\text{C}$.

Fig. R21 Effect of initial pH on SMX removal in **a** Fe@N-C-800/ H_2O_2 and **b** homogenous Ferrous/ H_2O_2 systems. Condition: $[SMX]_0 = 10 \times 10^{-6} \text{ M}$, $[H_2O_2]_0 = 1 \times 10^{-3} \text{ M}$, $[Fe@N-C-800] = 50 \text{ mg L}^{-1}$, $[Ferrous \text{ ions}] = 2 \text{ mg L}^{-1}$, $T = 25 \text{ }^\circ\text{C}$.

Fig. R22 Comparison of catalytic performance of the Fe@N-C-800 and Fe NPs. Condition: [Catalyst] = 50 mg L⁻¹, [SMX]₀ = 10 × 10⁻⁶ M, [H₂O₂]₀ = 1 × 10⁻³ M, T = 25 °C.

References

1. Yan, P. et al. Highly efficient K-Fe/C catalysts derived from metal-organic frameworks towards ammonia synthesis. *Nano Res.* **12**, 2341–2347 (2019).
2. Mamtani, K. et al. Probing the oxygen reduction reaction active sites over nitrogen-doped carbon nanostructures (CN_x) in acidic media using phosphate anion. *ACS Catal.* **6**, 7249–7259 (2016).
3. Guo, D., Shibuya, R., Akiba, C., Saji, S., Kondo, T., Nakamura, J. Active sites of nitrogen-doped carbon materials for oxygen reduction reaction clarified using model catalysts. *Science* **351**, 361–365 (2016).
4. Mo, Y., Xu, J., Zhu, L. Molecular Structure and Sulfur Content Affect Reductive Dechlorination of Chlorinated Ethenes by Sulfidized Nanoscale Zerovalent Iron. *Environ. Sci. Technol.* **56**, 5808–5819 (2022).

Comment 2 Moreover, the author claimed that Fe@N-C have porous structure (L355), the characterization for the porous structure should be performed. This porous structure should expose some Fe sites, which will also contact with H₂O₂ and act as active sites.

Answer: Thanks for the comment. Nitrogen-sorption measurements were carried out to measure the porous characteristics of the catalysts calcined at different temperatures.

As shown in **Fig. R23**, all the samples presented the similar type IV of N₂ adsorption-desorption isotherms. Besides, the calculated average pore size of the catalysts centered at 2 ~ 20 nm, indicating the existence of mesoporous structure. As the reviewer said, the porous structure might expose some Fe sites and then activate H₂O₂. However, we found that the concentration of leached ions for the Fe@N-C-800 catalyst was only 0.076 mg L⁻¹ by ICP-OES analysis, which could be attributed to the well protected carbon layer on the Fe NPs. In this case, the content of exposed Fe active sites should be relatively small and have negligible contribution to the SMX removal. Combined with comment 1, we could conclude that reaction was mainly occurred on the N-doped carbon layer while the embedded Fe NPs could significantly promote this process through electron transfer. Thanks again!

Fig. R23 a N₂ adsorption/desorption isotherms of Fe@N-C-X. Effect of pyrolysis temperatures on the b pore size and c specific surface area of Fe@N-C-X.

Comment 3 The author demonstrated that the high activity of Fe@N-C stem from pyridinic-N coordinated Fe sites, which can induce electron transfer from catalyst to H₂O₂. This coordinated structure is usually in the atomic catalyst, where N and Fe has a special ratio. While in this manuscript, the Fe is in the form of nanoparticles, the ratio of Fe is much larger than N, how did the pyridinic-N coordinate with the Fe atoms? I believe that most of the Fe atoms are not coordinated with N. How about the percentage of N? Are there enough pyridinic-N to coordinated with Fe? How about the evolution of these N species after Fe leaching?

Answer: Thanks for the comment. Just as the reviewer speculated, the Fe species were in the form of nanoparticles on the catalyst, implying that most of the Fe atoms exhibited the Fe-Fe coordination structure. In this case, only the surface part of Fe nanoparticles was coordinated with the N moieties (the N contents was calculated to be 3.0 at% according to the XPS test), thus facilitating the electron transfer from the Fe core to the carbon surface, which had been also reported in other M-N-C catalysts¹. In this work, the electronic interaction between Fe nanoparticles and N-doped carbon layer was confirmed by the XPS test. As shown in **Fig. R24**, compared to the N-C-800, a negative shift in the binding energy for pyridinic-N was observed, suggesting that the introduced Fe NPs could donate electrons to the N-doped carbonaceous matrix and increase the charge density on pyridinic-N moieties². To explore the role of metal, a harsh acid-leaching process was adopted to remove the encapsulated Fe nanoparticles. From the XPS result we could find that the peak of pyridinic-N shifted back to the higher binding energy similar to that of N-C-800, further confirming the strong interaction between the Fe nanoparticles and N-doped carbon layers.

Fig. R24 High resolution of N 1s XPS spectra of N-C-800, Fe@N-C-800 before and after acid washing.

References

1. Wang, XR. et al. Identifying the key role of pyridinic-N–Co bonding in synergistic electrocatalysis for reversible ORR/OER. *Adv. Mater.* **30**, 1800005 (2018).
2. Marshall-Roth, T. et al. A pyridinic Fe-N₄ macrocycle models the active sites in Fe/N-doped carbon electrocatalysts. *Nat. Commun.* **11**, 5283 (2020).

Comment 4 What's the exact active sites for this Fe@N-C material? In fact, the activation of H₂O₂ should be an electron consuming process. So what are these electrons come from? Since lots of electrons are needed, the catalysts after the long-time testing should be totally different compared with the pristine one. The materials after reaction should be carefully analyzed, including the Fe, the N and carbon species.

Answer: Thank you very much for this constructive comment. Based on the H₃PO₄ poisoning experiment, pH adaptability experiment, iron ion leaching analysis as well as the density functional theory calculations, the authors confirmed that the direct active site on Fe@N-C-800 catalyst was the pyridinic-N moieties, while the embedded Fe NPs

could donate electrons to the N-doped carbonaceous matrix and increase the charge density on pyridinic-N moieties, thus facilitating the adsorption and activation of H₂O₂ on the pyridinic-N site. Since lots of electrons were needed in the Fenton reaction, it was great of significance to improve the metal loading on the catalyst, which was also the important goal achieved in this work. According to your suggestion, the catalyst after reaction were carefully analyzed. As shown in **Fig. R25 a and b**, there were no obvious changes in the patterns of XRD and TEM image after the reaction, suggesting that the whole structure and morphology of Fe@N-C-800 were well maintained during the Fenton reaction. In addition, we further explored the composition variation of Fe species by ⁵⁷Fe Mössbauer spectroscopy measurements. As shown in **Fig. R25 c and d**, the relative proportion of Fe^{III} increased (from 12.3% to 27.1%) while that of α-Fe decreased (from 64% to 51.3%), suggesting that the encapsulated Fe NPs played an important role in the Fenton reaction by donating electrons. Moreover, the XPS results showed that the pyridinic-N content decreased (40.3% to 34.9%) after reaction (**Fig. R25e**), while inconspicuous changes were detected in C 1s spectra (**Fig. R25f**), indicating that the reaction mainly occurred on the pyridinic-N site of carbon layers.

In addition, the successive degradation experiments were performed to reveal the cycling performance of the catalyst. As shown in **Fig. R26**, although the removal rate was gradually decreased during the consecutive degradation tests, there was still over 95% of SMX removed after six cycling tests, which could be attributed to the abundant electrons supplied by the Fe NPs and the continuous H₂O₂ activation on the N-doped carbon layers. The relevant discussion has been supplemented in the revised manuscript. Thanks again!

Fig. R25 **a** XRD patterns of Fe@N-C-800 before and after reaction. **b** TEM image of Fe@N-C-800 after reaction. ^{57}Fe Mössbauer spectrum of Fe@N-C-800 **c** before and **d** after reaction. High resolution of **e** N 1s and **f** C 1s XPS spectra of Fe@N-C-800 before and after reaction.

Fig. R26 Cyclic degradation of SMX by the recycled Fe@N-C-800. Condition: $[\text{Fe@N-C-800}] = 50 \text{ mg L}^{-1}$, $[\text{SMX}]_0 = 10 \times 10^{-6} \text{ M}$, $[\text{H}_2\text{O}_2]_0 = 1 \times 10^{-3} \text{ M}$, $T = 25 \text{ }^\circ\text{C}$.

Comment 5 Can the relationship between leached Fe and consumed H_2O_2 be analyzed? This will be helpful for the analysis of the active sites.

Answer: Thanks for the valuable suggestion. The concentration of leached ions for the Fe@N-C-800 catalyst was 0.076 mg L^{-1} by ICP-OES analysis. Accordingly, the

equivalent amount of ferrous ions was added into the 1×10^{-3} M H_2O_2 solution containing the pollutant to start the reaction. As shown in **Fig. R27**, only 4.2% of SMX was removed in the homogenous Fenton system with the minimal consumption of H_2O_2 (2.1%), which further confirmed that the leached Fe played the insignificant role in activating H_2O_2 for the removal of pollutant. Thanks again!

Figure R27. **a** Time profiles of SMX removal in different systems. **b** H_2O_2 consumption efficiency of different catalysts. Condition: $[\text{Fe@N-C-800}] = 50 \text{ mg L}^{-1}$, $[\text{Ferrous ions}] = 0.076 \text{ mg L}^{-1}$, $[\text{H}_2\text{O}_2]_0 = 1 \times 10^{-3} \text{ M}$, $[\text{SMX}]_0 = 10 \times 10^{-6} \text{ M}$, $T = 25 \text{ }^\circ\text{C}$.

We believe that we have addressed the reviewers' comments, and will be glad to answer any further questions you might have. We tried our best to improve the manuscript and made some changes in the revised manuscript. These changes will not influence the content and framework of the paper. And here we did not list the changes but marked in red in revised paper.

We appreciate for Editors/Reviewers' warm work earnestly, and hope that the correction will meet with approval.

Once again, thank you very much for your comments and suggestions.

Sincerely yours,

Mingyang Xing

REVIEWER COMMENTS

Reviewer #1 (Remarks to the Author):

I consider the authors did the proper changes in the revised version of the manuscript, now the work is suitable for publication.

Reviewer #3 (Remarks to the Author):

Comments:

Although the authors made some improvements in the revision. The reviewer still cannot support its publication in this high-impact journal due to the following major concerns.

1) Fe@N-C materials were well reported and investigated in tremendous literatures. Using the dopamine or polydopamine as precursor is also well established. The reviewer did not see the advance of the present strategy compared with the reported ones. The higher metal loading can be achieved in the literatures. The application of such M-N-C like catalysts were also well investigated previously. The reviewer does not think the manuscript presented significant new insights or progress in the field to justify the publication in this journal.

2) In Figure 1d, the iron-based particles are so small. The selected area for electron diffraction experiment would not be single nanoparticle. How did the authors get the single-crystalline diffraction pattern? Such data are pretty weird.

3) In the synthesis, the authors claimed some facts without evidences or support, for example, 1) the polymerization of dopamine, 2) self-assembled layered structure of PDA, 3) intrinsic moisture adsorption and oxidizing capability as the precursor, 4) ferric ions were thoroughly coordinated with the abundant catechol groups in PDA, etc. All these claims are lack of evidences or support.

4) The lattice fringes in Figure 1f are hardly distinguished. How to exclude the existence of iron carbides, which share the very similar crystalline structure with iron.

5) The authors claimed that the Fe nanoparticles is around 4 nm. However, the XRD patterns shown in Figure R13, Supplementary Fig. 9, Supplementary Fig. 31, etc. showed very sharp peaks, which suggests much larger should exist in the sample. These data are not consistent.

6) Similar, the nanoparticles size calculated from TEM images or XRD patterns in Figure 6 should be compared and justified.

7) The catalytic performance is directly related to the density of accessible active sites. The N content is not very high compared with ones reported. The majority of Fe particles are embedded in the carbon. What makes the present catalyst unique or special for this application?

8) The authors claimed that "well-defined core-shell structure with carbon shells and a metal core was obviously detected (Supplementary Fig. 6)". This claim cannot be supported by the Supplementary Fig. 6.

9) The authors claimed that "Fe NPs had a stabilization effect on pyridinic-N moieties during high temperature calcination.", "Compared to the N-C-800, a negative shift in the binding energy for pyridinic-N was observed, suggesting that the introduced Fe NPs could donate electrons to the N-doped carbonaceous matrix and increase the charge density on pyridinic-N moieties". These claims might not be correct and lack of other evidences. The measurement and deconvolution of XPS data are very tricky.

10) The reviewer cannot list all the issues in the claims or writing in the manuscript.

Reviewer #4 (Remarks to the Author):

The revision is satisfactory.

Reviewer #5 (Remarks to the Author):

This manuscript has been improved greatly after revision, I recommend it can be accepted after minor revision, the detailed comments are as follows:

1. Does the thickness of the carbon layer will affect the activity? As we know, the N doped carbon cannot activate H₂O₂, why did it become active by encapsulate Fe core? If the Fe core will provide electrons to the NC, what will itself become? The Fe should be oxidized at the same time, then how the oxidant come into inside the carbon shell? The detailed electron and oxidant transfer routes should be demonstrated.
2. How about the synthesis cost of this method? The high temperature (>500 oC) will lead very expensive cost. Is it possible to decrease the synthesis temperature, thus to obtain a good catalytic performance as well as a good industrial potential?
3. The stability of Fe@N-C-700 is far more different with that of Fe@N-C-800. More discussion should be added to explain. The structures of them are totally different?
4. The author used H₃PO₄ to block the pyridinic-N, thus to conclude that pyridinic-N site is the active site. However, the introduction of H₃PO₄ will decrease the surface Fe ions by the formation of FePO₄. This should also decrease the activity.

A Response to the Comments of the Editor and Reviewers

Manuscript ID: NCOMMS-22-34544A

Title: "Universal Polymer Tethering Strategy Toward High-Metal-Loading Nanocatalyst for Dual High-Efficiency Fenton Reaction "

We appreciate the editor's timely response and the reviewers' comments concerning our manuscript. The comments are very valuable and helpful for revising the paper. At the same time, the comments provide an important guideline to our following work. According to the comments, we revised the manuscript carefully and point-to-point responses to comments are shown in the following. The revised parts have been marked in red in the manuscript.

Respective answer to the detailed comments:

Reviewer: 1

Comment I consider the authors did the proper changes in the revised version of the manuscript, now the work is suitable for publication.

Answer: We are grateful for the time and effort Review 1 has spent in reviewing our manuscript. We appreciate the reviewer's recommendation very much.

Reviewer: 3

Although the authors made some improvements in the revision. The reviewer still cannot support its publication in this high-impact journal due to the following major concerns.

Comment 1 Fe@N-C materials were well reported and investigated in tremendous literatures. Using the dopamine or polydopamine as precursor is also well established. The reviewer did not see the advance of the present strategy compared with the reported

ones. The higher metal loading can be achieved in the literatures. The application of such M-N-C like catalysts were also well investigated previously. The reviewer does not think the manuscript presented significant new insights or progress in the field to justify the publication in this journal.

Answer: We thank the reviewer's valuable comment on the novelty of our work, and we are pleased to clarify this issue. We agree that quite a number of M-N-C materials have been reported in the last few years. However, evaluating the impact/novelty of work should rely on what it has done to solve the remaining scientific or engineering challenges of the studied materials or reactions, not only the number of published papers. Considering the fundamental role of M-N-C catalysts in extensive heterogeneous catalysis domains, the efforts for developing high-performance M-N-C catalysts through a swift, general and scalable approach have never been stopped. Just recently, the two-step annealing method (*Nat. Nanotechnol.* **17**, 174-181, 2022) and mechanochemical abrasion method (*Nat. Nanotechnol.* **17**, 403-407, 2022) have been reported to synthesize a series of M-N-C catalysts^{1, 2}. However, as we have mentioned in our introduction, the present strategies for the synthesis of high-metal-loading M-N-C catalysts still rely heavily on the delicate control of support matrix and synthetic procedures, which generally require substantial cost and time-consuming treatment, hampering expansion to scale-up for industrialization.

Aiming at this issue, a self-polymerization confinement strategy is proposed in this work via a grind-assisted pyrolysis method, which requires only several minutes to prepare the metal-precursor without additional solvent and auxiliary operation. Although a few reports have adopted the dopamine to synthesize the M-N-C materials (*Adv. Mater.* **30**, 1706508, 2018; *Nat. Commun.* **12**, 6766, 2021)^{3, 4}, our approach owns several breakthroughs compared with these studies: First, in the previous studies, the reaction between dopamine and metal ions are always performed in the solution requiring dozens of hours, which can cause inevitable waste of the residual reagents and need an extra energy-consuming drying process before calcination. Thus, the advance in terms of production efficiency and eco-friendly economy in our approach can be established. Second, due to the hydrophobic interaction of dopamine in the solution,

the final materials always present the spherical morphology in the previous works. In contrast, the catalyst with 2D nanosheet structure is obtained in this study through the stacking of planar polydopamine formed by mono-catechol-Fe coordination at the quasi-solid-state. Compared with the spherical morphology, it is obvious that the nanosheet with abundant accessible active sites can facilitate the mass transport and electron transfer, ensuring high catalytic activity. Third, the previous studies have mainly focused on the metal-chelating capability and nitrogen-containing groups when selecting dopamine as the carbon precursor, ignoring its polymer nature for the confinement of metal ions during pyrolysis process. In our case, the important role of polymer confinement effect is experimentally validated through the gel permeation chromatography analysis and polymerization blocking test, which can conduct the following research for the synthesis of M-N-C materials. More importantly, the selection of dopamine is only the proof-of-concept test to confirm the important roles of chelating groups and polymer nature in synthesizing the well-dispersed M-N-C materials. Following this principle, we can even create abundant copolymerization systems through intentionally selecting the catechol derivatives and amine compounds based on Schiff base and Michael addition reactions, which can offer a broad space to regulate the comprehensive property of the final catalysts and will greatly advance their application potential in diverse fields. Therefore, our approach to synthesize a series of M-N-C with high metal loading via a polymer tethering strategy is conceptually novel and can also be extended to other copolymerization systems.

As for the performance in Fenton reaction, it is acknowledged that besides the catalytic activity, the H_2O_2 utilization efficiency is also a critical evaluation index for practical application. However, it is very difficult to improve the oxidant utilization efficiency at a relatively high reaction rate because the fiercely produced high concentration of radicals will be consumed by self-quenching in solution. Considering the high cost percentage of H_2O_2 in Fenton reaction (over 50%), it is of great significance to simultaneously achieve the high catalytic activity and H_2O_2 utilization efficiency. In this work, the obtained Fe@N-C catalyst exhibits the dual high-efficiency performance in Fenton reaction through the continuous generation of surface-bound

$\cdot\text{OH}$ on the catalyst, where the surface-bound $\cdot\text{OH}$ with an increased lifetime could significantly improve the reaction probabilities towards contaminants and decrease the risk of self-quenching side reactions, thus overcoming the trade-off between reaction rate and H_2O_2 utilization efficiency. To the best of our knowledge, there is no report that has achieved such high reaction rate and H_2O_2 utilization efficiency simultaneously before. Therefore, we anticipate that the continuous generation of surface-bound $\cdot\text{OH}$ reported in this work can be one of the solutions to the dilemma of the present dual high-efficiency Fenton systems.

Taken together, we reasonably believe our strategy on fabrication of high-metal-loading M-N-C catalysts for achieving dual high-efficiency performance in Fenton reaction is both conceptually novel and practically valuable, and the existence of the related work could not affect the novelty of our work. Thanks again!

References

1. Hai, X. et al. Scalable two-step annealing method for preparing ultra-high-density single-atom catalyst libraries. *Nat. Nanotechnol.* **17**, 174-181 (2022).
2. Han, GF. et al. Abrading bulk metal into single atoms. *Nat. Nanotechnol.* **17**, 403-407 (2022).
3. Han, A. et al. A polymer encapsulation strategy to synthesize porous nitrogen-doped carbon-nanosphere-supported metal isolated-single-atomic-site catalysts. *Adv. Mater.* **30**, 1706508 (2018).
4. Kumar, A. et al. Moving beyond bimetallic-alloy to single-atom dimer atomic-interface for all-pH hydrogen evolution. *Nat. Commun.* **12**, 6766 (2021).

Comment 2 In Figure 1d, the iron-based particles are so small. The selected area for electron diffraction experiment would not be single nanoparticle. How did the authors get the single-crystalline diffraction pattern? Such data are pretty weird.

Answer: We sincerely thank the reviewer for the comment. Just as the reviewer said, the metal NPs in Fig. 1d were too small and we could not get the unambiguous SAED

data in this condition. Therefore, the aforementioned SAED data in Fig. 1d was specially collected in a magnified area to get a more clearer crystal structure of the metal NPs. To avoid misunderstanding, the SAED data collected from other areas was removed in the revised manuscript and **Fig. R1** only demonstrated the uniform dispersion of Fe NPs on the carbon support. Thank you again for your valuable comment.

Fig. R1 TEM image of the Fe@N-C-800.

Comment 3 In the synthesis, the authors claimed some facts without evidences or support, for example, 1) the polymerization of dopamine, 2) self-assembled layered structure of PDA, 3) intrinsic moisture adsorption and oxidizing capability as the precursor, 4) ferric ions were thoroughly coordinated with the abundant catechol groups in PDA, etc. All these claims are lack of evidences or support.

Answer: Thanks for the valuable comment, we have provided solid evidences to support the description in the synthesis part. To confirm the polymerization of dopamine, we have adopted the gel permeation chromatography (GPC) to obtain the weight-average molecular weight (M_w) of polydopamine (**Fig. R2**), and the value was calculated to be 1.48×10^4 , which was consistent with the previous report that used single-molecule force spectroscopy to validate the polymeric nature of polydopamine (*Angew. Chem. Int. Ed.* **58**, 1077-1082, 2019)¹. Besides, we have collected the Fe-PDA complex after mixing for 5 min to detect the precursor structure before calcination. The sample was freeze-drying and then transferred for transmission electronic microscopy

detection. As shown in **Fig. R3**, the Fe-PDA precursor presented the disc structure with the in-plane size from 50 to 200 nm, verifying the self-assembled layered structure of PDA. Moreover, the intrinsic moisture adsorption of $\text{FeCl}_3 \cdot 6\text{H}_2\text{O}$ could be observed when placing the metal salts under air ambience, where the $\text{FeCl}_3 \cdot 6\text{H}_2\text{O}$ was gradually deliquesced within 20 min (**Fig. R4**). The oxidizing capability of $\text{FeCl}_3 \cdot 6\text{H}_2\text{O}$ was also explored by detecting the UV-Vis absorbance at 420 nm with reaction time for various dopamine solutions, which was considered as the indicator of dopamine polymerization (*Angew. Chem. Int. Ed.* **55**, 3054-3057, 2016)². As shown in **Fig. R5**, the color change and UV-Vis absorbance at 420 nm with reaction time conformed that the Fe^{III} could readily trigger the oxidization of dopamine. Additionally, according to the previous reports, the coordination bond between catechol and Fe^{III} is one of the highest stability constants of metal-ligand complexes (*Angew. Chem. Int. Ed.* **43**, 448-450, 2004)³. Therefore, the Fe^{III} could be readily coordinated with the catechol groups in dopamine. Moreover, we also adopted the UV-Vis spectroscopy to verify the catechol- Fe^{III} coordination bond. As depicted in **Fig. R6**, an absorption peak around 720 nm was clearly observed for the dopamine- Fe^{III} mixture solution, suggesting that the coordination was mainly in the form of mono-catechol- Fe^{III} (*Thin Solid Films* **600**, 76-82, 2016)⁴. To be prudent, the claim of “ferric ions were thoroughly coordinated with the abundant catechol groups in PDA” was changed to “ferric ions could be coordinated with the abundant catechol groups in PDA” in the manuscript. Thanks again!

MW Averages

Peak No	Mp	Mn	Mw	Mz	Mz+1	Mv	PD
1	1502	2321	14787	98983	160649	9802	6.37096

Fig. R2 GPC data of the Fe-PDA complex under the normal grind-assisted polymerization conditions.

Fig. R3 TEM image of Fe-PDA complex.

Fig. R4 Time course of the $\text{FeCl}_3 \cdot 6\text{H}_2\text{O}$ deliquescence situation under air.

Fig. R5 a Photographs of dopamine and dopamine-Fe^{III} mixture solutions with different grinding times (0.03 g sample dissolved in 100 mL solution). **b** Time-dependence of absorbance at 420 nm for various dopamine solutions formed in different polymerization conditions.

Fig. R6 UV-vis spectra of solution containing dopamine and dopamine-Fe^{III} mixture.

References

1. Delparastan, P. et al. Direct evidence for the polymeric nature of polydopamine. *Angew. Chem. Int. Ed.* **131**, 1089-1094 (2019).
2. Zhang, C. et al. CuSO₄/H₂O₂-induced rapid deposition of polydopamine coatings with high uniformity and enhanced stability. *Angew. Chem. Int. Ed.* **55**, 3054-3057 (2016).

3. Sever, MJ. et al. Metal-mediated cross-linking in the generation of a marine-mussel adhesive. *Angew. Chem. Int. Ed.* **43**, 448-450 (2004).
4. Xu, H. et al. Spraying layer-by-layer assembly film based on the coordination bond of bioinspired polydopamine-Fe^{III}. *Thin Solid Films* **600**, 76-82 (2016).

Comment 4 The lattice fringes in Figure 1f are hardly distinguished. How to exclude the existence of iron carbides, which share the very similar crystalline structure with iron.

Answer: Thanks for your valuable comment. The HRTEM image in Fig. 1f had been improved and the distinct lattice fringes could be found. As shown in **Fig. R7**, the lattice fringe of the nanoparticle was calculated to be 0.206 nm, which was close to the characteristic (110) crystal plane of metallic iron (PDF# 06-0696) with the lattice spacing of 0.203 nm. Although the iron carbides owned the similar crystalline structure with iron, the characteristic diffraction peaks corresponding to (211), (031) and (220) planes of iron carbides (PDF# 35-0772) were not found in the XRD results, but an obvious diffraction peak of metallic iron at 44.8° was clearly detected, suggesting that the nanoparticle in Fig. 1f was mainly composed of metallic iron. To further analyze the iron species in Fe@N-C, ⁵⁷Fe Mössbauer spectroscopy measurements were conducted. As shown in **Fig. R8**, the major sextet (Sext1), with a high hyperfine magnetic field (330.65 KOe) and a zero-isomer shift, was assigned to body-centered cubic (bcc) α -Fe coordination with a relative content of 64.0% (*J. Am. Chem. Soc.* **138**, 635-640, 2016)¹, while the relative content of iron carbides (Sext4) with a high hyperfine magnetic field of 207.12 KOe was only 16.5% (*Angew. Chem.* **133**, 8971-8977, 2021)². These results proved that the main species of iron in the catalyst was metallic iron, accompanied by a small amount of iron carbides. Thank you again.

Fig. R7 **a** HRTEM image of the Fe@N-C-800. **b** Intensity profile of selected crystallized zone in **a** showing the consistent lattice spacings of 0.206 nm.

Fig. R8 ⁵⁷Fe Mössbauer spectrum of pristine Fe@N-C-800 measured at room temperature.

References

1. Kramm, UI. et al. On an easy way to prepare metal-nitrogen doped carbon with exclusive presence of MeN₄-type sites active for the ORR. *J. Am. Chem. Soc.* **138**, 635-640 (2016).
2. Xing, Y. et al. Fe/Fe₃C boosts H₂O₂ utilization for methane conversion overwhelming O₂ generation. *Angew. Chem. Int. Ed.* **133**, 8971-8977 (2021).

Comment 5 The authors claimed that the Fe nanoparticles is around 4 nm. However,

the XRD patterns shown in Figure R13, Supplementary Fig. 9, Supplementary Fig. 31, etc. showed very sharp peaks, which suggests much larger should exist in the sample. These data are not consistent.

Answer: Thanks for the valuable comment. Following the reviewer's suggestion, we have collected the XRD data of all the samples again and found that these newly obtained XRD patterns were consistent with the previous results. Generally, the peak intensity of XRD patterns would decrease as the nanoparticles get smaller. However, many other aspects including instrumental and environmental factors as well as the sample property (atomic type, plane multiplicity, Lorentz factor, absorption effect and so on) would affect the peak intensity and shape of XRD patterns (*X-ray diffraction and identification and analysis of clay minerals*, 1989)¹, which should also be considered. For instance, the two-dimensional nature and magnetism of the prepared catalysts in this work might form the preferred orientation during XRD test, which could cause the unusual increase of the peak intensity (*Materials*, **14**, 473-476, 2007)². Actually, small-size nanoparticles do not always show the broad peak shape. Similar phenomenon could also be observed in previous studies, where the metal NPs ranged from 3 to 15 nm also exhibited the sharp diffraction peaks (**Fig. R9**, *J. Am. Chem. Soc.* **142**, 7116–7127, 2020; *Angew. Chem. Int. Edit.* **133**, 10463-10471, 2021; *J. Am. Chem. Soc.* **129**, 10602–10606, 2007; *J. Mater. Chem. A* **7**, 6849–6858, 2019)³⁻⁶. These results confirmed that many factors could contribute to the peak intensity and shape of the XRD patterns.

Besides, the authors also agreed with the reviewer that the large NPs could most likely to enhance the intensity of diffraction peak. Since the mole ratio of iron salt and dopamine was 2:1 in the preparation stage, it might be inevitable that some iron species were not complexed with the dopamine and then formed the large NPs during calcination. To this end, we also prepared the pristine Fe NPs through the liquid-phase reduction method by using NaBH₄ to further compare the morphology between pristine Fe NPs and Fe@N-C-800. As shown in **Fig. R10a**, the pristine Fe NPs was composed of necklace-like nanowires assembled by Fe spheres with average size of 200 nm. However, none of these Fe spheres was observed in Fe@N-C-800 catalyst throughout the field of view (**Fig. R10b**). Moreover, we also evaluated the catalytic performance

of Fe NPs in Fenton reaction. As shown in **Fig. R10c**, only 11.2% of SMX was removed in the Fe NPs/H₂O₂ system, indicating that the Fe NPs had inferior catalytic ability due to the rapid passivation effect in water. The above results collectively confirmed that even there existed a few of large metal NPs contributing to the peak intensity in XRD patterns, the low percentage of large metal NPs could hardly affect the statistics of size distribution and the final Fenton catalytic ability of the Fe@N-C-800 in this work. The XRD patterns were mainly conducted to identify the crystalline structure of the prepared catalyst, and the relevant interpretation concerning the peak intensity was also added in the supplementary information. Thanks again!

Fig. R9 a Co₂Fe₁@NC synthesized by the metastable rock salt oxide-mediated strategy. **b** Co_{0.5}Fe_{0.5}C derived from the calcination of phenolic polymer aerogel. **c** Palladium nanoparticles synthesized by the assistance of dendritic polymer. **d** nZVI/biochar composite prepared by the sheering/carbonization process.

Fig. R10 **a** SEM image of pristine Fe NPs prepared by the liquid-phase reduction method. **b** SEM image of Fe@N-C-800. **c** Comparison of catalytic performance of the Fe@N-C-800 and Fe NPs. Condition: [Catalyst] = 50 mg L⁻¹, [SMX]₀ = 10 × 10⁻⁶ M, [H₂O₂]₀ = 1 × 10⁻³ M, T = 25 °C.

References

1. Moore, DM. et al. *X-ray diffraction and identification and analysis of clay minerals*, London: Oxford University Press, 332 (1989).
2. Yan, S. et al. Charge and discharge curves: a unique reliable evidence for the electrochemical properties of LiCoO₂. *J. Univ. Sci. Technol. Beijing, Min. Met. Mater.* **14**, 473-476 (2007).
3. Tang, T. et al. Metastable rock salt oxide-mediated synthesis of high-density dual-protected M@NC for long-life rechargeable zinc-air batteries with record power density. *J. Am. Chem. Soc.* **142**, 7116-7127 (2020).
4. Xiao, F. et al. Selective electrocatalytic reduction of oxygen to hydroxyl radicals via 3-electron pathway with FeCo alloy encapsulated carbon aerogel for fast and

- complete removing pollutants. *Angew. Chem. Int. Ed.* **60**, 10375-10383 (2021).
5. Moisan, S. et al. General approach for the synthesis of organic-inorganic hybrid nanoparticles mediated by supercritical CO₂. *J. Am. Chem. Soc.* **129**, 10602-10606 (2007).
 6. Dai, XH. et al. Solvent-free synthesis of a 2D biochar stabilized nanoscale zerovalent iron composite for the oxidative degradation of organic pollutants. *J. Mater. Chem. A* **7**, 6849-6858 (2019).

Comment 6 Similar, the nanoparticles size calculated from TEM images or XRD patterns in Figure 6 should be compared and justified.

Answer: Thanks for the comment. Considering that the precursor composition and preparation method of other M@N-C composites are similar to the Fe@N-C material, the reasons for the different results of TEM images and XRD patterns in determining the nanoparticle size should be identical, which have been listed in *Comment 5*. Similarly, the relevant discussion has been added in the supplementary information. Thanks again!

Comment 7 The catalytic performance is directly related to the density of accessible active sites. The N content is not very high compared with ones reported. The majority of Fe particles are embedded in the carbon. What makes the present catalyst unique or special for this application?

Answer: Thanks for the valuable comment. Although the majority of Fe NPs were embedded in the carbon support, the high loading of metallic Fe could successively transfer electrons to the N-doped carbon surface via the intimately interacted metal-carbon interface, enabling a sustained Fenton-like reaction for the continuous generation of hydroxyl radicals, which had been verified by XPS, EPR, DFT analysis as well as the contaminant removal experiments in this work. Based on these results, the exceptional performance of Fe@N-C-800 in terms of the high utilization of H₂O₂ and efficient elimination of organic pollutants could be attributed to the following critical factors: 1) the nanometer thickness of the porous carbon 2D framework enabled

the sufficient mass transfer for both H₂O₂ and target contaminants, 2) the uniform distribution of ultrafine Fe nanoparticles with high loading as the electron reservoir could serially donate electrons to the N-doped carbon layer and significantly modulate the electronic structure on the carbon surface, which could favor the adsorption of H₂O₂ and lower the energy barrier for H₂O₂ activation, 3) the continuous generation but not fierce production of $\cdot\text{OH}_{\text{surface}}$ with an increased lifetime could significantly improve the reaction probabilities toward contaminants and decrease the risk of self-quenching side reactions that usually occurred in other Fenton systems, such as the unfavorable reactions of $\cdot\text{OH}$ between Fe^{II}, H₂O₂ and itself. All the above factors collectively render the obtained Fe@N-C-800 catalyst highly efficient in Fenton reaction. Additionally, the authors also agreed with the reviewer that elevating the N content might further increase the catalytic activity of the catalyst, while its impact on the H₂O₂ utilization efficiency should be carefully evaluated, and we would continue to address this issue in our following work, thanks again!

Comment 8 The authors claimed that “well-defined core-shell structure with carbon shells and a metal core was obviously detected (Supplementary Fig. 6)”. This claim cannot be supported by the Supplementary Fig. 6.

Answer: Thanks for the valuable comment. We fully understand the reviewer’s concern that the HRTEM image in Supplementary Fig. 6 might not provide convincing information to support the claim. This should be attributed to the fact that double grids were used for TEM test due to the strong magnetism of the prepared catalyst, which could increase difficulty in obtaining the clear encapsulated structure of small metal NPs in carbon support. As a result, we removed the previous result and also the claim regarding the core-shell structure from the HRTEM image. However, it is acknowledged that during pyrolysis, the carbon will first dissolve into metal crystal at high temperature and then precipitate from the C/metal solid solution during the cooling process, thus forming the carbon-wrapped structure (*J. Am. Chem. Soc.* **142**, 7116–7127, 2020)¹. This phenomenon was also verified by the Fe@N-C-1000, where the structure with carbon shell and a metal core was clearly detected due to its large nanoparticle and

increased carbon layers compared with that of Fe@N-C-800. In addition, based on the ICP-OES, XPS analysis as well as the Fenton reaction experiments, we could still confirm the existence of protective carbon layers coated on the Fe NPs in Fe@N-C-800. The relevant description has been revised in the manuscript, thanks again!

Fig. R11 HRTEM images of the Fe@N-C-1000.

References

1. Tang, T. et al. Metastable rock salt oxide-mediated synthesis of high-density dual-protected M@NC for long-life rechargeable zinc-air batteries with record power density. *J. Am. Chem. Soc.* **142**, 7116-7127 (2020).

Comment 9 The authors claimed that “Fe NPs had a stabilization effect on pyridinic-N moieties during high temperature calcination.”, “Compared to the N-C-800, a negative shift in the binding energy for pyridinic-N was observed, suggesting that the introduced Fe NPs could donate electrons to the N-doped carbonaceous matrix and increase the charge density on pyridinic-N moieties”. These claims might not be correct and lack of other evidences. The measurement and deconvolution of XPS data are very tricky.

Answer: Thanks for the comment. Recently, several researches have pointed out that the introduction of metal element into the carbon matrix could endow the catalysts a higher content of pyridinic-N (*ACS Nano*, **12**, 9441-9450, 2018, *Adv. Energy Mater.* **11**, 2100303, 2021)^{1,2}. Du et al. further demonstrated the high linear correlation of pyridinic-N and metal contents in multiple M@N-C samples (*Appl. Catal. B-Environ.* **262**, 118302, 2020)³. During calcination, the C-N bond was vulnerable due to the

relatively low binding energy (305 kJ mol^{-1}), and the pyridinic-N was more susceptible than the pyrrolic-N and graphitic-N with increasing pyrolysis temperature (*Adv. Mater.* **34**, 2113, 2022)⁴. While after the inclusion of metal, the pyridinic-N moieties with lone-pair electrons could coordinate with metal species and maintain their stability. This result could also be found in our work through the XPS test, where the relative content of pyridinic-N was increased with the existence of Fe NPs. Therefore, the description of “Fe NPs had a stabilization effect on pyridinic-N moieties during high temperature calcination” should be reasonable.

In addition, the XPS data in this work were all calibrated by taking the C 1s at 284.8 eV as a benchmark, and the deconvolution was rechecked. As shown in **Fig. R12**, compared to the N-C-800, a negative shift in the binding energy for pyridinic-N was observed, suggesting that the introduced Fe NPs could donate electrons to the N-doped carbonaceous matrix and increase the charge density on pyridinic-N moieties. To further explore the role of metal, a harsh acid-washing process was adopted to remove the encapsulated Fe NPs. From the XPS result we could find that the peak of pyridinic-N shifted back to the higher binding energy similar to that of N-C-800, further confirming the strong interaction between Fe NPs and N-doped carbon layers (the decreased content of pyridinic-N after acid-washing might be attributed to the protonation of pyridinic-N sites (*Angew. Chem. Int. Ed.* **58**, 7035-7039, 2019))⁵. Besides the XPS test, we also employed density functional theory (DFT) calculations to confirm the interaction between Fe NPs and N-doped carbon layer. Compared with the carbon structure without Fe NPs (**Fig. R13a**), the charge density difference plots displayed in **Fig. R13b** showed that the embedded Fe NPs could donate electrons to the N-doped carbon layer and significantly modulate the electronic structure on the carbon surface, resulting in an increased charge density for enhanced catalysis. Taken together, the role of Fe NPs as the electron donors could be consolidated. Thanks again!

Fig. R12 High resolution of N 1s XPS spectra of N-C-800, Fe@N-C-800 before and after acid washing.

Fig. R13 The charge density differences in the (a) N-C and (b) Fe@N-C materials, yellow and cyan regions represent electron accumulation and depletion, respectively.

References

1. An, S. et al. High-density ultra-small clusters and single-atom Fe sites embedded in graphitic carbon nitride ($g\text{-C}_3\text{N}_4$) for highly efficient catalytic advanced oxidation processes. *ACS Nano* **12**, 9441-9450 (2018).

2. Zhang, Q. et al. Electronically modified atomic sites within a multicomponent Co/Cu composite for efficient oxygen electroreduction. *Adv. Energy Mater.* **11**, 2100303 (2021).
3. Du, W. et al. Sulfate saturated biosorbent-derived Co-S@NC nanoarchitecture as an efficient catalyst for peroxymonosulfate activation. *Appl. Catal. B-Environ.* **262**, 118302 (2020).
4. Xiong, Y, et al. Single-atom Fe catalysts for Fenton-like reactions: Roles of different N species. *Adv. Mater.* **34**, 2110653 (2022).
5. Li, J. et al. Ultrahigh-loading zinc single-atom catalyst for highly efficient oxygen reduction in both acidic and alkaline media. *Angew. Chem. Int. Ed.* **58**, 7035-7039 (2019).

Comment 10 The reviewer cannot list all the issues in the claims or writing in the manuscript.

Answer: We sincerely thank the reviewer for carefully reviewing our manuscript, which can certainly improve the quality of this work. According to your suggestion, we have carefully checked the whole manuscript and some inappropriate descriptions have been revised in the manuscript. Some examples are as follows:

Page 8: We have revised “Such an ultrathin 2D sheet-like hybrid structure not only effectively protects Fe NPs from surface passivation” to “Such a 2D sheet-like hybrid structure not only effectively protects Fe NPs from surface passivation”.

Page 11: We have revised “where the relative content of pyridinic-N showed a marked increase from 34.1% to 40.4%” to “where the relative content of pyridinic-N was increased from 34.1% to 40.4%”.

Page 14: We have revised “giving rise to unprecedented performance in the following catalytic reaction” to “This interaction mode might give rise to the unexpected performance in the following catalytic reaction”.

Page 15: We have revised “The increased lifetime of $\cdot\text{OH}$ could promote the target reaction efficiency” to “The increased lifetime of $\cdot\text{OH}$ should increase the reaction probabilities towards other substances”.

The authors also admit that not all questions can be solved in one manuscript, and further work is encouraged to perform for the complement of this system. Nonetheless, the findings in this study still have important implications for M-N-C catalyst synthesis and environmental remediation related to H₂O₂-based chemical transformation. First, the metal-polyphenol based polymer network extend the scope of efficient and eco-friendly methods for synthesizing the well-dispersed M-N-C materials with high metal loading, and this nanoplatform can be a guidance for the following design of copolymerization systems through intentionally selecting other catechol derivatives and amine compounds based on Schiff base and Michael addition reactions, which can offer a broad space to regulate the comprehensive property of the final catalysts. Second, the dissociation of H₂O₂ and subsequent confinement of [•]OH on the N-doped carbon surface mediated by Fe NPs represents a new H₂O₂/catalyst interaction mode for the enhanced [•]OH formation and preservation, which can offer a novel way for the simultaneous achievement of high catalytic activity and H₂O₂ utilization efficiency in Fenton reactions. In brief, we believe that the work reported in this manuscript is both of scientific significance and practical value in catalyst synthesis and environmental remediation, and we are glad to answer any further questions from the reviewer concerning this work. Once again, we truly thank the reviewer for the insightful comments and kind suggestions.

Reviewer: 4

Comment The revision is satisfactory.

Answer: We are grateful for the time and effort Review 4 has spent in reviewing our manuscript. We appreciate the reviewer's recommendation very much.

Reviewer: 5

This manuscript has been improved greatly after revision, I recommend it can be accepted after minor revision, the detailed comments are as follows:

Comment 1 Does the thickness of the carbon layer will affect the activity? As we know, the N doped carbon cannot activate H₂O₂, why did it become active by encapsulate Fe core? If the Fe core will provide electrons to the NC, what will itself become? The Fe should be oxidized at the same time, then how the oxidant come into inside the carbon shell? The detailed electron and oxidant transfer routes should be demonstrated.

Answer: Thank you for your constructive comment. As for the Fe@N-C catalyst in this work, the underlying Fe NPs could donate electrons to the N-doped carbon layer and modulate its electronic structure via the intimately interacted metal-carbon interface, thus favoring the dissociation of H₂O₂ through obtaining electrons from the carbon surface, which had been verified by the experimental results and DFT analysis. This process might be similar to the previous report by using the metal-free N-doped carbon materials as the cathode in electron-Fenton reaction, where the pyridinic-N site could transfer an electron to activate H₂O₂ to generate [•]OH (*Appl. Catal. B-Environ.* **256**, 117774, 2019)¹. In addition, the number of carbon layers would influence the penetration of metal d-orbital electrons (*ACS. Catal.* **11**, 11129-11159, 2021)², subsequently defining the communications between Fe NPs and outermost carbon layer. Therefore, the increased carbon layers would weaken the electron donation capability of Fe NPs and result in an inferior H₂O₂ activation process. This could be one of the reasons for the decreased catalytic ability of Fe@N-C-1000 with the covered carbon shells over ten layers, while the Fe@N-C-800 with few carbon layers could readily activate H₂O₂ for the continuous generation of [•]OH.

During the Fenton reaction in this work, the Fe NPs could firstly donate electrons to the N-doped carbon layer, and the pyridinic-N site with increased charge density could then activate H₂O₂ to generate [•]OH. With the in-situ oxidation of internal Fe NPs, to maintain the electroneutrality, the excess oxidized Fe ions in the lattice would migrate outward to the surface (*Nat. Commun.* **13**, 5365, 2022)³. In this work, the Fe ions would mainly accumulate at the metal-carbon interface, and the oxidized Fe^{II} and Fe^{III} ions would finally react with the permeated H₂O and dissolved O₂ to form the iron (oxyhydr) oxide. Due to the well-protected carbon layer, the formation of iron (oxyhydr) oxide was not obvious according to the XRD result after reaction, which was conducive

to elevating the metal stability against leaching. According to your suggestion, the relevant discussion has been supplemented in the revised manuscript. Thanks again!

References

1. Haider, MR. et al. In-situ electrode fabrication from polyaniline derived N-doped carbon nanofibers for metal-free electro-Fenton degradation of organic contaminants. *Appl. Catal. B-Environ.* **256**, 117774 (2019).
2. Zhang, P. et al. Density functional theory calculations for insight into the heterocatalyst reactivity and mechanism in persulfate-based advanced oxidation reactions. *ACS Catal.* **11**, 11129-11159 (2021).
3. Dong, H. et al. Depletable peroxidase-like activity of Fe₃O₄ nanozymes accompanied with separate migration of electrons and iron ions. *Nat. Commun.* **13**, 5365 (2022).

Comment 2 How about the synthesis cost of this method? The high temperature (>500 °C) will lead very expensive cost. Is it possible to decrease the synthesis temperature, thus to obtain a good catalytic performance as well as a good industrial potential?

Answer: Thanks for the comment. The synthesis cost of the Fe@N-C in our strategy was estimated to be about RMB4.95 g⁻¹ (\$0.708 g⁻¹). The raw materials (dopamine: RMB720 per 100 g, iron (III) chloride hexahydrate: RMB68 per 500 g) made up the major cost and were purchased from Shanghai Macklin Biochemical Co., Ltd.. The tube furnace used in the synthesis process was the OTF-1200X from Hefei Kejing Co., Ltd., which possessed a maximum rated power of 1200 kW. One batch experiment required 37.84 g of iron (III) chloride hexahydrate and 13.28 g of dopamine. Then, the raw materials were calcined at 800 °C in a tube furnace for 120 min with a heating rate of 5 °C min⁻¹ to obtain 20.53 g target catalyst. The total cost of one batch was approximately RMB101.62 or \$14.54. Considering that the prepared catalysts in this work exhibited satisfactory recyclability in contaminant removal, the cost could be further reduced in real wastewater treatment. Additionally, we fully agreed that the

synthesis cost was an indispensable factor of assessing industrial potential. Just as the reviewer suggested, we tried our best to synthesize the target catalyst under the calcination temperature of 500 °C or even lower. Nevertheless, a lot of experiments demonstrated that the catalyst synthesized at 800 °C had an optimal property such as high reaction rate and prominent stability against ion leaching. Currently, we are also looking for a new strategy to further reduce the synthesis cost by creating various copolymerization systems through intentionally selecting the catechol derivatives and amine compounds based on Schiff base and Michael addition reactions, which can replace the expensive dopamine with cheaper raw materials. Thanks again!

Comment 3 The stability of Fe@N-C-700 is far more different with that of Fe@N-C-800. More discussion should be added to explain. The structures of them are totally different?

Answer: Thanks for the constructive comment. Just as the reviewer speculated, the Fe@N-C-700 was significantly less stable than the Fe@N-C-800, which was attributed to the different structure and composition. During pyrolysis, the evolution of iron ions into Fe NPs was insufficient at 700 °C, and a large amount of ferrous chloride was found on the surface of the Fe@N-C-700 by XRD and HRTEM analyses (**Fig. R14a, b**). Although the Fe^{II} could directly activate H₂O₂ to produce ·OH, the inevitably serious ion leaching occurred during reaction (**Fig. R14c**), which caused poor catalytic performance of Fe@N-C-700 in cyclic test compared with that of Fe@N-C-800 (**Fig. R14d, e**). With further increasing the calcination temperature to 800 °C, all Fe^{II} components evolved into Fe NPs. Meanwhile, the carbon would first dissolve into metal crystal at high temperature and then precipitate from the C/metal solid solution during the cooling process to form the carbon-wrapped structure (*J. Am. Chem. Soc.* **142**, 7116–7127, 2020)¹, which greatly protected the internal Fe core from the interference of the complex reaction environment. In brief, the structure of catalyst prepared at 700 °C was different from that of 800 °C, which was mainly reflected in nanoparticle composition and the presence of wrapped structure, further giving rise to the distinct catalytic performance in Fenton reaction.

Fig. R14 **a** XRD patterns of the Fe@N-C-700 and Fe@N-C-800. **b** HRTEM image of the Fe@N-C-700. **c** Leached iron ions of Fe@N-C-X synthesized under different pyrolysis temperatures. Cyclic degradation of SMX by the recycled **d** Fe@N-C-700 and **e** Fe@N-C-800 (condition: $[\text{Fe@N-C-X}] = 50 \text{ mg L}^{-1}$, $[\text{SMX}]_0 = 10 \times 10^{-6} \text{ M}$, $[\text{H}_2\text{O}_2]_0 = 1 \times 10^{-3} \text{ M}$, $T = 25 \text{ }^\circ\text{C}$).

References

- Tang, T. et al. Metastable rock salt oxide-mediated synthesis of high-density dual-protected M@NC for long-life rechargeable zinc-air batteries with record power density. *J. Am. Chem. Soc.* **142**, 7116-7127 (2020).

Comment 4 The author used H_3PO_4 to block the pyridinic-N, thus to conclude that pyridinic-N site is the active site. However, the introduction of H_3PO_4 will decrease the surface Fe ions by the formation of FePO_4 . This should also decrease the activity.

Answer: Thank you very much for this constructive comment. The authors agreed with the reviewer that H_3PO_4 could react with Fe ions to form FePO_4 precipitates. While in this work, by combining the ICP-OES, XPS and TEM results as well as the catalytic performance, we found that the Fe nanoparticles were mainly encased in the carbon layer, and the contribution of surface Fe ions in Fenton reaction was negligible. To

further support the above speculation, we employed SCN^- ions as the poisoning agent due to the fact that SCN^- could strongly complex with the surface Fe species (*ACS Catal.* **12**, 14954–14963, 2022)¹. As shown in **Fig. R15**, we found that after the introduction of SCN^- ions, the removal rate of SMX was hardly affected, which demonstrated that Fe exposed on the surface played a minor role in H_2O_2 activation. Therefore, we concluded that H_3PO_4 was introduced to mainly block the pyridinic-N moieties and cause the decrease in catalytic activity of Fe@N-C, rather than interact with Fe ions. Thanks again!

Fig. R15 The SMX degradation performance of the Fe@N-C-800 before and after the addition of 10 mM KSCN. Condition: $[\text{Fe@N-C-800}] = 50 \text{ mg L}^{-1}$, $[\text{H}_2\text{O}_2]_0 = 1 \times 10^{-3} \text{ M}$, $[\text{SMX}]_0 = 10 \times 10^{-6} \text{ M}$, $[\text{KSCN}] = 10 \times 10^{-3} \text{ M}$, initial pH = 6.2, T = 25 °C.

References

1. Cui, J. et al. Regulating the metal-support interaction: Double jump to reach the efficiency apex of the Fe-N₄-catalyzed Fenton-like reaction. *ACS Catal.* **12**, 14954–14963 (2022).

We believe that we have addressed the reviewers' comments, and will be glad to answer any further questions you might have. We tried our best to improve the manuscript and made some changes in the revised manuscript. These changes will not influence the content and framework of the paper. And here we did not list the changes

but marked in red in revised paper.

We appreciate for Editors/Reviewers' warm work earnestly, and hope that the correction will meet with approval.

Once again, thank you very much for your comments and suggestions.

Sincerely yours,

Mingyang Xing

REVIEWER COMMENTS

Reviewer #3 (Remarks to the Author):

The authors responded some of concerns from this reviewer in this version. Unfortunately, there are still too many inconsistent data and arguable claims in the manuscript.

1) About the novelty or new insights into the field.

As mentioned in the previous comments, the reviewer does not think the reported synthetic method presents the advances compared with tremendous strategies reported in the literatures in either processing, controllability, catalyst uniformity, or scalability. The reviewer does not see what 'scientific or engineering challenges' in synthesis of Fe-N-C materials, which the published method did not/cannot achieve, were solved by the present method.

For the scalable synthesis of high loading Fe-N-C nanoparticles-based materials with a simple processing (which is actually never a big challenge), quite a lot of methods such as one-step template-free, molten salt assisted/polymer-assisted pyrolysis strategies, have been well developed. The scalable synthesis of high loading single-atomic Fe-N-C materials might be still a challenge, but it is, unfortunately, NOT the topic of this manuscript. Both references the authors mentioned in the response letter (Nat. Nanotechnol. 17, 174-181, 2022; Nat. Nanotechnol. 17, 403-407, 2022) are talking about single atomic materials. The other two mentioned references are also focused on single-atomic catalysts. The scalable synthesis of high-loading small-size well-protected Fe-N-C nanoparticle-based materials is also of interest in the field. However, the data shown in the manuscript did not support that the authors achieved such materials (as detailed below). Moreover, the polymer tethering strategy reported in the manuscript has been widely reported to prepare M-N-C catalysts for other electrocatalytic applications such as oxygen reduction reaction.

Here are some of related publications similar to this manuscript:

- a) Wang et al (Environ. Sci. Technol. 55, 1260–1269 (2021)) reported a simple melting strategy to prepare a series of FeNC@C catalysts with a particle size of ~15 nm and mass loading of 28 wt% exhibits excellent electro-Fenton reactivity, the catalyst characteristic is similar to this manuscript.
- b) A similar Fe nanoparticle catalyst was reported by Wu et al. (Appl. Catal., B, 286: 119940 (2021)) with simple mechanical chemical grinding methods and showed excellent electro-Fenton reactivity.
- c) Zhao et al. (ACS EST Eng. 3, 36–44(2023)) reported a citric acid and melamine assistant method similar to that reported in the manuscript to achieve a Fe-N_x catalyst with mass loading of 4.8 wt% showed excellent electro-Fenton reactivity.

Besides, high-loading Fe-based single-atom catalysts have also been reported through various simple strategies, for example, the precursor-dilution strategy reported in Nat Commun 10, 3663 (2019), the ligand-mediated method reported in Nat Commun 10, 4585 (2019), coordination–condensation strategy reported in the Small 18, 2107799 (2022), the pyrolyzing coordinated polymer strategy reported in Adv. Mater. 32, 2000896 (2020). The supermolecule method reported in the Angew. Chem. Int. Ed. 60, 21751–21755 (2021), the multilayer stabilization strategy reported in the Nat Commun 11, 5892 (2020), and the scalable wet-chemistry method reported in Nat. Nanotechnol. 17, 174–181 (2022). The single atomic Fe content in the catalysts prepared through those strategies can be up to 15 wt%.

The reviewer presents these facts here and leave the editor to make the judgement on this aspect.

2) Inconsistent data/concerns remained

- a) The authors presented the size distribution of nanoparticles in Figure 1e and claimed the average size is 4 nm according to TEM images. However, SEM image (Figure 1b) clearly showed a lot of much larger particles in at least tens of nanometers. What makes such inconsistency?

The authors provided a single-crystalline SAED pattern in the previous version, which obviously cannot

be obtained from a nanoparticle in a couple of nanometers. They responded that “the SAED data collected from other areas was removed in the revised manuscript” .

About XRD analysis, the authors discussed: The sharp diffraction peaks in the XRD patterns might be ascribed to the following reasons: First, the two-dimensional nature and magnetism of the Fe@N-C-X might form the preferred orientation during XRD test, which could cause the unusual increase of the peak intensity. This discussion is misleading. The authors can simply use Scherrer equation to calculate the particle size based on the XRD peak in Supplementary Fig. 11 to see the inconsistency. TEM observation is microscopic analysis while XRD analysis reflect bulk conditions. Moreover, the peak intensity does not reflect the particle size while the peak width at half-peak height does.

b) The authors claimed that the Fe loading is 30.5 wt% by measured by ICP-OES while the XPS data shows the Fe content is 2.3 wt%. This means the majority of Fe species are buried deep in the carbon. However, only the Fe species on the surface or sub-surface could directly participate in the catalytic process or affect the surface C or N species to participate in the reaction. Given the XPS could detect the species in a couple of nanometer depth at least, the Fe species which cannot be detected will be not expected to involve or affect the surface active sites. If this the case, the overwhelming majority of Fe species is ineffective in the catalyst, which makes the claim of high metal loading meaningless.

c) From the TEM images, Fig. R7 or Fig. 1f, it cannot say that the nanosize Fe particles are encapsulated or protected by carbon layers. The authors have removed the description about core-shell structure. Although they still claimed the particles are encapsulated by carbon, it seems no direct evidences to support it. If these 4 nm metallic nanoparticles are not well protected by defect-free or defect-less carbon layers, they will be inevitably oxidized or etched away during the electrocatalytic process given such a small size. The presented data such as ^{57}Fe Mössbauer spectra support this fact. The authors also discussed “the relative proportion of Fe(III) increased tremendously (from 12.3% to 27.1%) while that of α -Fe markedly decreased (from 64.0% to 51.3%)” before and after Fenton reaction.

In this case, all the analyses on the active sites and calculations for understanding active sites are based on the metallic Fe species. If the Fe species changed during the reaction, these analyses may be groundless. And, how justify the stability the authors claimed if the active sites changed?

d) The authors should measure the Fe concentration after the reaction and durability test to see how many Fe species dissolved.

e) The main purpose of the electrochemical Fenton reaction is to remove organic pollutants for wastewater purification. If during the processing, Fe dissolution caused the Fe contamination in water, the reviewer did not see the advances of such catalysts.

f) For the active site identification: the authors claimed the metallic Fe assisted pyridinic-N are responsible for the Fenton reaction. However, the catalysts is very complicated and composed of multiple potentials sites such as Fe-N_x-C species, Fe nanoparticles, Fe assisted N species in other form, Fe assisted C species, how to exclude the contribution from these species? Moreover, the authors also discussed “After reaction, the pyridinic-N content showed a marked decrease (40.4% to 29.7%), while the pyrrolic-N content increased significantly (37.1% to 46.0%)”. If the pyridinic-N species are the active sites, what makes the authors to claim the catalysts showed good stability?

g) The authors claimed one of the merits of their catalysts/method is the 2D nanosheet structure which can provide abundant accessible active sites and facilitate the mass transport and electron transfer, ensuring high catalytic activity, compared with spherical morphology. The reviewer does not think it is correct or supported by the data. 2D materials are usually prone to aggregate which will cause the mass transfer issues and give less surface area for the reaction involving solid-liquid phase. This is can be supported by the fact that the highest BET surface area of the present catalyst is only 194.42 m² g⁻¹ which is much less those reported in the literatures. The reviewer does think the reported catalysts hold the advantages over those in the literatures.

h) To be honest, there are many arbitrary claims and discussion without evidences or logic in the manuscript, which should be carefully checked. Here are some of examples:

-- Such a 2D sheet-like hybrid structure not only effectively protects Fe NPs from surface passivation....

-- Fe NPs had a stabilization effect on pyridinic-N moieties during high temperature calcination....

-- the carbon atoms next to pyridinic-N could react with $\bullet\text{OH}$ species, inducing the transformation of pyridinic-N to pyridonic-N....

-- a significant inhibition effect was observed in the Fe@N-C-800/H₂O₂ system (Supplementary Fig. 14), indicating the crucial role of pyridinic-N site in Fenton reaction....

-- the successive electron donation by the Fe center to carbon layer for the continuous activation of H₂O₂,...

-- As shown in Fig. R3, the Fe-PDA precursor presented the disc structure with the in-plane size from 50 to 200 nm, verifying the self-assembled layered structure of PDA.

(TEM images cannot tell whether it is the disc structure)

.....

Reviewer #5 (Remarks to the Author):

This revision is satisfying, thus I recommend it can be accepted now.

A Response to the Comments of the Editor and Reviewers

Manuscript ID: NCOMMS-22-34544-T

Title: "Universal Polymer Tethering Strategy Toward High-Metal-Loading Nanocatalyst for Dual High-Efficiency Fenton Reaction"

We sincerely appreciate the editor's timely response and patience concerning our manuscript, and we are truly grateful for the time and effort reviewers 3 has spent in reviewing our manuscript. The comments provide an important guideline to our following work of synthesizing high-loading single-atomic M-N-C materials, which can further enrich the content of our proposed polymer tethering strategy. According to the comments, we revised the manuscript carefully and the point-to-point responses to comments are shown in the following. The revised parts have been marked in red in the manuscript.

Respective answer to the detailed comments:

Reviewer: 3

The authors responded some of concerns from this reviewer in this version. Unfortunately, there are still too many inconsistent data and arguable claims in the manuscript.

Comment 1 About the novelty or new insights into the field.

As mentioned in the previous comments, the reviewer does not think the reported synthetic method presents the advances compared with tremendous strategies reported in the literatures in either processing, controllability, catalyst uniformity, or scalability. The reviewer does not see what 'scientific or engineering challenges' in synthesis of Fe-N-C materials, which the published method did not/cannot achieve, were solved by the present method.

For the scalable synthesis of high loading Fe-N-C nanoparticles-based materials with a simple processing (which is actually never a big challenge), quite a lot of methods such as one-step template-free, molten salt assisted/polymer-assisted pyrolysis strategies, have been well developed. The scalable synthesis of high loading single-atomic Fe-N-C materials might be still a challenge, but it is, unfortunately, NOT the topic of this manuscript. Both references the authors mentioned in the response letter (Nat. Nanotechnol. 17, 174-181, 2022; Nat. Nanotechnol. 17, 403-407, 2022) are talking about single atomic materials. The other two mentioned references are also focused on single-atomic catalysts. The scalable synthesis of high-loading small-size well-protected Fe-N-C nanoparticle-based materials is also of interest in the field. However, the data shown in the manuscript did not support that the authors achieved such materials (as detailed below).

Answer: Thanks for the comment. The authors have carefully read the references the reviewer listed below and find these works are very helpful for us to make the manuscript more comprehensive. Compared with the reported works, the authors would like to elaborate the novelty of this work through the following three aspects: **i) The innovation of concept.** The principle of preparing M-N-C with both high dispersion and high metal loading is to design a rational precursor that can anchor large amounts of metal ions and prevent their migration under high temperature calcination. However, the commonly used precursors with either limited anchoring sites (e.g. existing carbon supports) or insufficient confinement (e.g. small organic molecules) can hardly achieve such materials (Nat. Chem. 13, 887-894, 2021). To address these issues, polymers with abundant functional groups have emerged as a promising candidate to stabilize metal species, while the direct impregnation of metal ions into the polymers is difficult due to the strong intra and intermolecular interaction between polymers. Therefore, it has to anchor the metal ions by small organic complexes first, and then trigger the polymerization process for further stabilization. To avoid the tedious two-stage synthesis process and additional energy/reagent input for subsequent polymerization, we conceive a self-polymerization confinement strategy by combining the complexing and polymerization process in one-step grind process, which shows great promise for

efficient M-N-C production. **ii) The advancement of method.** Under this original scheme, we adopt the $\text{FeCl}_3 \cdot 6\text{H}_2\text{O}$ and dopamine (DA) as the proof-of-concept system, in which DA with catechol groups can anchor large numbers of metal ions, and in turn, the Fe^{III} with strong oxidizing capability can trigger the fast polymerization of DA to form strong covalent bonds, which provides a stable coordinate environment to suppress the metal aggregation under high temperature calcination, thus guaranteeing the formation of well-dispersed metal nanoparticles. To the best of our knowledge, similar synthesis strategy with such comprehensive properties, including efficient processing (require only 5 min to prepare the metal-polymer precursor), eco-friendly economy (without additional solvent or drying process), generality (a series of M-N-C materials have been prepared) and scalability (the simple grind process can be readily scaled up), has not been reported before this study, despite the abundant works in the development of high-performance M-N-C catalysts. **iii) The breakthrough of performance.** The authors also find that the obtained Fe@N-C catalyst exhibit an unrepresented excellent performance in Fenton reaction through the continuous generation of surface-bound $\cdot\text{OH}$ on the catalyst. The generated surface-bound $\cdot\text{OH}$ with an increased lifetime can significantly decrease the risk of self-quenching side reactions, thus overcoming the trade-off between reaction rate and H_2O_2 utilization efficiency. To the best of our knowledge, there are no reports that have simultaneously achieved such a high reaction rate and a H_2O_2 utilization efficiency before. Therefore, we anticipate that the continuous generation of surface-bound $\cdot\text{OH}$ reported in this work can be one of the most effective solutions to the dilemma of present dual high-efficiency Fenton systems, which provides valuable guidance for the research on Fenton-based systems.

Moreover, as the reviewer said, the scalable synthesis of high-loading small-size well-protected Fe-N-C nanoparticle-based materials and high-loading single-atomic Fe-N-C materials are of interest in many fields. In this manuscript, a series of high-loading small-size M-N-C nanoparticle-based materials have already been prepared though the polymer tethering strategy, and this versatile strategy can be simply modified to synthesize the high-loading single-atomic Fe-N-C materials as well. For example,

we tried to modify the method by introducing additional dicyandiamide into the metal-polymer precursor at a mass ratio of 1:1 during the grinding process, which could construct more N coordination sites to trap metal for preparing high-loading single-atomic Fe-N-C materials. The corresponding XRD result (**Fig. R1a**) revealed that no diffraction peaks of Fe phase were observed, preliminarily demonstrating the absence of Fe nanoparticles and possible existence of single-atomic Fe. This conclusion was also confirmed by the TEM and SAED results (**Fig. R1b**), where no Fe nanoparticles were found in the carbon matrix. Moreover, the highly-dispersed single-atomic Fe could be clearly observed by the HAADF-STEM analysis in **Fig. R1c-d**, where the isolated bright Fe dots were marked by yellow circles. In addition, the overall Fe loading was determined to be 23.3 wt% by ICP-OES analysis, further supporting the existence of high-loading single-atomic Fe. These exciting results will be explored in detail in our following work. Based on the aforementioned results, we conclude that the high-loading single-atomic Fe-N-C materials can also be facilely prepared by our modified polymer tethering strategy, which further confirm the universal and advance of our work in fabricating high-loading small-size or even single-atomic Fe-N-C materials.

Fig. R1 **a** XRD pattern of single-atomic Fe-N-C material. **b** TEM image of single-atomic Fe-N-C material (the inset is the corresponding SAED pattern). **c-d** HAADF-STEM images of single-atomic Fe-N-C material.

Moreover, the polymer tethering strategy reported in the manuscript has been widely reported to prepare M-N-C catalysts for other electrocatalytic applications such as oxygen reduction reaction. Here are some of related publications similar to this manuscript:

a) Wang et al (Environ. Sci. Technol. 55, 1260 – 1269 (2021)) reported a simple melting strategy to prepare a series of FeNC@C catalysts with a particle size of ~15 nm and mass loading of 28 wt% exhibits excellent electro-Fenton reactivity, the catalyst characteristic is similar to this manuscript.

b) A similar Fe nanoparticle catalyst was reported by Wu et al. (Appl. Catal., B, 286: 119940 (2021)) with simple mechanical chemical grinding methods and showed excellent electro-Fenton reactivity.

c) Zhao et al. (ACS EST Eng. 3, 36 – 44(2023)) reported a citric acid and melamine assistant method similar to that reported in the manuscript to achieve a Fe-N_x catalyst with mass loading of 4.8 wt% showed excellent electro-Fenton reactivity.

Answer: Thanks for the comment. Indeed, the relevant works you mentioned are very helpful for us to understand the synthesis strategy of M-N-C more comprehensively, and these references have been added in the revised manuscript. It should be noted that the mentioned references adopt the small organic molecules (e.g. melamine, citric acid, glucose) as the precursors, while the polymerization processes have not been observed or addressed in these systems before calcination. Therefore, the concept of synthesizing M-N-C catalyst in these articles are different from the polymer tethering strategy reported in our work, which also leads to totally different performance of the catalyst in H₂O₂ activation. Thanks again!

Besides, high-loading Fe-based single-atom catalysts have also been reported through various simple strategies, for example, the precursor-dilution strategy reported

in Nat Commun 10, 3663 (2019), the ligand-mediated method reported in Nat Commun 10, 4585 (2019), coordination - condensation strategy reported in the Small 18, 2107799 (2022), the pyrolyzing coordinated polymer strategy reported in Adv. Mater. 32, 2000896 (2020). The supermolecule method reported in the Angew. Chem. Int. Ed. 60, 21751 - 21755 (2021), the multilayer stabilization strategy reported in the Nat Commun 11, 5892 (2020), and the scalable wet-chemistry method reported in Nat. Nanotechnol. 17, 174 - 181 (2022). The single atomic Fe content in the catalysts prepared through those strategies can be up to 15 wt%.

Answer: Thanks for the comment. In fact, most of the synthetic methods for single-atomic M-N-C catalysts listed by the reviewer have been cited and compared with our strategy in the first-round response. For instance, the precursor-dilution strategy reported in Nat. Commun. 10, 3663 (2019) contained a polymerization reaction in solvent for 24 h, followed by purification and drying for 48 h before calcination, and the final metal loading was 0.48 wt%; The ligand-mediated method reported in Nat. Commun. 10, 4585 (2019) required the dispersion of precursor in ethanol for 4 h, and then removed the solvent through drying for 8 h, the highest metal loading was 5.3 wt%; The coordination-condensation strategy reported in Small 18, 2107799 (2022) involved a long procedure, where a silica template should be prepared firstly and then mixed with metal and carbon precursors for 16 h, the highest metal loading was 4.2 wt%; The pyrolyzing coordinated polymer strategy reported in Adv. Mater. 32, 2000896 (2020) required the dispersion of metal/organic precursors in water and the water volatilization treatment for 12 h before calcination, the final highest metal loading was 30.0 wt%; The supermolecule method reported in Angew. Chem. Int. Ed. 60, 21751-21755 (2021) needed the dispersion of metal and organic precursors in water, and then collected the mixture through centrifugation and dried at 60 °C overnight before calcination, the highest metal loading was 11.2 wt%; The multilayer stabilization strategy reported in Nat Commun 11, 5892 (2020) included mixing, hydrothermal, freeze-drying and calcination treatment, the highest metal loading was 15.7 wt%; The scalable wet-chemistry method reported in Nat. Nanotechnol. 17, 174-181 (2022) required the pre-preparation of carbon support and the subsequent mixing and rotary evaporation

processes before a two-step annealing treatment, the highest metal loading was 23 wt%. Comparably, our modified polymer tethering strategy could facilely synthesize the single-atomic Fe-N-C catalyst with metal content of 23.3 wt%, demonstrating the significant advances in terms of production efficiency and metal loading, which had never been simultaneously achieved in the previous reports.

Based on the above results and analyses, we hope we can convince the reviewer that our strategy on the fabrication of high-metal-loading Fe-N-C catalysts for achieving dual high-efficiency performance in Fenton reaction is both conceptually novel and practically valuable, which may also open new avenues for the efficient production of high-performance M-N-C catalysts for other applications. Thanks again!

Comment 2 Inconsistent data/concerns remained.

a) The authors presented the size distribution of nanoparticles in Figure 1e and claimed the average size is 4 nm according to TEM images. However, SEM image (Figure 1b) clearly showed a lot of much larger particles in at least tens of nanometers. What makes such inconsistency?

Answer: Thanks for pointing out this issue. In fact, the distributed particles on the carbon nanosheets could also be found in N-C-800 materials without the existence of Fe species in our work (**Fig. R2a**). This phenomenon was also very common in previously reported PDA derived carbon support-based catalysts, for example, there were also similar larger particles exposed on the surface of single-atom-dimer NiCo-N-C catalyst (**Fig. R2b**, *Nat. Commun.* **12**, 6766, 2021). Therefore, it could be speculated that the bulges on the support surface were approximately the carbon particles rather than the metal particles. Consequently, there was no inconsistency existed in our SEM and TEM images. Thanks again!

(*Nat. Commun.* 12, 6766, 2021)

Fig. R2 **a** SEM image of N-C-800. **b** SEM image of NiCo-SAD-NC.

The authors provided a single-crystalline SAED pattern in the previous version, which obviously cannot be obtained from a nanoparticle in a couple of nanometers. They responded that “the SAED data collected from other areas was removed in the revised manuscript” .

Answer: Thanks for the comment. As for the SAED data, not only the nanoparticle size, but also the orientation of nanoparticle on the support accounts for the diffraction pattern in SAED test. Both the nanoparticle size and the nanoparticle orientation can make a difference on the SAED diffraction pattern. As shown in **Fig. R3**, when the nanoparticles are epitaxially aligned on the support, the SAED pattern reveals the diffraction spots of the crystal planes, while the diffraction rings will appear in the SAED pattern when the nanoparticles exhibit a random alignment on the support (*Nat. Mater.* 21, 1042-1049, 2022). Therefore, it is possible to obtain the diffraction spots of SAED pattern from the metal supported materials with small particle size.

(*Nat. Mater.* **21**, 1042-1049, 2022)

Fig. R3 **a** TEM image and **b** SAED pattern of a typical epitaxial CsPbBr₃/PEA₂PbBr₄ heterostructure formed by oriented assembly. **c** TEM image and **d** SAED pattern of a typical CsPbBr₃/PEA₂PbBr₄ heterostructure formed by random assembly.

About XRD analysis, the authors discussed: The sharp diffraction peaks in the XRD patterns might be ascribed to the following reasons: First, the two-dimensional nature and magnetism of the Fe@N-C-X might form the preferred orientation during XRD test, which could cause the unusual increase of the peak intensity. This discussion is misleading. The authors can simply use Scherrer equation to calculate the particle size based on the XRD peak in Supplementary Fig. 11 to see the inconsistency. TEM observation is microscopic analysis while XRD analysis reflect bulk conditions. Moreover, the peak intensity does not reflect the particle size while the peak width at half-peak height does.

Answer: We thank the reviewer for pointing out this important issue. Commonly, the samples for microscopic analysis are collected from the suspension after dispersing the pristine samples in solvent. To clarify the reviewer mentioned inconsistency between the microscopic images and XRD results in the manuscript, we performed the XRD test for the catalyst collected from the suspension without the sediment. As shown in **Fig.**

R4, the diffraction peaks of Fe nanoparticles were greatly weakened in the catalyst collected from the suspension compared with that of the pristine Fe@N-C-800 sample. The width at half maximum height of peaks located at 44.6° was 1.23° , thus the average size of Fe nanoparticles was calculated to be 6.76 nm via the Scherrer equation, which was comparable with the result obtained from the TEM images.

In addition, the N_2 adsorption-desorption test of Fe@N-C-800 after acid washing demonstrated that the Fe nanoparticles were centered at 4-5 nm on the support (see details in **Comment 2g, Fig. R9**), which was also consistent with the TEM images. The above results confirmed that even though there were a few of large metal nanoparticles contributing to the sharp diffraction peak in XRD patterns, the part of large metal nanoparticles could hardly affect the statistics of size distribution of the Fe@N-C-800 in this work. Moreover, as mentioned in **Comment 1 (Fig. R1)**, the control of metal nanoparticle size (even to single-atomic size) is never a challenge in our proposed strategy, and the authors just select the most feasible and concise approach to achieve the optimal catalytic performance of the Fe@N-C catalyst with both high reaction rate and H_2O_2 utilization efficiency, despite the existence of few large metal nanoparticles. Thanks again!

Fig. R4 XRD pattern of Fe@N-C-800 collected from the suspension.

b) The authors claimed that the Fe loading is 30.5 wt% by measured by ICP-OES while the XPS data shows the Fe content is 2.3 wt%. This means the majority of Fe species are buried deep in the carbon. However, only the Fe species on the surface or sub-

surface could directly participate in the catalytic process or affect the surface C or N species to participate in the reaction. Given the XPS could detect the species in a couple of nanometer depth at least, the Fe species which cannot be detected will be not expected to involve or affect the surface active sites. If this the case, the overwhelming majority of Fe species is ineffective in the catalyst, which makes the claim of high metal loading meaningless.

Answer: Thanks for pointing out this interesting issue. Generally, the XPS spectroscopy is surface sensitive and its characteristic photoelectrons escaped from surface follow the Beer-Lambert law ($I_d = I_0 \times e^{-d/\lambda}$). This means that about 65% of the XPS signal is obtained from the sample with the depth less than λ (the inelastic mean free path of the escaped electrons, which is ranged from 0.5 ~ 2 nm according to certain materials), and the XPS signal will be further exponentially decreased with the increase of detection depth. In this case, the Fe nanoparticles well protected by carbon layers will show much lower signal than that of carbon surface. In other words, the actual Fe content in the range of penetration depth of XPS should be much larger than the detected signal when the Fe nanoparticles are coated by carbon layers. Therefore, the different Fe contents obtained by XPS and ICP-OES test just further confirmed the well protected Fe nanoparticles by carbon layers (*Nano Res.* **12**, 2341–2347, 2019), but could not draw a conclusion that the majority of Fe species were buried deep in the carbon by simply subtracting the Fe content detected by XPS from the value obtained by ICP-OES analysis. Moreover, we have provided the solid evidence in the supporting information to confirm the important role of metal loading, which is also in agreement with the previous metal supported catalysts with encapsulated structure (*J. Am. Chem. Soc.* **142**, 7116–7127, 2020). As shown in **Fig. R5**, the pollutant removal capability of the catalysts was significantly improved with the increase of FeCl₃•6H₂O/DA ratio (from 1:3 to 2:1), demonstrating the pivotal role of high metal loading in Fenton reaction.

Fig. R5 **a** Contaminant removal performance of Fe@N-C-800 prepared at different ratios of FeCl₃·6H₂O and DA. **b** Leached iron ions of Fe@N-C-800 prepared at different ratios of FeCl₃·6H₂O and DA. Condition: [Catalyst] = 50 mg L⁻¹, [H₂O₂]₀ = 1 × 10⁻³ M, [SMX]₀ = 10 × 10⁻⁶ M, initial pH = 6.2, T = 25 °C.

c) From the TEM images, Fig. R7 or Fig. 1f, it cannot say that the nanosize Fe particles are encapsulated or protected by carbon layers. The authors have removed the description about core-shell structure. Although they still claimed the particles are encapsulated by carbon, it seems no direct evidences to support it. If these 4 nm metallic nanoparticles are not well protected by defect-free or defect-less carbon layers, they will be inevitably oxidized or etched away during the electrocatalytic process given such a small size. The presented data such as ⁵⁷Fe Mössbauer spectra support this fact. The authors also discussed “the relative proportion of Fe(III) increased tremendously (from 12.3% to 27.1%) while that of α-Fe markedly decreased (from 64.0% to 51.3%)” before and after Fenton reaction.

In this case, all the analyses on the active sites and calculations for understanding active sites are based on the metallic Fe species. If the Fe species changed during the reaction, these analyses may be groundless. And, how justify the stability the authors claimed if the active sites changed?

Answer: Thanks for the comment. According to reviewer’s suggestion, additional HRTEM test was conducted to reveal the encapsulated structure of Fe nanoparticles on the carbon support. As shown in **Fig. R6a**, the dispersed Fe nanoparticles were covered

by brighter carbon layers, which could be evidenced by the lattice fringes of 0.335 nm corresponding to the (002) crystal plane of graphitic carbon. Moreover, the well-protected Fe nanoparticles with the interplanar spacing of 0.210 nm could be clearly observed at the edge of carbon nanosheet (**Fig. R6b**), confirming that the Fe nanoparticles were coated by thin carbon layers.

Besides, the comment that the analyses on the active site identification is groundless due to the change of Fe species after reaction needs further consideration. In fact, the density functional theory (DFT) calculations are always conducted based on the simulated structures that can mostly represent the real materials. Therefore, the initial structure of the catalyst before reaction is often adopted to explore the active sites that are more likely to trigger the reaction (*Nat. Commun.* **12**, 303, 2021, *Proc. Natl. Acad. Sci.* **119**, e2119492119, 2022, *J. Am. Chem. Soc.* **140**, 12469-12475, 2018, *Angew. Chem. Int. Ed.* **61**, e202200670, 2022, *Angew. Chem. Int. Ed.* **61**, e202200406, 2022, *Adv. Funct. Mater.* **31**, 2007877, 2021, *Nano. Lett.* **20**, 6807-6814, 2020, *ACS Catal.* **10**, 14857-14870, 2020). Many crucial information, such as the adsorption behavior, electron transfer and reaction pathway of H₂O₂ on the catalyst surface, can be obtained from the theoretical calculation for predicting catalytic performance and explicating the potential catalytic mechanism. Combined these facts with the detailed characterizations of the used catalyst, we can finally give a clear scenario of how the catalyst works during reaction. Therefore, the change of Fe species after reaction can further help us to understand the reaction mechanism and find out the active sites participated in Fenton reaction. Moreover, after the electron donation from Fe surface to the carbon layers, the Fe species inside the Fe nanoparticles can still transfer its electron to the heterointerface through Fe-Fe and Fe-O-Fe bonds, retrieving the surface activity and providing the dynamics for the sustained catalytic reaction (*Nat. Commun.* **13**, 5365, 2022, *Angew. Chem. Int. Ed.* **60**, 17115-17122, 2021). Therefore, the theoretical calculations based on metallic Fe species in this manuscript is reasonable and reliable.

In addition, the authors also understand the reviewer's concern that the change of catalyst composition might affect the H₂O₂ activation capability and the accuracy of the proposed model in DFT calculations. It should be noted that the work described in this

manuscript is the Fenton-like system composed with only the catalyst and H₂O₂, not the electron-Fenton or photo-Fenton systems, so the required electrons for H₂O₂ activation is mainly from the metal species in catalyst. As a result, the increased content of oxidized metal composition after reaction is commonly obtained in these similar systems (*Nat. Commun.* **13**, 5365, 2022, *Angew. Chem. Int. Ed.* **58**, 8134-8138, 2019, *ACS Nano* **10**, 11532-11540, 2016, *Adv. Funct. Mater.* **31**, 2106311, 2021, *Environ. Sci. Technol.* **53**, 9081-9090, 2019). In this case, the catalyst composition is constantly changing during reaction, so the pristine structure of the catalyst rather than the varied structure during reaction, is selected as the research model to predict the active sites. Besides, it also makes sense to construct other DTF models based on the detailed characterizations of the used catalysts to expound the variation of catalytic performance toward H₂O₂, while it is not the focus of this manuscript.

As for the stability of the catalyst in this work, it can be verified through the storage test, cyclic degradation, suppressed Fe leaching and well-maintained structure after reaction, which has already been well established in Fenton-like catalysis fields (*Angew. Chem. Int. Ed.* **61**, e202200670, 2022, *Angew. Chem. Int. Ed.* **61**, e202207268, 2022, *Angew. Chem. Int. Ed.* **61**, e202202338, 2022, *Environ. Sci. Technol.* **53**, 9081-9090, 2019). All the above properties are based on the well-protected Fe nanoparticle structure in the obtained catalyst. Thanks again!

Fig. R6 HRTEM images of Fe@N-C-800.

d) The authors should measure the Fe concentration after the reaction and durability

test to see how many Fe species dissolved.

Answer: Thanks for the comment. The related data of leached Fe ions after reaction have been provided in the manuscript and supporting information, where only 0.076 mg L⁻¹ of Fe ions was leached by ICP-OES analysis. Based on the high metal loading of the prepared catalyst (30.5 wt%), the percentage of the total leached Fe ions only accounted for less than 0.5% of Fe in the catalyst, furthering confirming the well protected Fe nanoparticles by carbon layers and the stability of the catalyst during Fenton reaction.

e) The main purpose of the electrochemical Fenton reaction is to remove organic pollutants for wastewater purification. If during the processing, Fe dissolution caused the Fe contamination in water, the reviewer did not see the advances of such catalysts.

Answer: Thanks for the comment. The authors should first clarify the issue that the work described in this manuscript is the Fenton-like system composed with only the catalyst and H₂O₂, not the electric-Fenton system. Besides, the concentration of leached Fe ions during reaction was only 0.076 mg L⁻¹, which was far below the EU standards of 2.0 mg L⁻¹ and met the basic emission standard for wastewater treatment. More importantly, the obtained catalyst exhibited the dual high-efficiency performance in Fenton reaction with both impressive catalytic activity and H₂O₂ utilization efficiency by using sulfamethoxazole as the probe, which had not yet been achieved simultaneously. These issues have already been clearly described in the manuscript and the past two round responses, which further confirm the advances of our catalyst compared with other Fenton-like catalyst reported before. Thanks again!

f) For the active site identification: the authors claimed the metallic Fe assisted pyridinic-N are responsible for the Fenton reaction. However, the catalysts is very complicated and composed of multiple potentials sites such as Fe-N_x-C species, Fe nanoparticles, Fe assisted N species in other form, Fe assisted C species, how to exclude the contribution from these species? Moreover, the authors also discussed “After reaction, the pyridinic-N content showed a marked decrease (40.4% to 29.7%), while

the pyrrolic-N content increased significantly (37.1% to 46.0%)” . If the pyridinic-N species are the active sites, what makes the authors to claim the catalysts showed good stability?

Answer: Thanks for the comment. The authors have carefully explored the active sites on the catalyst through both the detailed experimental results and density functional theory analysis. First, we employed SCN^- ions as the poisoning agent due to that the SCN^- could strongly complex with the surface Fe species (*ACS Catal.* **12**, 14954–14963, 2022). As shown in **Fig. R7**, we found that after the introduction of SCN^- ions, the removal rate of SMX was hardly affected, which demonstrated that the surface Fe species were minimal and played a negligible role in H_2O_2 activation. This result was also consistent with our conclusion that the Fe nanoparticles were well protected by the carbon layers. Second, we have performed the density functional theory calculations to identify the active sites on carbon surface, where the adsorption behaviors of H_2O_2 on the different N configurations and their adjacent C atoms were all explored. Based on the adsorption energy, electron transfer and reaction pathway analyses, we finally confirmed that the Fe nanoparticle assisted pyridinic-N sites were primarily responsible for the excellent performance in Fenton reaction, while the other sites on the carbon surface played the secondary roles. Third, the notable compositional variation of the pyridinic-N species after reaction further confirmed that these active sites were participated in the catalytic process, and the change of surface composition was almost inevitable due to the electron transfer or intermediate adsorption during the typical Fenton-like reaction. Although the surface variation could lead to the relative decrease of the catalytic performance than that of pristine one, the stability of the catalyst could still be consolidated through the storage test, cyclic degradation, low Fe leaching as well as the characterizations of catalyst after reaction in this manuscript, which had commonly been well established in Fenton-like catalysis fields (*Angew. Chem. Int. Ed.* **61**, e202200670, 2022, *Angew. Chem. Int. Ed.* **61**, e202207268, 2022, *Angew. Chem. Int. Ed.* **61**, e202202338, 2022, *Environ. Sci. Technol.* **53**, 9081-9090, 2019).

Fig. R7 The SMX degradation performance of Fe@N-C-800 before and after the addition of KSCN. Condition: [Fe@N-C-800] = 50 mg L⁻¹, [H₂O₂]₀ = 1 × 10⁻³ M, [SMX]₀ = 10 × 10⁻⁶ M, [KSCN] = 10 × 10⁻³ M, initial pH = 6.2, T = 25 °C.

g) The authors claimed one of the merits of their catalysts/method is the 2D nanosheet structure which can provide abundant accessible active sites and facilitate the mass transport and electron transfer, ensuring high catalytic activity, compared with spherical morphology. The reviewer does not think it is correct or supported by the data. 2D materials are usually prone to aggregate which will cause the mass transfer issues and give less surface area for the reaction involving solid-liquid phase. This can be supported by the fact that the highest BET surface area of the present catalyst is only 194.42 m² g⁻¹ which is much less than those reported in the literatures. The reviewer does think the reported catalysts hold the advantages over those in the literatures.

Answer: Thanks for the comment. In fact, the advantages and characteristics of 2D structure materials have been well established in many distinguished works (*Nat. Energy* **1**, 16184, 2016, *Adv. Mater.* **27**, 403-427, 2015). In addition, although the PDA-derived carbon nanospheres in some reports had the BET surface area of 300 ~ 400 m² g⁻¹ (*Angew. Chem. Int. Ed.* **54**, 588-593, 2015, *Adv. Mater.* **30**, 1706508, 2018; *Nat. Commun.* **12**, 6766, 2021), their diameters were only 100 ~ 200 nm, which exhibited severe aggregation as shown in **Fig. R8**. Moreover, a relatively small BET surface area of the Fe@N-C catalyst in this work was resulted from the high metal loading. When the metal nanoparticles were removed through acid washing, the BET surface area of

2D carbon nanosheets increased to $756.3 \text{ m}^2 \text{ g}^{-1}$ (**Fig. R9a**), far beyond the value of the reported carbon nanospheres. These interesting 2D porous carbon nanosheets could also find their widespread applications in oxygen reduction reaction and supercapacitors (*Science* **332**, 1537-1541, 2011, *Angew. Chem. Int. Ed.* **55**, 6858-6863, 2016). Moreover, the pore-size distribution analysis showed that peak located at 5 nm was clearly observed after acid washing (**Fig. R9b**), which were mainly attributed to the holes caused by Fe leaching, further confirming that the average size of Fe nanoparticles was centered at 4-5 nm on the 2D carbon support.

More importantly, just as the reviewer said in **Comment 2b**, the metal species buried deep in carbon substrate could not be involved in the catalytic reaction. This situation was more likely to occur in spherical structure, while most of the metal nanoparticles would locate near the support surface due to the thin thickness of the 2D nanosheets, which further confirmed the obvious advantage of 2D structure in catalytic reaction.

Fig. R8 **a** PDA-derived carbon nanospheres prepared using block polymer PS-*b*-PEO as a template. **b-c** PDA-derived carbon nanospheres prepared through solution polymerization.

Fig. R9 **a** N_2 adsorption-desorption isotherms of Fe@N-C-800 before and after acid washing. **b** Pore size distribution of Fe@N-C-800 before and after acid washing.

h) To be honest, there are many arbitrary claims and discussion without evidences or logic in the manuscript, which should be carefully checked. Here are some of examples:
 -- Such a 2D sheet-like hybrid structure not only effectively protects Fe nanoparticles from surface passivation...

Answer: Thanks for the comment. The Fe nanoparticles were proved to be well protected by carbon layers on 2D support in this work, and the encapsulation of metal crystals into protected shells could certainly elevate the metal stability against passivation and leaching (*Acc. Chem. Res.* **51**, 678–687, 2018, *Angew. Chem. Int. Ed.* **58**, 2-11, 2019, *ACS Catal.* **11**, 7422–7428, 2021). This conclusion had been widely accepted and the relevant references had already been cited in the introduction. Thanks again.

-- Fe nanoparticles had a stabilization effect on pyridinic-N moieties during high temperature calcination...

Answer: Thanks for the comment. The authors had clarified this issue in detail in the second-round response and listed several references to support this claim (*ACS Nano*, **12**, 9441-9450, 2018, *Adv. Energy Mater.* **11**, 2100303, 2021, *Appl. Catal. B-Environ.* **262**, 118302, 2020). The relevant reference had already been cited in the manuscript. Thanks again.

-- the carbon atoms next to pyridinic-N could react with $\cdot\text{OH}$ species, inducing the transformation of pyridinic-N to pyridonic-N...

Answer: Thanks for the comment. This fact was confirmed by the XPS results and well supported by the cited references (*Science* **351**, 361-365, 2016, *Adv. Energy Mater.* **10**, 2002592, 2020), so the authors think these descriptions are appropriate. Thanks again.

-- a significant inhibition effect was observed in the Fe@N-C-800/H₂O₂ system (Supplementary Fig. 14), indicating the crucial role of pyridinic-N site in Fenton reaction...

Answer: Thanks for the comment. The phosphate acid as a block agent for pyridinic-N site had been explored in detail in previous reports (*ACS Catal.* **6**, 7249-7259, 2016, *Angew. Chem. Int. Ed.* **58**, 2-11, 2019). Therefore, the significant inhibition effect in the Fe@N-C-800/H₂O₂ system after the addition of phosphate acid could confirm the important role of pyridinic-N site in Fenton reaction. Thanks again.

-- the successive electron donation by the Fe center to carbon layers for the continuous activation of H₂O₂

Answer: Thanks for the comment. The continuous activation of H₂O₂ to generate $\cdot\text{OH}$ was firmly supported by the EPR analysis, and the required electrons for H₂O₂ activation was supplied by the Fe nanoparticles, which was confirmed by the Mössbauer spectra and DFT analysis. Therefore, the authors think these descriptions are appropriate.

-- As shown in Fig. R3, the Fe-PDA precursor presented the disc structure with the in-plane size from 50 to 200 nm, verifying the self-assembled layered structure of PDA. (TEM images cannot tell whether it is the disc structure)

Answer: Thanks for the comment. From the TEM image in **Fig. R10a** we could preliminarily find the disc structure of the Fe-PDA precursor. According to the suggestion from the reviewer, the authors performed the atomic force microscopy

(AFM) test and the result was shown in **Fig. R10b**. The AFM image showed that the Fe-PDA precursor possessed the in-plane size of 150 nm and thickness of 10 nm, further verifying the disc structure of the Fe-PDA precursor.

Fig. R10 a TEM and **b** AFM images of Fe-PDA precursor.

Overall, the authors appreciate the insightful comments raised by reviewer 3, which can certainly improve the quality of this work. Although some issues may not be the focus of our work, we still try our best to give the point-to-point responses to comments and hope the revised manuscript become appealing to the reviewer. Thanks again!

Reviewer: 5

Comment This revision is satisfying, thus I recommend it can be accepted now.

Answer: We are grateful for the time and effort Review 5 has spent in reviewing our manuscript. We appreciate the reviewer's recommendation very much.

We believe that we have addressed the reviewers' comments. We tried our best to improve the manuscript and made corresponding changes in the manuscript. These changes will not influence the content and framework of the paper. And here we did not list the changes but marked in red in revised paper.

We appreciate for Editors/Reviewers' warm work earnestly, and hope that the correction will meet with approval.

Once again, thank you very much for your comments and suggestions.

Sincerely yours,

Mingyang Xing

REVIEWER COMMENTS

Reviewer #3 (Remarks to the Author):

Although this reviewer still did not see much advances in the synthesis reported in this work compared with other reported ones, it is no need to make more argument on this point. In the response letter, there are still quite a few major concerns which could be made clearer for improving the quality of the manuscript. Especially, the authors arbitrarily changed the inconsistent SAED and XRD data without convincible explanation for why they did.

The reviewer does not benefit from whether this manuscript is published or not, but hope the results could be reported more scientifically.

In the response letter:

- 1) The author claimed that "Comparably, our modified polymer tethering strategy could facilely synthesize the single-atomic Fe-N-C catalyst with metal content of 23.3 wt%", it is quite amazing for the single-atomic Fe-N-C with a metal content of 23.3 wt%. Does the authors means 'single-atomic Fe-N-C' or the one with metal nanoparticles? If the former, any evidences?
- 2) The authors attributed the nanoparticles in the SEM images to the carbon nanoparticles. Does the SEM image show the similar carbon nanoparticles?
- 3) The authors ascribed the provided single-crystalline SAED pattern to the many nanoparticles in the exactly same orientation like the one epitaxially grown on the single-crystalline substrate. The reviewer did not think the pyrolyzed nanoparticles in this manuscript should hold such exactly same orientation on a pyrolyzed carbon substrate.
- 4) For XRD data inconsistency, the authors simply changed the XRD data with new one without explanation for why they change the data. The same situation happens for the SAED pattern.
- 5) The authors did not explain well that why a majority of Fe which cannot be detected by surface intensive XPS can play pivotal role in the catalysts (the XPS data shows the Fe content is 2.3 wt%, while the ICP-OES measured Fe loading is 30.5 wt%). Does it need such a high Fe loading? If not, what is the merit for making such a high Fe loading.
- 6) In the response 2c, the authors provided the HRTEM to confirm that the Fe nanoparticles were coated by thin carbon layers. The authors also confirmed that the metallic Fe changed during the reaction. However, the authors also claimed that the use of the pure metallic Fe to do the simulation is reasonable and reliable. The reviewer did not think it is correct.
- 7) They also claimed that "the required electrons for H₂O₂ activation is mainly from the metal species in catalyst." and "All the above properties are based on the well-protected Fe nanoparticle structure in the obtained catalyst." Are these claims consistent?
- 8) The authors claimed that "only 0.076 mg L⁻¹ of Fe ions was leached by ICP-OES analysis." How does the continuously production of H₂O₂ from metallic Fe species while they did not dissolve?

A Response to the Comments of the Editor and Reviewers

Manuscript ID: NCOMMS-22-34544-C

Title: “Universal Polymer Tethering Strategy Toward High-Metal-Loading Nanocatalyst for Dual High-Efficiency Fenton Reaction”

We are truly grateful for the editor’s timely response and patience concerning our manuscript, and we also sincerely appreciate the scientific and scrupulous attitude of reviewers 3 in reviewing our manuscript. The comments are very valuable and helpful for revising the paper, which can certainly improve the quality of this work. According to the comments, we revised the manuscript carefully and the point-to-point responses to comments are shown in the following. The revised parts have been marked in red in the manuscript.

Respective answer to the detailed comments:

Reviewer #3

Although this reviewer still did not see much advances in the synthesis reported in this work compared with other reported ones, it is no need to make more argument on this point. In the response letter, there are still quite a few major concerns which could be made clearer for improving the quality of the manuscript. Especially, the authors arbitrarily changed the inconsistent SAED and XRD data without convincing explanation for why they did.

The reviewer does not benefit from whether this manuscript is published or not, but hope the results could be reported more scientifically.

Comment 1 The author claimed that “Comparably, our modified polymer tethering strategy could facilely synthesize the single-atomic Fe-N-C catalyst with metal content of 23.3 wt%”, it is quite amazing for the single-atomic Fe-N-C with a metal

content of 23.3 wt%. Does the authors means 'single-atomic Fe-N-C' or the one with metal nanoparticles? If the former, any evidences?

Answer: Thanks for the comment. To further confirm the versatility of our proposed polymer tethering strategy, the single-atomic Fe-N-C material was facilely prepared by introducing additional dicyandiamide into the metal-polymer precursor at a mass ratio of 1:1 during the grinding process, which was followed by the pyrolysis treatment to obtain the target material with the metal content of 23.3 wt%. As shown in **Fig. R1a**, the XRD result revealed that no diffraction peaks of Fe phase were observed, preliminarily demonstrating the absence of Fe nanoparticles and possible existence of single-atomic Fe. This conclusion was also confirmed by the TEM and SAED results (**Fig. R1b**), where no Fe nanoparticles were found in the carbon matrix. Moreover, the highly-dispersed single-atomic Fe could be clearly observed by the HADDF-STEM analysis in **Fig. R1c-d**, where the isolated bright Fe dots were marked by yellow circles. In addition, the overall Fe loading was determined to be 23.3 wt% by ICP-OES analysis, further supporting the existence of high-loading single-atomic Fe. These exciting outcomes will be explored in detail in our following work. Based on the aforementioned results, we conclude that the high-loading single-atomic Fe-N-C materials can also be facilely prepared by our modified polymer tethering strategy, which further confirms the universal and advance of our strategy in fabricating high-loading small-size or even single-atomic Fe-N-C materials. Thanks again!

Fig. R1 **a** XRD pattern of single-atomic Fe-N-C material. **b** TEM image of single-atomic Fe-N-C material (the inset is the corresponding SAED pattern). **c-d** HAADF-STEM images of single-atomic Fe-N-C material.

Comment 2 The authors attributed the nanoparticles in the SEM images to the carbon nanoparticles. Does the SEM image show the similar carbon nanoparticles?

Answer: Thanks for the comment. The authors think the reviewer want to ask whether the TEM images show the similar carbon nanoparticles on the nanosheets. According to the suggestion from the reviewer, the authors performed the TEM test and the result was shown in **Fig. R2**. The TEM images demonstrated that the relatively darker carbon region with the size from 20 to 80 nm could be found on the nanosheets (marked by yellow circles), further verifying the existence of carbon nanoparticles on the catalyst surface. Thanks again!

Fig. R2 TEM images of Fe@N-C-800.

Comment 3 The authors ascribed the provided single-crystalline SAED pattern to the many nanoparticles in the exactly same orientation like the one epitaxially grown on the single-crystalline substrate. The reviewer did not think the pyrolyzed nanoparticles in this manuscript should hold such exactly same orientation on a pyrolyzed carbon substrate.

Answer: Thanks for the comment. The authors fully agree with the reviewer's point that the metal nanoparticles could not exactly share the same orientation on the whole carbon support during pyrolysis process. In fact, the authors have selected several areas to obtain the SAED patterns in the previous tests, where some of the patterns exhibited the diffraction rings and some showed the diffraction spots. Since the nanoparticle size, quantity and orientation can all account for the diffraction pattern in SAED test, it is possible to obtain the diffraction spots of SAED pattern in some areas from the carbon supported metal materials obtained in this work. Considering that the previous diffraction spots of SAED pattern was not obtained from the TEM image in Fig. 1d in the manuscript, we thus removed the SAED result in the revised manuscript to avoid misunderstanding. Thanks again!

Comment 4 For XRD data inconsistency, the authors simply changed the XRD data with new one without explanation for why they change the data. The same situation happens for the SAED pattern.

Answer: Thanks for the comment. The authors should first clarify the point that we

just supplement the new XRD pattern of the catalyst collected from the suspension to illustrate the consistent results with the microscopic images, and the XRD patterns of the pristine materials are all retained in the revised manuscript.

We also thank the reviewer for raising the important issue in the previous response that the XRD patterns of the pristine materials with sharp diffraction peaks were inconsistent with the microscopic results. To clarify this issue, the authors performed the XRD test for the catalyst collected from the suspension due to the fact that the samples for microscopic analysis were also collected from the suspension after dispersing the pristine samples in solvent. The XRD pattern revealed that the diffraction peaks of Fe nanoparticles were greatly weakened in the catalyst collected from the suspension compared with that of the pristine Fe@N-C-800 sample. The width at half maximum height of peaks located at 44.6° was 1.23° , thus the average size of Fe nanoparticles was calculated to be 6.76 nm via the Scherrer equation, which was comparable with the result obtained from the TEM images. The relevant discussion has been added in the supplementary information, thanks again!

Comment 5 The authors did not explain well that why a majority of Fe which cannot be detected by surface intensive XPS can play pivotal role in the catalysts (the XPS data shows the Fe content is 2.3 wt%, while the ICP-OES measured Fe loading is 30.5 wt%). Does it need such a high Fe loading? If not, what is the merit for making such a high Fe loading.

Answer: Thanks for the comment. First, the authors would like to point out that the different Fe contents obtained by XPS and ICP-OES test were attributed to the well protected Fe nanoparticles by thin carbon layers (*Nano Res.* **12**, 2341–2347, 2019) in our catalyst, which was also verified by the HRTEM test. Second, the underlying Fe nanoparticles could donate electrons to the N-doped carbon layer and give rise to denser local electronic states of the carbon surface at the interaction region (*Acc. Chem. Res.* **51**, 678-687, 2018, *Angew. Chem. Int. Ed.* **58**, 8134-8138, 2019), thus favoring the dissociation of H₂O₂ on catalyst surface, which had been verified by the SCN⁻ poison experiment that the Fenton activity should be mainly ascribed to

metallic-nanoparticle-boosted C-N_x sites rather than the surface Fe species (**Fig. R3**). Moreover, after the electron donation from Fe surface to the carbon layers, the Fe species inside the Fe nanoparticles that might not be detected by surface sensitive XPS spectroscopy could also transfer its electron to the heterointerface through Fe-Fe and Fe-O-Fe bonds (*Nat. Commun.* **13**, 5365, 2022, *Angew. Chem. Int. Ed.* **60**, 17115-17122, 2021), thus retrieving the surface activity. Third, since the required electrons for H₂O₂ activation were mainly from the metallic-nanoparticle-boosted C-N_x sites in the catalyst, the high loading of well-dispersed Fe nanoparticles meant the greatly increased active regions on carbon surface and the huge electron reservoir for the continuous activation of H₂O₂, thus guaranteeing the exceptional performance in Fenton-like reaction. Finally, we have provided the solid evidence in the supporting information by preparing different catalysts with varied FeCl₃•6H₂O/DA ratios, where the pollutant removal capability of the catalysts was significantly improved with the increased content of metal precursor, demonstrating the pivotal role of high metal loading in catalytic performance (*J. Am. Chem. Soc.* **142**, 7116–7127, 2020). Thanks again!

Fig. R3 The SMX degradation performance of Fe@N-C-800 before and after the addition of KSCN. Condition: [Fe@N-C-800] = 50 mg L⁻¹, [H₂O₂]₀ = 1 × 10⁻³ M, [SMX]₀ = 10 × 10⁻⁶ M, [KSCN] = 10 × 10⁻³ M, initial pH = 6.2, T = 25 °C.

Comment 6 In the response 2c, the authors provided the HRTEM to confirm that the Fe nanoparticles were coated by thin carbon layers. The authors also confirmed that

the metallic Fe changed during the reaction. However, the authors also claimed that the use of the pure metallic Fe to do the simulation is reasonable and reliable. The reviewer did not think it is correct.

Answer: Thanks for the comment. The authors really understand the reviewer's concern that the change of catalyst composition might affect the H₂O₂ activation capability and the accuracy of the proposed model in DFT calculations. However, since the work described in this manuscript was the Fenton-like system composed with only the catalyst and H₂O₂, the required electrons for H₂O₂ activation were mainly from the metallic-nanoparticle-boosted carbon surface in the catalyst. In this case, the increased content of oxidized metal composition after reaction was commonly obtained in the similar systems (*Nat. Commun.* **13**, 5365, 2022, *Angew. Chem. Int. Ed.* **58**, 8134-8138, 2019, *ACS Nano* **10**, 11532-11540, 2016, *Adv. Funct. Mater.* **31**, 2106311, 2021, *Environ. Sci. Technol.* **53**, 9081-9090, 2019). Considering that the catalyst composition was constantly changing during reaction, it was very difficult to monitor the catalyst structure and construct the corresponding simulation models at every moment. Therefore, the most practical method is to conduct the DFT calculations based on the pristine structure of the catalyst to predict the active sites, and then combined these results with the detailed characterizations of the used catalyst to find out the potential reaction mechanism and give a reasonable scenario of how the catalyst worked during reaction, which was also a universally accepted method in the previous works (*J. Am. Chem. Soc.* **140**, 12469-12475, 2018, *ACS Catal.* **10**, 14857-14870, 2020, *Adv. Funct. Mater.* **31**, 2007877, 2021). Nonetheless, we believe that the reviewer raised a very important point that has not been thoroughly discussed in the previous publications regarding the reasonable construction of simulation models during reaction, which should be carefully addressed in future works to better reflect the real conditions. Thanks again!

Comment 7 They also claimed that “the required electrons for H₂O₂ activation is mainly from the metal species in catalyst.” and “All the above properties are based on the well-protected Fe nanoparticle structure in the obtained catalyst.” Are these claims

consistent?

Answer: Thanks for the comment. The authors have confirmed that the embedded Fe nanoparticles could donate electrons to the N-doped carbon layers and modulate its electronic structure via the intimately interacted metal-carbon interface, thus favoring the dissociation of H₂O₂ through obtaining electrons from the carbon surface, which had been verified by the experimental result that the catalyst without Fe nanoparticles displayed a drastically decreased activity (**Fig. R4**). And at the meantime, the encapsulation of Fe nanoparticles into protected shells could certainly elevate the metal stability against passivation and leaching (*Acc. Chem. Res.* **51**, 678–687, 2018, *Angew. Chem. Int. Ed.* **58**, 8134-8138, 2019, *ACS Catal.* **11**, 7422–7428, 2021), thus achieving the satisfactory catalytic performance in storage test and cyclic degradation with suppressed Fe leaching. That is to say, the encapsulation of metal crystals into carbon shells could combine the merits of the two compositions (*ACS Catal.* **10**, 14857-14870, 2020) and improve the reusability of Fe nanoparticles and the catalytic performance of carbon. Therefore, the claims raised by the authors are consistent, thanks again!

Fig. R4 The catalytic performance of N-C-800/H₂O₂ and Fe@N-C-800/H₂O₂ systems in SMX degradation. Condition: [Catalyst] = 50 mg L⁻¹, [SMX]₀ = 10 × 10⁻⁶ M, [H₂O₂]₀ = 1 × 10⁻³ M, T = 25 °C.

Comment 8 The authors claimed that “only 0.076 mg L⁻¹ of Fe ions was leached by ICP-OES analysis.” How does the continuously production of H₂O₂ from metallic Fe

species while they did not dissolve?

Answer: Thanks for the comment. The authors should first clarify the fact that in this work the catalyst and H₂O₂ were both added in the solution to trigger the Fenton-like reaction, **but not involved the in-situ H₂O₂ production process during reaction.** Besides, the electron penetration through carbon layer from the embedded Fe nanoparticles to the carbon surface could stimulate the catalytic reaction (*Angew. Chem. Int. Ed.* **60**, 10375-10383, 2021), thus favoring the continuous activation of H₂O₂ to generate [•]OH through obtaining electrons from the carbon surface. It should be noted that neither the Fe species on catalyst surface or dissolved in solution were directly involved in H₂O₂ activation, which had been verified by SCN⁻ poison experiment described in **Comment 5**. This result was also supported by the fluorescence microscopy test in this work, where the fluorescence signals of [•]OH were not found in solution but accumulated on the catalyst surface, further confirming that the H₂O₂ was activated on the catalyst surface but not by the Fe ions dissolved in solution. Thanks again!

We believe that we have addressed the reviewers' comments. We tried our best to improve the manuscript and made corresponding changes in the manuscript. These changes will not influence the content and framework of the paper. And here we did not list the changes but marked in red in revised paper.

We appreciate for Editors/Reviewers' warm work earnestly, and hope that the correction will meet with approval.

Once again, thank you very much for your comments and suggestions.

Sincerely yours,

Mingyang Xing

REVIEWER COMMENTS

Reviewer #6 (Remarks to the Author):

In the study, the researchers proposed a universal pyrolysis approach for the synthesis of high-loading Fe particles possessing an average size of 4 nm. For the concerns about materials characterization highlighted by Reviewer 3, the discrepancies might be attributed to the methodological strategies employed by the authors for sample characterization.

The researchers explained that “the samples for microscopic analysis are collected from the suspension after dispersing the pristine samples in solvent. Therefore, the XRD test for the catalyst collected from the suspension without the sediment was conducted”. It can be guessed that the dispersion process results in a solution composed of two distinct regions: the upper suspension and the lower sediment (Figure S32). The former likely comprises of Fe/C with a lower Fe content and smaller particle sizes, whereas the latter might contain Fe/C characterized by larger particle sizes and a significantly greater Fe content.

The disparity between the Fe content reported from XPS and ICP-OES could potentially stem from this heterogeneity in sample composition. If it is true that ICP is conducted for the whole sample and XPS is conducted for the upper suspension. The assumption here adequately accounts for why the Fe content displayed in the XPS data is 2.3 wt%, whereas ICP-OES indicates a Fe loading of 30.5 wt%. To eliminate the possibility of sample heterogeneity causing the observed inconsistencies, the authors are advised to perform Ar etching for XPS. This technique could effectively remove the proposed "carbon layer", further ensuring the accuracy and reliability of their experiments.

The assumption also explains the observed differences in the XRD patterns between the revised draft and the initial submission. The inconsistency observed might be ascribed to the upper suspension's composition, potentially featuring a lower concentration and a smaller size of Fe particles.

Overall, a potential explanation for the materials' characterization is being proposed herein, some other small comments are listed:

1. Figure S8 appears to show that the majority of Fe in the sample to be amorphous, and the XRD data suggests a highly crystalline nature of the iron. Could this be attributable to the previously mentioned assumption?
2. In the main manuscript, the authors claimed that “the well-protected Fe NPs could be clearly observed at the edge of carbon nanosheet (Supplementary Fig. 8b)”, The crystallinity of the carbon layer appears difficult to distinguish based on Figure 8b.
3. In Figure S11, dopamine loses almost all its mass at around 300 °C in N₂ atmosphere, what is the possible reaction mechanism? I think carbon should not react with N₂. On the other hand, for Fe-PDA, it retained around 30% of its mass at 800 °C, but remember 23.7% of Fe is Fe₃O₄, how does this data align with the ICP analysis?
4. For FeCl₃ 6H₂O TGA analysis, the final mass retained is around 25%, what is the final product, and is it reasonable?
5. With regard to Figure S12, there are numerous minor peaks that have not been assigned or analyzed.

Respective answer to the detailed comments:

Reviewer 6

In the study, the researchers proposed a universal pyrolysis approach for the synthesis of high-loading Fe particles possessing an average size of 4 nm. For the concerns about materials characterization highlighted by Reviewer 3, the discrepancies might be attributed to the methodological strategies employed by the authors for sample characterization.

The researchers explained that “the samples for microscopic analysis are collected from the suspension after dispersing the pristine samples in solvent. Therefore, the XRD test for the catalyst collected from the suspension without the sediment was conducted” . It can be guessed that the dispersion process results in a solution composed of two distinct regions: the upper suspension and the lower sediment (Figure S32). The former likely comprises of Fe/C with a lower Fe content and smaller particle sizes, whereas the latter might contain Fe/C characterized by larger particle sizes and a significantly greater Fe content.

The disparity between the Fe content reported from XPS and ICP-OES could potentially stem from this heterogeneity in sample composition. If it is true that ICP is conducted for the whole sample and XPS is conducted for the upper suspension. The assumption here adequately accounts for why the Fe content displayed in the XPS data is 2.3 wt%, whereas ICP-OES indicates a Fe loading of 30.5 wt%. To eliminate the possibility of sample heterogeneity causing the observed inconsistencies, the authors are advised to perform Ar etching for XPS. This technique could effectively remove the proposed "carbon layer", further ensuring the accuracy and reliability of their experiments.

The assumption also explains the observed differences in the XRD patterns between the revised draft and the initial submission. The inconsistency observed might be ascribed to the upper suspension's composition, potentially featuring a lower concentration and a smaller size of Fe particles.

Answer: We sincerely appreciate reviewer’s professional and reasonable comment.

Just as the reviewer speculated, the differences in XRD patterns collected from the pristine sample and suspension could be attributed to the relative heterogeneity in sample composition. Since the molar ratio of iron salt and dopamine was 2:1 in the preparation stage, it might be inevitable that some iron species were not complexed with the dopamine and then formed the large NPs during the one-step calcination process. In fact, the sample heterogeneity was also very common in the preparation of carbon-supported metal materials through pyrolysis method due to the complicated chemical reactions and physical changes. Similar phenomenon could be observed in previous studies, where the materials with metal nanoparticles ranging from 4 to 15 nm also exhibited the sharp diffraction peaks in XRD patterns (*J. Am. Chem. Soc.* **142**, 7116–7127, 2020, *Angew. Chem. Int. Edit.* **133**, 10463-10471, 2021, *Appl. Catal. B-Environ.* **286**, 119940, 2021, *J. Mater. Chem. A* **7**, 6849–6858, 2019). Besides, as mentioned in our previous responses, the N₂ adsorption-desorption test of the pristine Fe@N-C-800 sample after acid washing demonstrated that the holes caused by Fe leaching were centered at 4-5 nm on the support, which was consistent with the TEM images. The above results confirmed that even though there were a few of large metal nanoparticles contributing to the sharp diffraction peaks in XRD patterns, the part of large metal nanoparticles could hardly affect the statistics of size distribution of the Fe@N-C-800 in this work. Moreover, the control of metal nanoparticle size (even to single-atomic size) was not a challenge in our proposed polymer tethering strategy, which had been explored in our previous responses and further verified the universal and advance of our strategy in fabricating high-loading small-size or even single-atomic Fe@N-C materials. In this work, the authors just selected the most feasible and concise approach to achieve the optimal catalytic performance of the Fe@N-C catalyst with both high reaction rate and H₂O₂ utilization efficiency in Fenton reaction, despite the existence of relative heterogeneity in sample composition.

As for the XPS and ICP-OES characterizations, all the data were collected from the whole pristine sample. The different Fe contents obtained by XPS and ICP-OES test were attributed to the well protected Fe nanoparticles by thin carbon layers (*Nano Res.* **12**, 2341–2347, 2019), which was also verified by the HRTEM test in this work.

To further identify this structure in the catalyst, the authors have explored the XPS depth profiles with different Ar etching times according to the suggestion from the reviewer. As shown in **Fig. R1a**, the Fe signals gradually increased with prolonging etching times due to the removal of carbon layers. The Fe content after etching for 150 s was calculated to be 19.1 wt%, which exhibited several-fold improvements compared with that of the pristine sample. In addition, the existence of Fe^{II} and Fe^{III} species in the high-resolution Fe 2p spectra could be attributed to the electron donation and surface oxidation of Fe nanoparticles (**Fig. R1b**), with increasing etching depth, the characteristic peak of metallic Fe became more obvious. Similar XPS depth profile results were also found in other metal composites with protective layers (*Nat. Commun.* **12**, 3765, 2021, *Surf. Coat. Tech.* **441**, 128599, 2022), which further confirmed the carbon coated Fe structure in our catalyst. According to your suggestion, the relevant result had been added in the revised manuscript. Thank you again!

Fig. R1 a XPS survey spectra and b high-resolution Fe 2p spectra of Fe@N-C-800 with different Ar etching times.

Comment 1 Figure S8 appears to show that the majority of Fe in the sample to be amorphous, and the XRD data suggests a highly crystalline nature of the iron. Could this be attributable to the previously mentioned assumption?

Answer: Thanks a lot for reviewer's valuable comment. The authors agree with the reviewer's opinion that the discrepancies between microscopic analysis and XRD data (collected from the pristine sample) should be attributed to the relative heterogeneity in sample composition. With the decrease of nanoparticle size, the metal crystallinity will generally decline. This phenomenon could be verified by the XRD data of Fe@N-C-800 collected from the suspension, where the diffraction peaks of Fe nanoparticles were greatly weakened compared with that of the pristine sample. Nevertheless, the lattice fringes of 0.210 nm corresponding to the (110) plane of metallic Fe could be still identified in many nanoparticles of Figure S8b, which was consistent with the diffraction peak located at 44.6° in the XRD pattern. Thanks again!

Comment 2 In the main manuscript, the authors claimed that "the well-protected Fe NPs could be clearly observed at the edge of carbon nanosheet (Supplementary Fig. 8b)", the crystallinity of the carbon layer appears difficult to distinguish based on Figure 8b.

Answer: Thanks for the comment. According to the reviewer's suggestion, the carbon layers around Fe nanoparticles with the lattice fringes of 0.335 nm corresponding to the (002) crystal plane of carbon were marked for better understanding (**Fig. R2a**). Meanwhile, it was true that the crystallinity of the carbon layer was not obvious in Fe@N-C-800, which was consistent with the XRD results in Figure S12. With the increase of pyrolysis temperature, the metal nanoparticles became larger, and more carbon would dissolve into metal crystal at high temperature and then precipitate from the C/metal solid solution during the cooling process, thus forming the distinct carbon-wrapped structure (*J. Am. Chem. Soc.* **142**, 7116–7127, 2020). As shown in **Fig. R2b**, the core-shell structure was more obvious in Fe@N-C-1000 due to its large

nanoparticle and increased carbon layers compared with that of Fe@N-C-800, which further confirmed the existence of protective carbon layers on the Fe nanoparticles.

Thanks again!

Fig. R2 HRTEM images of **a** Fe@N-C-800 and **b** Fe@N-C-1000 materials.

Comment 3 In Figure S11, dopamine loses almost all its mass at around 300 °C in N₂ atmosphere, what is the possible reaction mechanism? I think carbon should not react with N₂. On the other hand, for Fe-PDA, it retained around 30% of its mass at 800 °C, but remember 23.7% of Fe is Fe₃O₄, how does this data align with the ICP analysis?

Answer: Thanks for the comment. The weight loss of dopamine during thermogravimetry analysis could be ascribed to the following reasons: i) The boiling point of dopamine was around 337.7 °C, so a lot of dopamine could be vaporized during the temperature range of 300-400 °C. The similar decomposition behavior of dopamine had been reported in previous studies (*ACS Omega*. **6**, 1352-1360, 2021). ii) The carbonization of dopamine under inert atmosphere involved complicated chemical reactions and physical changes (*Nanoscale* **8**, 1770–1788, 2016), which could decompose into carbon monoxide (CO), carbon dioxide (CO₂), hydrogen (H₂), various hydrocarbon (C_xH_y) and nitrogen species (NH₃, HCN, et al.). The approximate reaction mechanism could be described by the Equation R1, which was similar to other carbon sources during the high temperature pyrolysis treatment (*Nat. Commun.* **6**, 6835, 2015, *Energy Convers. Manage.* **89**, 83-91, 2015, *Environ. Sci. Technol.* **47**, 8955-8961, 2013).

As for the Fe-PDA complex, it retained around 30% of its mass at 800 °C in thermogravimetry analysis. The residual composite was our final catalyst (Fe@N-C-800), which had the metal content of 30.5 wt% through the ICP-OES measurement. To further analyze the Fe species in our catalyst, ⁵⁷Fe Mössbauer spectroscopy measurement was conducted and we found that the Fe₃O₄ and iron carbide together accounted for 23.7% of iron species, while the α-Fe with a relative content of 64.0% was the dominant Fe component in Fe@N-C-800 catalyst. Therefore, the data of catalyst yield, metal content and different Fe components presented in the manuscript were reasonable and reliable. Thanks again!

Comment 4 For FeCl₃•6H₂O TGA analysis, the final mass retained is around 25%, what is the final product, and is it reasonable?

Answer: Thanks for the comment. According to the reviewer's suggestion, the XRD characterization was conducted to analyze the final product of FeCl₃•6H₂O after calcination treatment. As shown in **Fig. R3**, the final product exhibited the typical diffraction peaks of Fe₃O₄ (JCPDS No. 75-0033), indicating that Fe₃O₄ was formed after the decomposition of FeCl₃•6H₂O. During the calcination process, the large weight loss below 150 °C could be attributed to the loss of surface adsorbed water and crystalline water caused by the dissociation of FeCl₃•6H₂O. As the temperature increased, the reaction between ferric chloride and residual oxygen species was intensified, resulting in the conversion of ferric chloride to iron oxides (*Hydrometallurgy* **191**, 105187, 2020, *Hydrometallurgy* **202**, 105614, 2021). Moreover, the decomposition behavior and final retained mass of FeCl₃•6H₂O in this work were also consistent with the previous reports (*Materials*. **12**, 3326, 2019, *Adv. Mater.* **29**, 1700707, 2017), further demonstrating that the results reported in this work were reasonable. Thanks again!

Fig. R3 XRD pattern of the $\text{FeCl}_3 \cdot 6\text{H}_2\text{O}$ after calcination treatment.

Comment 5 With regard to Figure S12, there are numerous minor peaks that have not been assigned or analyzed.

Answer: Thanks a lot for the kind comment. Following the reviewer's suggestion, the diffraction peaks of XRD patterns in Figure S12 were further analyzed to give a more clarified demonstration of the material compositions under different pyrolysis temperatures. As shown in **Fig. R4**, the sample calcined at 600 °C exhibited the obvious diffraction peaks of $\text{FeCl}_2 \cdot 2\text{H}_2\text{O}$, accompanied by the small amounts of $\text{FeCl}_3 \cdot 6\text{H}_2\text{O}$, $\text{FeCl}_2 \cdot 4\text{H}_2\text{O}$, FeCl_2 and Fe_2O_3 . With the increase of pyrolysis temperature, the diffraction peaks corresponding to the (110) and (200) crystal planes of Fe nanoparticles became more dominant. When the temperature further elevated to 1000 °C, the diffraction peaks of carbon and iron carbide appeared, indicating the increased crystallinity of these components under high temperature treatment. The relevant modification had been added in the revised manuscript, thanks again!

Fig. R4 XRD patterns of Fe@N-C-X synthesized under different temperatures.

We believe that we have addressed the reviewers' comments. We tried our best to improve the manuscript and made corresponding changes in the manuscript. These changes will not influence the content and framework of the paper. And here we did not list the changes but marked in red in revised paper.

We appreciate for Reviewers' warm work earnestly, and hope that the correction will meet with approval.

Once again, thank you very much for your comments and suggestions.

Sincerely yours,

Mingyang Xing

REVIEWERS' COMMENTS

Reviewer #6 (Remarks to the Author):

All my concerns have been addressed accordingly. Those revisions have greatly improved the depth and rigor of this work. I highly recommend that this version be accepted for publication.

A Response to the Comments of the Editor and Reviewers

Manuscript ID: NCOMMS-22-34544-E

Title: “Polymer Tethering Strategy Toward High-Metal-Loading Nanocatalyst for Dual High-Efficiency Fenton Reaction”

We sincerely appreciate the editor’s timely response and patience concerning our manuscript, and we are truly grateful for the time and effort reviewer 6 has spent in reviewing our manuscript. According to the comments, we revised the manuscript carefully and the point-to-point responses to comments are shown in the following. The revised parts have been marked in red in the manuscript.

Respective answer to the detailed comments:

Reviewer 6

All my concerns have been addressed accordingly. Those revisions have greatly improved the depth and rigor of this work. I highly recommend that this version be accepted for publication.

Answer: We are grateful for the time and effort Review 6 has spent in reviewing our manuscript. We appreciate the reviewer’s recommendation very much.

We appreciate for Editors/Reviewers’ warm work earnestly, and hope that the correction will meet with approval.

Once again, thank you very much for your comments and suggestions.

Sincerely yours,

Mingyang Xing